# Exploring ice sheet model sensitivity to ocean thermal forcing and basal sliding using the Community Ice Sheet Model (CISM)

Mira Berdahl[1,3], Gunter Leguy[2], William H. Lipscomb[2], Nathan M. Urban[3,4], and Matthew J. Hoffman[3]

[1]Department of Earth and Space Sciences, University of Washington, WA, USA
[2]Climate and Global Dynamics Laboratory, National Center for Atmospheric Research, Boulder, CO, USA
[3]T-3 Group, Los Alamos National Laboratory, Los Alamos, NM, USA
[4]Computational Science Initiative, Brookhaven National Laboratory, Upton, NY, USA

**Correspondence:** Mira Berdahl (mberdahl@uw.edu)

**Abstract.** Multi-meter sea level rise (SLR) is thought to be possible within the next few centuries, with most of the uncertainty originating from the Antarctic land ice contribution. One source of uncertainty relates to the ice sheet model initialization. Since ice sheets have a long response time (compared to other Earth system components such as the atmosphere), ice sheet model initialization methods can have significant impacts on how the ice sheet responds to future forcings. To assess this, we generated 25 different ice sheet spin-ups, using the Community Ice Sheet Model (CISM) at 4km resolution. During each spin-up we varied two key parameters known to impact the sensitivity of the ice sheet to future forcing: one related to the sensitivity of the ice-shelf melt rate to ocean thermal forcing, and the other related to the basal friction. The spin-ups all nudge toward observed thickness and enforce a no-advance calving criterion, such that all final spun-up states resemble observations but differ in their melt and friction parameter settings. Each spin-up was then forced with future ocean thermal forcings from 13 different CMIP6 models under the SSP5-8.5 emissions scenario, and modern climatological surface mass balance data. Our results show that the effects of the ice sheet and ocean parameter settings used during the spin-up are capable of impacting multi-century future SLR predictions by as much as 2 m. By the end of this century, the effects of these choices are more modest, but still significant, with differences of up to 0.2 m of SLR. We have identified a combined ocean and ice parameter space that leads to widespread mass loss within 500 years (low friction & high melt rate sensitivity). To explore temperature thresholds, we also ran a synthetically-forced CISM ensemble that is focused on the Amundsen region only. Given certain ocean and ice parameter choices, Amundsen mass loss can be triggered with thermal forcing anomalies between 1.5 and 2°C relative to the spin-up. Our results emphasize the critical importance of considering ice sheet/ocean parameter choices during spin-up for sea level rise predictions and suggest the importance of including glacial isostatic adjustment in ice sheet simulations.

## 1 Introduction

The Antarctic ice sheet (AIS) has the potential to contribute multiple meters to global mean sea level (GMSL) on timescales of several centuries. Yet, Antarctic contributions to sea level rise (SLR) remain the largest source of uncertainty in future projections, particularly on the multi-century timescale (Pattyn and Morlighem, 2020). This is largely due to inadequate model resolution and process representation (Berdahl et al., 2021) and climate uncertainty (Edwards et al., 2021; Seroussi et al.,

2020). Recent projections from the Ice Sheet Model Comparison Project for CMIP6 (ISMIP6) suggest SLR contributions ranging from -7.8 to 30 cm after 100 years under the Representative Concentration Pathway (RCP) 8.5 scenario – spanning the possibilities of either net continental mass loss or growth (Seroussi et al., 2020). Part of this large range is due to poorly known processes in glaciological dynamics (ie. no consensus on what processes to include or how to include them) (Berdahl et al., 2021; Kopp et al., 2017; Bakker et al., 2017). One study that included novel physics (eg. hydrofracture and cliff failure leading to Marine Ice Cliff Instability (MICI)) projected much higher 21st century SLR contributions of more than 1 m (DeConto and Pollard, 2016). Recent discussions by Edwards et al. (2021) and DeConto et al. (2021) highlight the continued debate not only regarding the degree of contribution to sea level from Antarctica over the coming centuries, but also the mechanisms that contribute to mass loss.

Despite these open questions, it remains well-known that the Antarctic Ice Sheet has been losing mass for at least the past four decades, with most of the melt concentrated in the Amundsen and Bellingshausen Seas in West Antarctica (Rignot et al., 2019). This is largely due to a radiative/wind-driven increase in delivery of relatively warm Circumpolar Deep Water (CDW) to the marine-based ice shelves in the West Antarctic Ice Sheet (WAIS) (Rignot et al., 2013; Holland et al., 2019). As the warmer water thins the shelves, the buttressing back-stress they provide to upstream flow is reduced, leading to increased grounded-ice discharge and a subsequent increase in SLR (Fürst et al., 2016; Gudmundsson et al., 2019). Due to a reversing bed slope under much of the WAIS, it is particularly susceptible to positive feedbacks in mass loss. It has been suggested that this process, called the Marine Ice Sheet Instability (MISI), (Weertman, 1974; Schoof, 2007), has already been triggered at glaciers such as the Thwaites and Pine Island Glaciers (Joughin et al., 2014; Favier et al., 2014).

Despite large advances in ice sheet modeling (Pattyn, 2018), the sensitivity of the WAIS to changing climate and its influence on local ocean conditions are challenging for models. One major unknown is the thermal forcing (TF) in the ice shelf cavity itself, which is rarely explicitly resolved in current Atmosphere-Ocean Global Climate Models (AOGCMs). Furthermore, understanding how ocean TF translates to melt rates at the grounding line is still an open question – the functional relationship between TF and melt rates remains speculative. It is therefore a vital question both how forcing will change and how sensitive the Antarctic Ice Sheet is to such forcings.

Borne from the need to systematically quantify the uncertainties in sea level rise from Antarctica, a number of ice sheet model intercomparison projects have been organized. The notion that initialization methods can impact ice sheet simulations is well-known and was explored with 16 different ice sheet models under the initial state intercomparison project (initMIP) framework (Seroussi et al., 2019). ISMIP6 is the most extensive ice sheet model intercomparison project to date (Seroussi et al., 2020). Detailed in Nowicki et al. (2020), 13 ice sheet modeling groups performed a suite of standardized and open experiments aimed at exploring the relative roles of climate forcings, climate warming scenarios, sub-shelf melt parameterizations, multi-model forcing and ice sheet model spread in SLR from Antarctica. ISMIP6 was tasked with generating ocean boundary conditions for stand-alone ice sheet models underneath ice shelves (unresolved in the AOGCMs). To do this, ocean variables were extrapolated horizontally from continental shelves into the ice shelf cavities. Then a melt rate parameterization was used to convert ocean TF to melt rates (more details in Section 1.1). In general, all of the proposed melt rate schemes are trying to account for complex ocean processes (i.e., translating far-field ocean characteristics to sub-shelf melt rates) with simple equations. However, many

parameters used in these approximations are not well constrained, and there remains no scientific consensus on the optimal

functional form of basal melt parameterizations. Indeed, Seroussi et al. (2020) concluded that sensitivity to melt rates was one of the largest sources of uncertainty in future projections of the AIS.

In their extended ISMIP6 study, Lipscomb et al. (2021) found two parameters to be especially important to the sensitivity of the ice sheet. The first, $\gamma_0$, is a constant in the TF parameterization that scales melt rates for a given ocean TF. This parameter controls the strength of ice shelf melt to ocean warming and cannot be uniquely calibrated from observations due to the need

for a poorly constrained TF bias correction term. Preliminary efforts using these parameterizations have focused on capturing the range of these effects by sampling high/moderate/low values for $\gamma_0$ (e.g. Jourdain et al. (2020); Lipscomb et al. (2021); Nowicki et al. (2020)). In their emulation study, Edwards et al. (2021) found that $\gamma_0$ was a similar magnitude, or larger, contributor to uncertainty in their projections of sea level rise as global warming under a particular emissions scenario. The second parameter, $p$, affects the effective pressure near the grounding line, and is specific to how CISM handles basal friction.

$p$ represents the proportion of marine based ice supported by sea water pressure. It essentially dictates the degree of basal slipperiness, particularly in marine-based ice. Both $\gamma_0$ and $p$ are set during the model spin-up (more detail in Section 2) and play a role in the conditioning of the ice sheet. As a result, a new ice sheet spin-up must be run for each combination of $p$ and $\gamma_0$ in future runs. A large range of $p$ and $\gamma_0$ combinations can yield acceptable spun up states that have different sensitivities to future ocean warming. In other words, the choices of $p$ and $\gamma_0$ during the spin-up affect the resulting basal friction field and

sub-shelf conditions, which in turn affect the ice sheet's sensitivity to ocean thermal forcing. Therefore, any future simulations of the ice sheet strongly hinge on what spin-up settings were used because they dictate how strongly the ice sheet will respond to a forcing. Indeed, it is possible that these parameters may be more important to mass loss projections than the future forcing itself.

In this study, we expand the scope of the previous initMIP and ISMIP6 studies by running a 25-member spin-up ensemble of

an ice sheet model, designed to probe the sensitivity of the ice sheet to $\gamma_0$ and $p$ in greater detail. The intent is for our spin-ups to reach steady state with thickness being close to today's observations. Therefore, each spun-up ice sheet state resembles a modern AIS configuration (ie. all spun-up states are valid in this regard, yet non-unique in the $p$ and $\gamma_0$ parameters). Each spin-up member is then forced with future ocean conditions from 13 different CMIP6 models. This allows us to test how future forcings manifest under different ice sheet sensitivities that occur simply by virtue of these two parameter choices. We also

perform synthetically-forced future runs in just the Amundsen region in order to more systematically assess the sensitivity of this critical region to both $p$ and $\gamma_0$. In the next section, we describe in more detail how $\gamma_0$ and $p$ fit into the mathematical framework of our ice sheet simulations.

## 1.1 Two important parameters: p and $\gamma_0$

In this section we summarize the sub-shelf melt rate parameterizations used in the ISMIP6 framework. More details can be

found in Jourdain et al. (2020). There are two main versions of the basal melt parameterization, known as *local* and *non-local*. The *local* parameterization assumes that the sub-shelf melt-induced circulation develops locally to reinforce turbulence and subsequent melting, and represents the influence of ocean stratification. The *non-local* version parameterizes melt rate as the

product of the local TF and the nonlocal TF (ie. sector-average TF). This is rooted in the idea that the melt rate is proportional to both the local TF and the cavity-scale circulation (Holland et al., 2008).

| Constant | Value | Description/Units |
|----------|-------|-------------------|
| $\rho_i$ | 918.0 | Ice density ($kg/m^3$) |
| $\rho_{sw}$ | 1028.0 | Sea water density ($kg/m^3$) |
| $L_f$ | $3.34 \times 10^5$ | Latent heat of fusion ($kg/m^3$) |
| $c_{pw}$ | 3974.0 | Specific heat of sea water ($Jkg^{-1}K^{-1}$) |

**Table 1.** Physical constants used in the quadratic melt parameterizations.

Both the local and non-local versions have two options for calibration, known as *MeanAnt* (Mean Antarctica) and *PIGL* (Pine Island Grounding Line). Parameters are calibrated at the scale of 16 regional sectors. The most basic form (*local*), not shown here, computes basal melt rates beneath ice shelves as a quadratic function of their forcing, with a TF correction suggested by Jourdain et al. (2020). The melt rate parameterization most commonly used in ISMIP6 is the *non-local* version, which takes the quadratic form:

$$m(x,y) = \gamma_0 \times \left( \frac{\rho_{sw} c_{pw}}{\rho_i L_f} \right)^2 \times (TF(x,y,z_{draft}) + \delta T_{sector}) \times |\langle TF \rangle_{draft \epsilon sector} + \delta T_{sector}| \quad (1)$$

where $z_{draft}$ is the ice shelf thickness below the waterline, $TF(x,y,z_{draft})$ is the TF at the ice-ocean interface and $\langle TF \rangle_{draft \epsilon sector}$ is the TF averaged over all the ice-shelves of an entire sector. $\delta T_{sector}$ is a temperature correction for a regional sector used as a means to reproduce observation-based melt rates from observation-based TF, and has a maximum negative value of -2°C. In other words, $\delta T_{sector}$ is used to correct for biases in sparse ocean observations, biases in climate model ocean temperature and

salinity, and in the melt parameterization itself. $\gamma_0$ is an empirical uniform coefficient with units of velocity. The constants, $\rho_i$ (density of ice), $\rho_{sw}$ (density of sea water), $c_{pw}$ (specific heat of seawater), and $L_f$ (ice density) are given in Table 1.

      There is also a non-local, slope-dependent quadratic melting parameterization of the form used in Lipscomb et al. (2021)

$$m(x,y) = \gamma_0 \times \left( \frac{\rho_{sw} c_{pw}}{\rho_i L_f} \right)^2 \times (TF(x,y,z_{draft}) + \delta T_{sector}) \times |\langle TF \rangle_{draft \epsilon sector} + \delta T_{sector}| \times \sin(\theta) \quad (2)$$

where $\theta$ is the local angle between the ice-shelf base and the horizontal. The slope can change as the geometry evolves in

the simulation. The slope dependence is included based on theoretical arguments by Jenkins (2016) and Little et al. (2009), suggesting that basal slope controls the entrainment of heat, therefore affecting melt rates. Jenkins (2016) shows that the basal slope plays a role in driving Ekman pumping and suction analogous to that of the wind stress curl in classical ocean circulation theory. Typically, the steeper the basal slope, the stronger the Ekman pumping.

      Jourdain et al. (2020) generated a distribution of possible $\gamma_0$ values in order to reproduce either the observed present-day

Antarctic melt rates (averaged over a sector), *MeanAnt* calibration, or the (much higher) *PIGL* calibration melt rates. ISMIP6 participants then sampled low (5th percentile), medium (median) and high (95th percentile) $\gamma_0$ values as a nominal exploration of the sensitivity of ice sheet projections to $\gamma_0$. Values of $\gamma_0$ for the non-local and non-local slope parameterizations are shown in Table 2.

| Parameterization | Calibration | 5th percentile $\gamma_0$ (m/s) | Median $\gamma_0$ (m/s) | 95th percentile $\gamma_0$ (m/s) |
|---|---|---|---|---|
| Non-local | MeanAnt | $9.62 \times 10^3$ | $1.44 \times 10^4$ | $2.10 \times 10^4$ |
| Non-local | PIGL | $8.80 \times 10^4$ | $1.59 \times 10^5$ | $4.71 \times 10^5$ |
| Non-local slope | MeanAnt | $1.47 \times 10^6$ | $2.06 \times 10^6$ | $2.84 \times 10^6$ |
| Non-local slope | PIGL | $2.93 \times 10^6$ | $5.37 \times 10^6$ | $2.94 \times 10^7$ |

**Table 2.** Calibrated $\gamma_0$ (m/s) values calculated for the non-local parameterizations in the ISMIP6 protocol in Jourdain et al. (2020) and Lipscomb et al. (2021).

To focus computing resources and analysis on one scheme, we choose to limit this study to the slope-dependent non-local form (Eq. 2), since, at the time the ensemble was run, it was believed to be the most realistic scheme (Jenkins et al., 2018). Since we are testing sensitivity to $\gamma_0$, we are not using a specific calibrated parameter range. The simulations in our paper differ from the ISMIP6 protocols in the treatment of $\delta T_{sector}$. Instead of using the values suggested by Jourdain et al. (2020) to match observational estimates of basal melting in each sector, we tune $\delta T_{sector}$ to obtain melt rates that drive the ice toward the observed ice thickness near the grounding line, as described by Lipscomb et al. (2021). In some basins, this results in basin-average melt rates that differ appreciably from observational estimates. For more details, see Section 3.1 of Lipscomb et al. (2021).

In addition to finding that $\gamma_0$ had a large impact on sea level projections, Lipscomb et al. (2021) found that mass loss from the ice sheet was strongly dependent on the degree of water-pressure support from the ocean. We use a basal sliding law based on Schoof (2005) (Eq. 6) in which the effective pressure exhibits a smooth transition from a finite value to zero at the grounding line. This is given by the following expression (suggested by Asay-Davis et al. (2016)) describing the basal shear stress, $\tau_b$ :

$$\tau_b = \frac{C_p C_c N}{[C_p^m |u_b| + (C_c N)^m]^{\frac{1}{m}}} |u_b|^{\frac{1}{m}-1} u_b, \tag{3}$$

where $C_p$ is an empirical coefficient for power-law behavior, $C_c$ is an empirical coefficient for Coulomb behavior (set to 0.5 as in Asay-Davis et al. (2016) and Lipscomb et al. (2021)), $u_b$ is the basal ice velocity, N is the effective pressure, and $m = 3$ is a power law exponent. (We acknowledge that the choice of $C_c$ can affect the results. This is addressed in Sec. 4.)

Following Leguy et al. (2014), a simple function for the effective pressure that accounts for connectivity between the subglacial drainage system and the ocean is given by

$$N(p) = \rho_i g H \left(1 - \frac{H_f}{H}\right)^p, \tag{4}$$

where $g$ is gravitational acceleration, $H_f = max(0, -\frac{\rho_{sw}}{\rho_i} b)$ is the flotation thickness and $b$ is the bed elevation, defined as negative below sea level. The parameter $p$ varies from zero (no basal water pressure) to one (the subglacial drainage system is hydrologically well connected to the ocean and there is full support near the grounding line). When $p = 0.5$, there is partial support of the ice overburden by subglacial water pressure. This parameterization only accounts for basal sliding for ice grounded below sea level. It does not account for subglacial hydrology in regions where the glacier bed is above sea level. A

hydrology model for CISM is currently in development. In the interior of the ice sheet, and when $p = 0$, this law asymptotes to power-law behavior:

$$\tau_b \approx C_p |u_b|^{\frac{1}{m}-1} u_b. \tag{5}$$

In the grounding line zone, when $p > 0$, the bed provides little resistance to sliding, and the basal shear stress approaches Coulomb friction behavior:

$$\tau_b \approx C_c N \frac{u_b}{|u_b|}. \tag{6}$$

Importantly, under Coulomb behavior, the ice becomes more sensitive to the loss of ice-shelf buttressing (Sun et al., 2020).

Lipscomb et al. (2021) tested the impacts of choosing $p = 0$, $p = 0.5$ and $p = 1$ on sea level contributions. They found differences of up to ∼500 mm in sea level contributions by 2500 compared to runs using a power-law shear stress formulation, concluding that weaker basal friction makes the ice more vulnerable to melt. In this study, we expand on this work by more extensively sampling across $p$ (25 values instead of two) in order to better understand the potential impacts on ice mass loss. We note that while $\gamma_0$ is forcing-related and therefore transferable across ice sheet models, $p$ is a model-internal parameter

and might not be directly transferable. Our formulation with $p$ applies to sliding laws in which basal friction depends on the effective pressure $N$. Other models with similar sliding laws can therefore benefit from this study.

## 2    Methods

### 2.1    Community Ice Sheet Model: Configuration & Spin-up Methodology

We use the Community Ice Sheet Model (CISM), a state-of-the-art 3D, parallel, thermo-mechanical model that runs on a

regular mesh grid (Lipscomb et al., 2019). CISM has participated in various ice sheet model intercomparisons (e.g. MISMIP+ (Cornford et al., 2020), LARMIP (Levermann et al., 2020), ABUMIP (Sun et al., 2020), initMIP (Seroussi et al., 2019) and ISMIP6 (Nowicki et al., 2020; Seroussi et al., 2020)), and its output was comparable to other higher-order ice sheet models, some of which use resolutions of 1 km or higher in the region containing the grounding line.

For continental-scale simulations, ice sheet models are typically run at resolutions of 4 km or coarser (Seroussi et al., 2019).

On century timescales, Lipscomb et al. (2021) found CISM was only moderately sensitive to grid resolution in ocean-forced AIS experiments, concluding that 4 km resolution was comparable to 2 km resolution. Leguy et al. (2021) found that CISM grid resolutions of 2-4 km may be sufficient to represent grounding line migration. Therefore, all continental-scale, Antarctic simulations were run on a uniform 4 km grid and used the following options:

– A depth-integrated higher-order solver based on Goldberg (2011).

– A basal sliding law based on Schoof (2005).

- Grounding line parameterizations for basal shear stress and basal melt rate (Leguy et al., 2014, 2021). Basal melting is applied to partially floating cells in proportion to the floating fraction of the cell, which is diagnosed from the thickness and bed topography.

- A no-advance calving criterion that holds the calving front near its observed location. During forward runs, the calving front is allowed to change location as the ice melts, and it can re-advance, but cannot advance past its original observed location.

- Geothermal heat flux from Shapiro and Ritzwoller (2004).

The original spin-up, taken from Lipscomb et al. (2021), is run with the nonlocal-slope parameterization and $\gamma_0 = 2.06 \times 10^6 \mathrm{m\ yr}^{-1}$ (Table 2). The spin-up method, described in Lipscomb et al. (2021), adjusts a 2D basal friction parameter field ($C_p$) beneath grounded ice and $\delta T_{sector}$ under floating ice in order to match observed ice sheet properties with little drift. The ice sheet is initialized to the present-day thickness using the BedMachineAntarctica data set (Morlighem et al., 2020). The surface mass balance (SMB) is from a late $20^{th}$ century simulation with the RACMO2.3 regional climate model (van Wessem et al., 2018). SMB is held constant using the RACMO2 1976-2016 climatology in the spin-up and forward runs. The basal melt rates are computed directly from the TF climatological data set spanning 1995-2018 from Jourdain et al. (2020), and the non-local slope parameterization described in Section 1.1. As the model is nudged toward observations, the ice thickness gradually evolves to a quasi-steady state. The result is a spun-up state with good agreement between observed and modeled surface velocity (Fig. 1), ice shelf extent, and ice thickness (Fig. 2), except in regions that are known to be out of steady state, such as the Amundsen sector and the Kamb Ice Stream (seen in Fig. 1).

While this initialization procedure works well to keep grounded ice near observed thicknesses and removes low-frequency oscillations associated with slow changes in basal temperature, the sensitivity of the ice sheet is highly impacted by the choice of parameters during forward runs. This study was devised as a way to address this concern directly. Here, we investigate how two key parameters ($p$ and $\gamma_0$) that condition the ice during spin-up affect sea level contributions under future forcing scenarios.

## 2.2 Spin-up ensemble design

In order to explore the effect of $p$ and $\gamma_0$ on the ice sheet sensitivity, a new spin-up must be run for each combination of parameters. We ran a 25-member spin-up ensemble with $p$ and $\gamma_0$ values shown in Table 3. We used a stratified Latin hypercube sampling technique (McKay et al., 1979) from a non-uniform distribution of $p$ and $\gamma_0$. Figure 3 shows the sampling distributions for $p$ and $\gamma_0$. From basic physical arguments, $p$ is constrained to be in the range $[0, 1]$. Previous experimental results (Lipscomb et al., 2021) revealed that the differences in SLR on multi-century timescales between $p = 0$ and $p = 0.5$ are smaller than the differences in SLR between $p = 0.5$ and $1.0$ and are mainly driven by ocean forcing rather than the value of $p$ (see Figure A1 for additional details). This suggests that the space could be explored more efficiently by having a greater sampling density for values near 1. That said, there is no *a priori* mechanistic argument for one end of the range being more physically correct than the other. We chose a truncated power distribution, with weighting heavier toward $p = 1$. Specifically, $\pi(p) = (\alpha + 1)p^{\alpha}$, bounded on $[0, 1]$ with $\alpha = 1.5$ (Figure 3, y-axis).

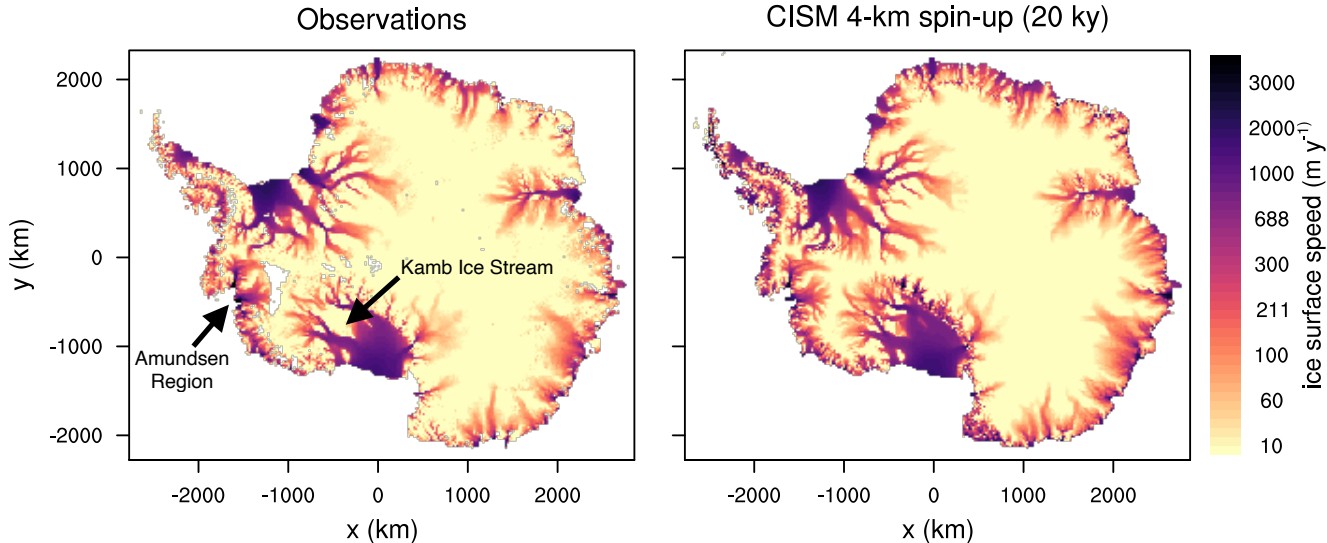

**Figure 1.** Observed (left panel) (Rignot et al., 2011) and modeled (right panel) Antarctic surface speed (m/yr, log scale) for the initial spin-up state used to initialize our 25 ensemble spin-ups (using $p = 0$ and $\gamma_0 = 2062539$). The root-mean-square error between observed and modeled velocity is 128.7 m/yr. White patches represent missing data.

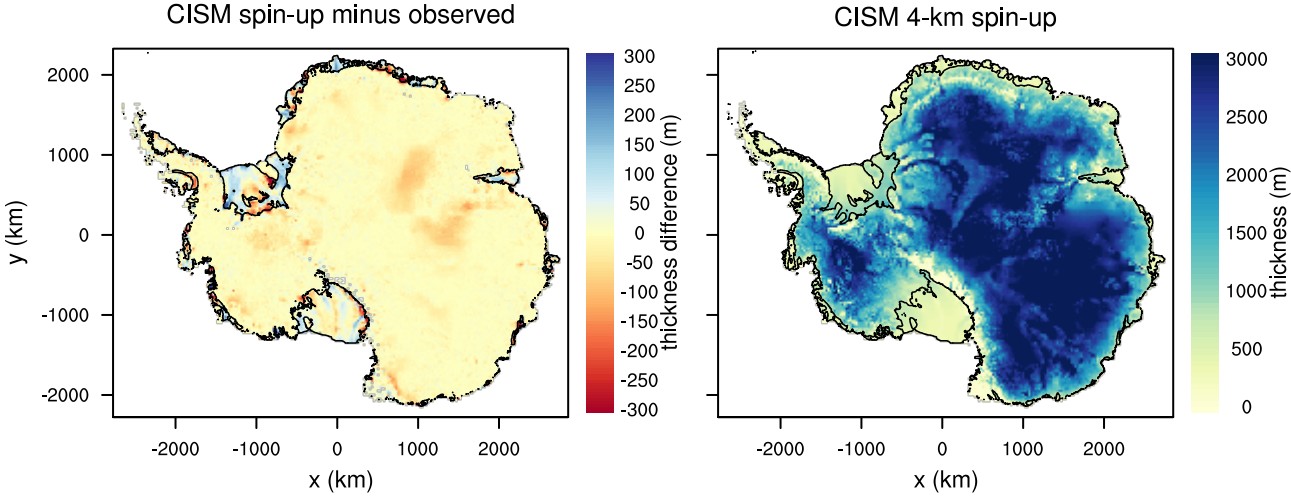

**Figure 2.** Difference between modeled and observed ice thickness (left) and modeled ice thickness (right) for the initial spin-up state used to initialize our 25 ensemble spin-ups (using $p = 0$ and $\gamma_0 = 2062539$), with root-mean-square error 51.8 m
. Observations are from the BedMachine Antarctica data set (Morlighem et al., 2020).

Suggested ISMIP6 calibrated median $\gamma_0$ values for the non-local parameterizations are shown in Table 2. The $\gamma_0$ value
is closely tied to the physical assumptions. With slope dependence, $\gamma_0$ needs to be about 100 times larger. We develop a

distribution of $\gamma_0$ that spans both the $MeanAnt$ and $PIGL$ ranges. We used the distribution $\pi(\gamma_0) \propto \frac{1}{(a\gamma_0-1)^2+1}$, bounded on $[1.47 \times 10^6, 1.0 \times 10^7]$. We chose $a = 3.5 \times 10^{-7}$ such that values would fall preferentially within the MeanAnt range rather than the high end of the PIGL range (Figure 3, x-axis). Note that the upper value is truncated to be $10^7$ instead of $\sim 3 \times 10^7$ as experimentation suggests that the latter value is far too high (N. Jourdain, personal communication, Nov 12, 2020).

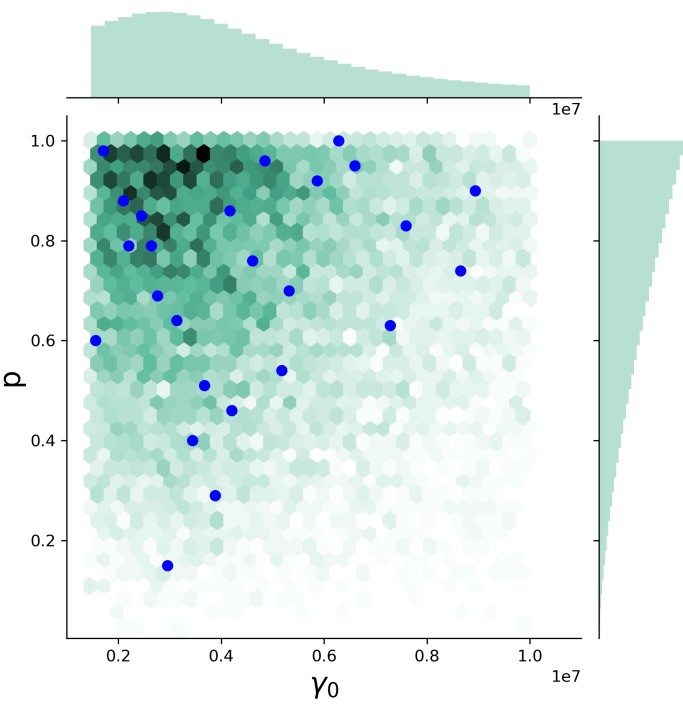

**Figure 3.** Joint sampling distribution for $p$ and $\gamma_0$ used for sampling the spin-up ensemble values (green), and actual chosen $p$ and $\gamma_0$ combinations for this ensemble, using a stratified Latin hypercube sampling from a non-uniform distribution of p and $\gamma_0$ (blue dots).

Each spin-up is branched from the original spin-up in Lipscomb et al. (2021) (Sec. 2.1) and run for at least 10,000 years further. To ensure the spin-up is in steady-state, the mass change rate must not exceed $1 \ \mathrm{Gtyr}^{-1}$. Figure 4 shows ice sheet metrics (ice mass, grounded ice mass, grounded ice area and grounding line flux) for each spin-up ensemble member, as well as current observational estimates. Spin-ups all converge toward similar states, and the total and grounded ice mass and grounded ice area are close to observed (BedMachine Antarctica V2, (Morlighem et al., 2020)) values. As noted earlier, the

RACMO2 historical SMB climatology is used, with spin-up SMB $\sim$2500 Gt/yr compared to observed $\sim$2300 Gt/yr (Mottram et al., 2021; Rignot et al., 2019)). Observational estimates of basal mass balance (BMB) are $\sim$1300 Gt/yr (Rignot et al., 2013; Depoorter et al., 2013), while typical spin-up values are about 630 Gt/yr. This discrepancy between observed and modeled BMB is in large part due to the large $\delta T_{Amundsen}$ values, discussed in greater detail below. Spun-up calving fluxes are around 2000 Gt/yr, while observed values are roughly 1300 Gt/yr (Depoorter et al., 2013). Since the spun-up BMB is reduced from

| Ensemble Member | $p$ | $\gamma_0$ (m/s) | $\delta T_{Amundsen}$ (°C) | Ensemble Member | $p$ | $\gamma_0$ (m/s) | $\delta T_{Amundsen}$ (°C) |
|---|---|---|---|---|---|---|---|
| 1 | 0.15 | 2954923 | -1.6 | 14 | 0.79 | 2200776 | -2 |
| 2 | 0.29 | 3886395 | -1.79 | 15 | 0.79 | 2640377 | -2 |
| 3 | 0.4 | 3440211 | -1.8 | 16 | 0.83 | 7593133 | -2 |
| 4 | 0.46 | 4205230 | -1.98 | 17 | 0.85 | 2450186 | -2 |
| 5 | 0.51 | 3677928 | -2 | 18 | 0.86 | 4167483 | -2 |
| 6 | 0.54 | 5175963 | -2 | 19 | 0.88 | 2098892 | -2 |
| 7 | 0.6 | 1560081 | -1.98 | 20 | 0.9 | 8939808 | -2 |
| 8 | 0.63 | 7280916 | -2 | 21 | 0.92 | 5864477 | -2 |
| 9 | 0.64 | 3139194 | -2 | 22 | 0.95 | 6598244 | -2 |
| 10 | 0.69 | 2760685 | -2 | 23 | 0.96 | 4849305 | -2 |
| 11 | 0.7 | 5321878 | -2 | 24 | 0.98 | 1710386 | -2 |
| 12 | 0.74 | 8654548 | -2 | 25 | 1.0 | 6285577 | -2 |
| 13 | 0.76 | 4609682 | -2 | | | | |

**Table 3.** p and $\gamma_0$ combinations for each ensemble member, along with the $\delta T_{Amundsen}$, showing the end of spin-up correction factor needed for this region.

present-day values as a result of ocean cooling, the calving fluxes must make up the difference, which results in spun-up calving fluxes larger than observed.

Here we choose to prioritize initializing the ice sheet to be close to equilibrium at the expense of a perfect match to the observed ice mass state. For the purposes of this work, we consider the end-of-spinup state to be representative of an ice sheet under 'current' conditions, in that thicknesses are close to today's ice sheet. (We acknowledge that a parameter study does depend on the initial state of the ice sheet (Reese et al., 2020), and our current spin-up strategy does not capture the recent observed Antarctic mass change which could impact the results of this study.) Therefore, forward runs that begin with forcing at 1995-2005 levels are applied directly to the spun-up ice sheet state. The end-of-spinup $\delta T_{Amundsen}$ values for each new parameter setting are given in Table 3. Figure 5 shows the ensemble mean and standard deviation end-of-spinup thickness, velocity, and $\delta T$ values for all regions.

The assumption of an ice sheet at equilibrium is unrealistic, especially for the Amundsen sector. The large negative values in Table 3 reflect this assumption. They show that in order to match the ice sheet's current configuration during spin-up, a large negative thermal correction was necessary to cool the ocean to prevent retreat. To overcome the artificially cooled ocean temperatures in the Amundsen, we also run a set of synthetic experiments targeting only the Amundsen region (further details in 2.4). More discussion on the TF correction factors in Table 3, specifically what they imply with respect to our assumption about a 'current' state, can be found in Section 3.

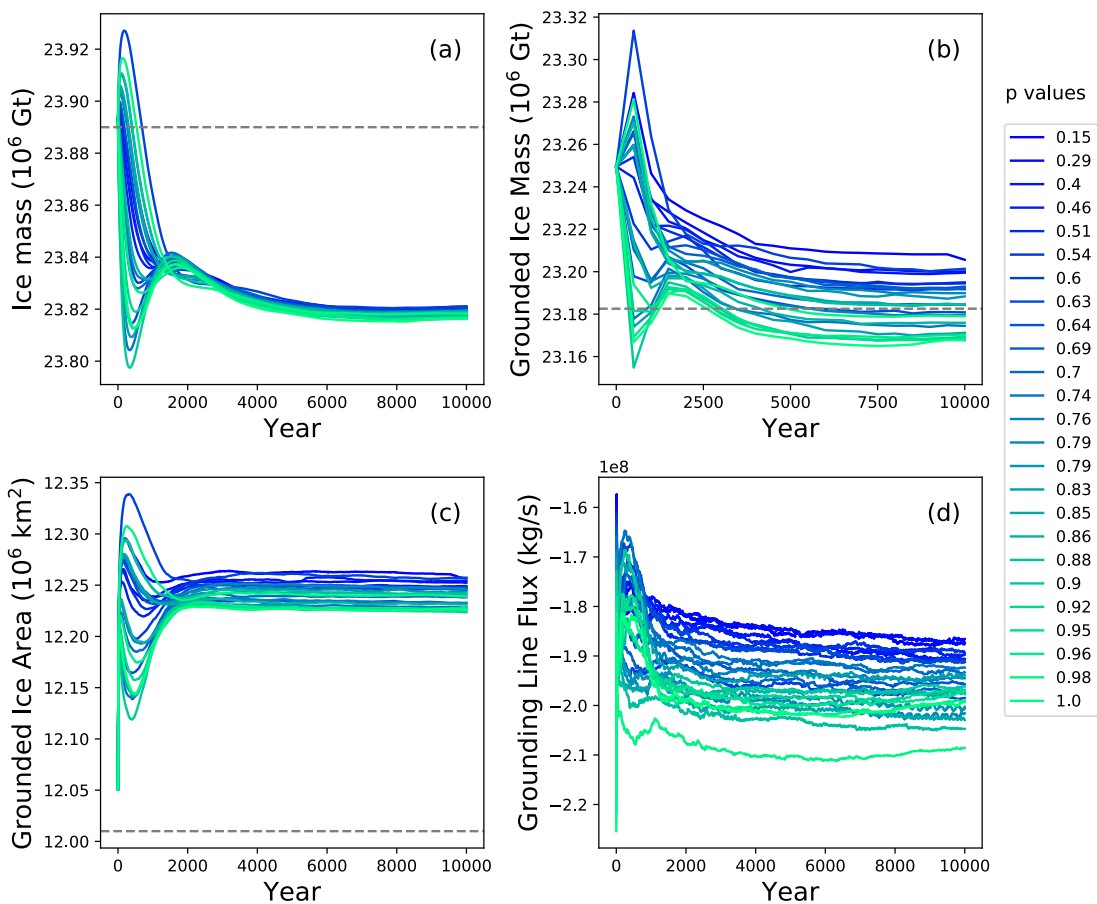

**Figure 4.** Time series for ice mass (a), grounded ice mass (b), grounded ice area (c) and grounding line flux (d) for the 25 spin-up ensemble members. Legend indicates the value of $p$ for each ensemble member. Grey dashed lines indicate observational estimates, as calculated from the BedMachine Antarctica V2 dataset (Morlighem et al., 2020). Note that the frequency of model output is sparser in panel (b) because it was derived using variables with less frequent output.

## 2.3 Forward simulations: CMIP6 SSP58.5

To assess the effect of $p$ and $\gamma_0$ on the sensitivity of the ice sheet in a multi-model framework, forward simulations were forced using AOGCM-derived ocean conditions. Since both of these parameters relate to ice shelf behavior, and in order to focus on the effects of ocean forcing in the forward runs, SMB is held constant at historical values. TF was computed from 13 CMIP6 climate models and applied as anomalies to each spun-up ice sheet state. Specifically, 3D fields of temperature, salinity and density were extracted from 13 CMIP6 climate models for the high emissions SSP8.5 scenario (Table 4) for two decadally-averaged time slices: 1995-2005 and 2090-2100. These were then area-averaged according to the Linear Antarctic Response to basal melting – Model Intercomparison Project (LARMIP) basins (Levermann et al., 2020): Antarctic Peninsula (AP), Weddell

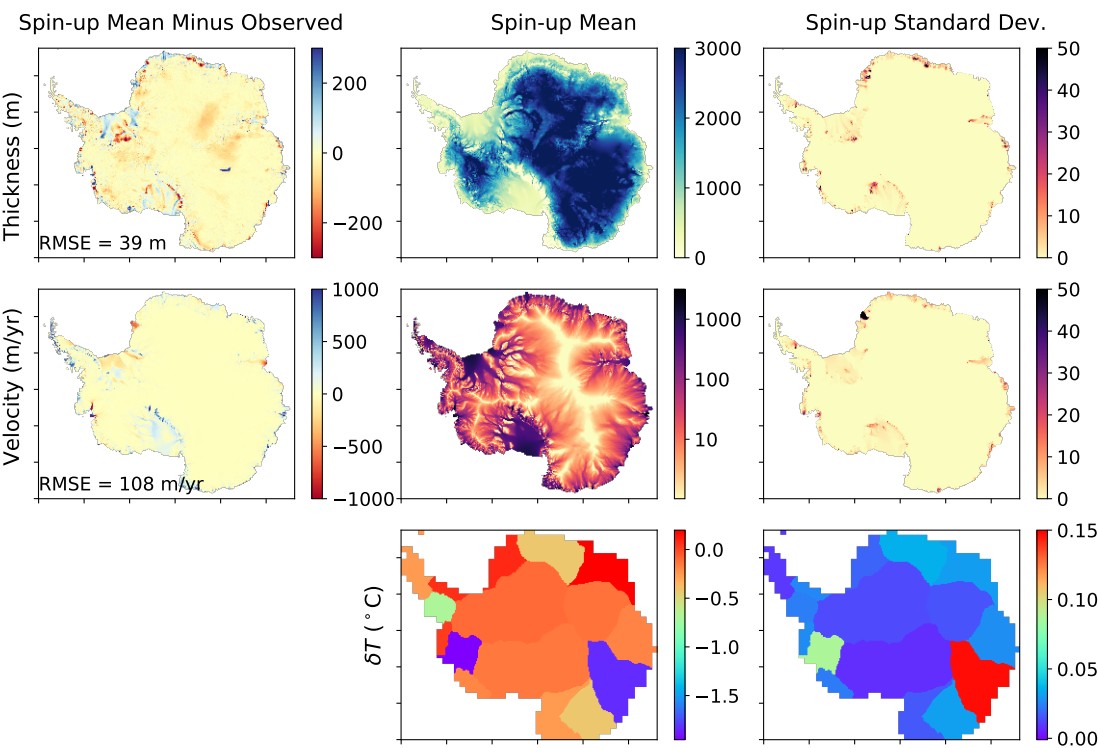

**Figure 5.** End-of-spinup statistics: Thickness (top row), surface velocity (center row) and $\delta T$ values (bottom row). Columns show the end-of-spinup mean minus observed values (left column), spin-up mean (center column) and standard deviation (STD) (right column). Spin-up mean and STD are computed across all 25 $p$ and $\gamma_0$ ensemble members. Observed thickness is from the BedMachine Antarctica V2 (Morlighem et al., 2020), and observed velocities are from Mouginot et al. (2014).

(aka Filchner–Ronne), Amundsen, Ross, and East Antarctic (Fig 6) and interpolated onto the CISM grid (30 depth layers from

0 to 1800 m at 60 m intervals). The TF was then computed by taking the difference between the *in situ* ocean temperature and the *in situ* freezing temperature.

CISM reads the midpoint of the depth grid. The TF at the lower ice surface is then linearly interpolated between the two adjacent TF values. In the case that the ice draft is located above the top level or below the bottom level, the nearest TF value is used. In forward runs, CISM is forced with a TF anomaly. Therefore, we subtracted the 1995–2005 mean TF profile from

the 2090-2100 mean TF profile which gave our 2090–2100 TF anomaly profile. CISM anomalies in the future runs begin with zero anomaly at all depths, and monotonically change at each depth level to the final 2090-2100 TF anomaly profile, shown in Figure 7.

| CMIP6 Model Name | Country | Atmos. Resolution (lon x lat) | Ocean Resolution (horizontal) | Ocean Vertical Levels | Key Reference |
|---|---|---|---|---|---|
| BCC-CSM2-MR | China | $1.9° \times 1.3°$ | $1° \times 1°$ | 40 | Wu et al. (2019) |
| CAMS-CSM1-0 | China | $1.1° \times 1.1°$ | $1° \times 1°$ | 50 | Xin-Yao et al. (2019) |
| CESM2 | USA | $1.3° \times 0.9°$ | $1° \times 1°$ | 60 | Danabasoglu et al. (2020) |
| CNRM-CM6-1 | France | $1.4° \times 1.4°$ | $1° \times 1°$ | 75 | Voldoire et al. (2019) |
| CNRM-ESM2-1 | France | $1.4° \times 1.4°$ | $1° \times 1°$ | 75 | Séférian et al. (2019) |
| CanESM5 | Canada | $2.8° \times 2.8°$ | $1° \times 1°$ | 45 | Swart et al. (2019) |
| EC-Earth3 | Europe | $0.7° \times 0.7°$ | $1° \times 1°$ | 75 | Döscher et al. (2021) |
| EC-Earth3-Veg | Europe | $0.7° \times 0.7°$ | $1° \times 1°$ | 75 | Wyser et al. (2020) |
| GFDL-CM4 | USA | $1° \times 1°$ | $0.25° \times 0.25°$ | 75 | Held et al. (2019) |
| GFDL-ESM4 | USA | $1.3° \times 1°$ | $0.5° \times 0.5°$ | 75 | Dunne et al. (2020) |
| IPSL-CM6A-LR | France | $2.5° \times 1.3°$ | $1° \times 1°$ | 75 | Lurton et al. (2020) |
| MPI-ESM1-2-HR | Germany | $0.9° \times 0.9°$ | $0.4° \times 0.4°$ | 40 | Müller et al. (2018) |
| NESM3 | China | $1.9° \times 1.9°$ | $1° \times 1°$ | 46 | Cao et al. (2018) |

**Table 4.** Names, resolutions and references for the CMIP6 models used in this study.

Thus, each spun-up CISM state (25 $p$ and $\gamma_0$ ensemble members) is branched into 13 forward runs, all forced by CMIP6-derived TFs under the SSP5-8.5 scenario. The forward runs are extended for another 400 years using constant 2090–2100 mean forcing profile, such that the full effects of end-of-century forcings are realized.

### 2.4 Forward simulations: Synthetic perturbations in the Amundsen Sea Sector

The glaciers in the Amundsen region have lost more mass than any other sector over the past several decades (Paolo et al., 2015), yet the thresholds and projections of future loss are still not well constrained (Nias et al., 2019). Therefore, in addition to the CMIP6-forced ensemble, we ran a set of synthetically-forced CISM runs, where TF anomalies are applied only in the Amundsen Region in order to explore parameter and forcing settings that lead to Thwaites mass loss or collapse. We ran forward simulations with a maximum TF anomaly of 1°C, 1.5°C and 2°C, applied uniformly with depth to the Amundsen region only, while the other regions are kept at zero TF anomaly for the duration of the run. The Amundsen anomaly is ramped up linearly starting from zero in the 1995–2005 period to the maximum value of the experiment (1°C, 1.5°C and 2°C) at the 2090–2100 mean. This final maximum forcing is then extended, remaining constant for another 400 years. These synthetic forcings are applied to the same spin-up ensemble used in the CMIP6 SSP5-8.5 experiments.

As discussed in Section 1.1, $\delta T_{sector}$ is a temperature correction, with units of temperature, for a regional sector used to reproduce observation-based melt rates from observation-based TF. It is important to note the final $\delta T_{Amundsen}$ values in Table 3 in the context of these synthetic TF experiments. The $\delta T_{Amundsen}$ corrections are consistently negative with values

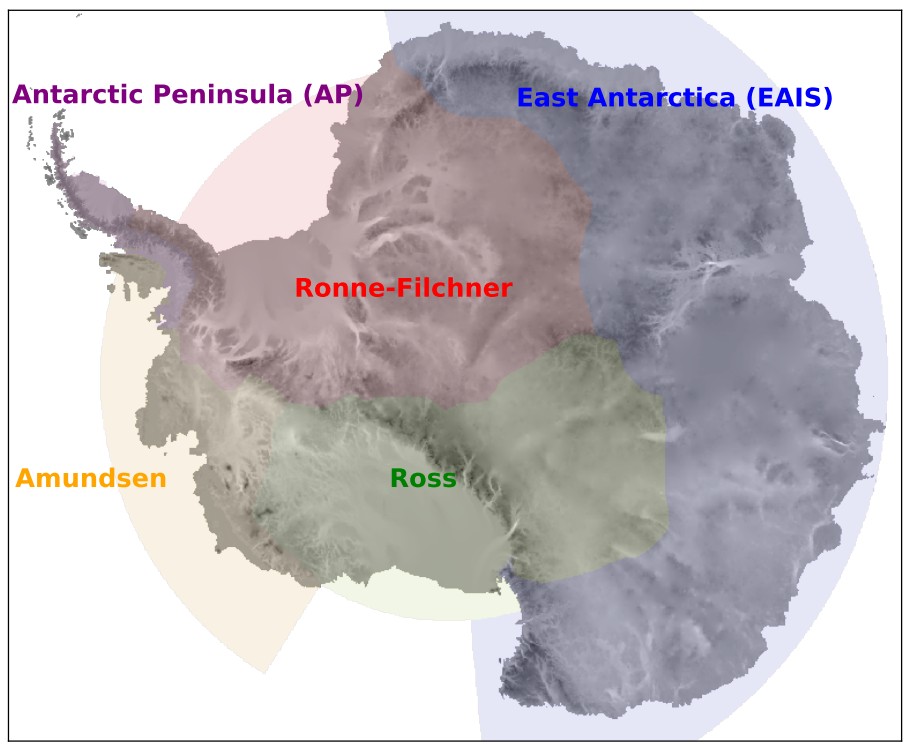

**Figure 6.** Map of basins used in this study, based on the LARMIP delineations Levermann et al. (2020).

ranging from -1.6°C to -2°C. This means that significant cooling is needed to slow grounding-line retreat that occurs under
climatological TF during the spin-up. Therefore, the spun-up melt rates in the Amundsen are lower than observed. Negative
values of $\delta T_{sector}$ may also be compensating for other errors such as biases in climatology or the misplacement of ocean
heat. Lipscomb et al. (2021) posit that another possibility for such large temperature corrections in the Amundsen Sea is that
the TF derived from the 1995–2018 climatology used in their spin-up exceeds the forcing that was typical in the mid 20th
century and before. In this case, negative $\delta T_{sector}$ would be correcting for the recent warming, to generate melt rates closer to
pre-industrial values. Therefore, in forward runs we would need a relatively large TF anomaly ($\sim$2°C) to raise melt rates to
observed present-day values. In that sense, the Amundsen anomaly experiments can also be viewed as estimates of committed
SLR under warming that has already occurred. Thus, we consider our synthetic experiments ranging from 1-2°C TF anomaly
to be physically sensible.

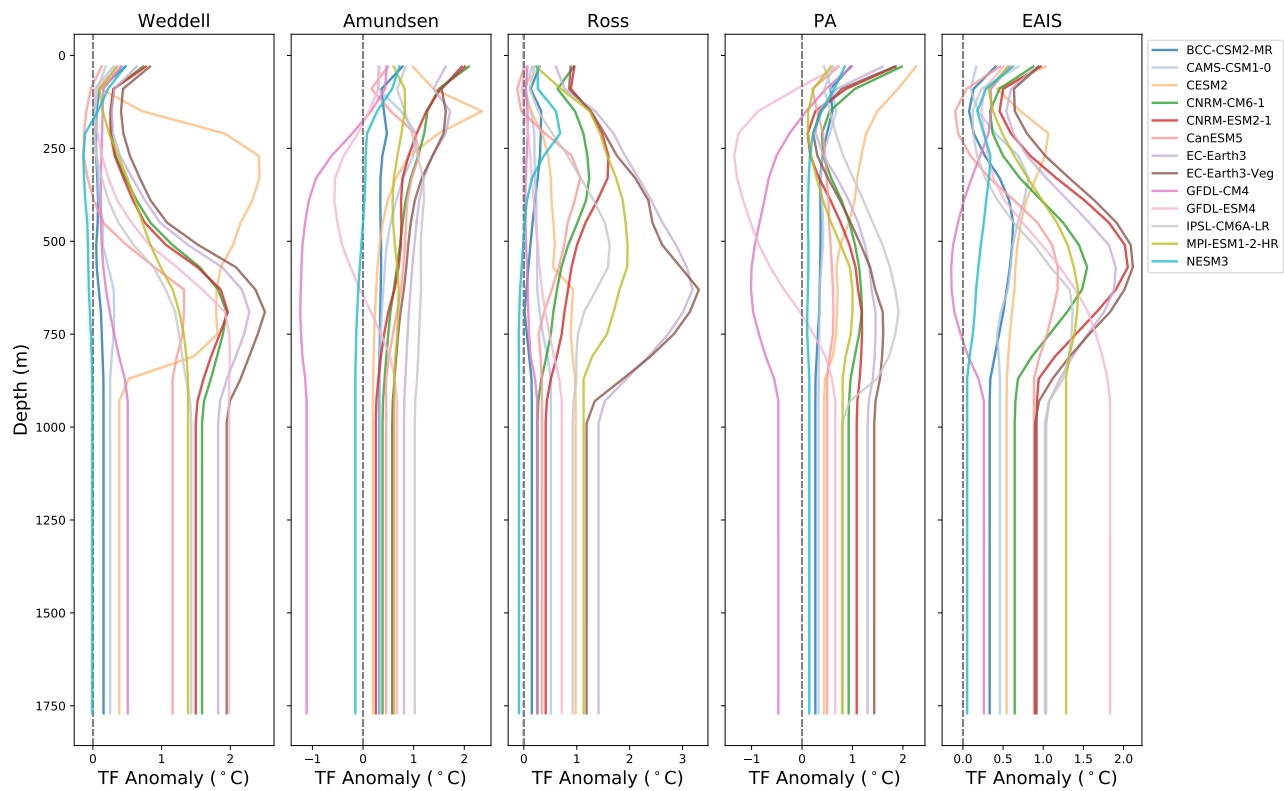

**Figure 7.** Final thermal forcing (TF) anomaly profile for each basin. The TF anomaly begins at zero at all ocean levels. The anomaly grows linearly at each level until it reaches the mean 2090-2100 anomaly. After this, the 2090-2100 mean value is held constant for another 400 years.

## 3 Results of Forward Experiments

### 3.1 CMIP6 TF simulations

#### 3.1.1 Continental Results

Given that the distributions of $p$ and $\gamma_0$ were non-uniform by design, our ensemble does not have a physically meaningful prior (e.g., the distribution on $p$ was intentionally chosen to over-sample the more sensitive values of p, possibly favoring high SLR more than physically warranted). Therefore, the results presented below such as the predicted ranges of SLR should not be over-interpreted. Similarly, the summary statistics shown in Figure A3 are presented to describe the qualitative behavior of the sea level rise.

The CMIP6-forced forward experiments result in a wide range of final sea level after 500 years, depending on the parameter and forcing combinations used (Table A1, Figure 8). The final SLR across all parameters and model forcings ranges between

2 mm and 300 mm after 100 years, and 47.5 mm and 3.17 m after 500 years. Critically, the combined choice of $p$ and $\gamma_0$ alone has the potential to generate large differences in final SLR contributions. Examining the absolute range of final SLR for a given forcing, the choice of $p$ and $\gamma_0$ causes anywhere from a modest difference of 70 mm (NESM3) to a large difference of almost 2 m (EC-Earth3-Veg, (Table A1 and Fig. A2)). For any given CMIP6 forcing, the choice of $p$ and $\gamma_0$ produces a 2–3 fold change between the highest and lowest final SLR value. The choice of $p$ and $\gamma_0$ has a more limited, but still significant impact on SLR after 100 years. The largest difference in SLR after 100 years for a given model is 215 mm (EC-Earth3-Veg), and the smallest is 3.1 mm (NESM3). Therefore, the impacts of the choice of $p$ and $\gamma_0$ during spin-up could mean the difference between basin-wide ice collapse or not on multi-century timescales. Even though the differences are less pronounced at year 100 than year 500, they still constitute critical impacts on end-of-century sea level estimates. The difference of 0.2 m for the EC-Earth3-Veg forced run, for example, has is highly relevant to societal decision making for low-lying coastal regions. It is worth noting that since the spin-up method can produce a steady state with a delayed response to warming, the differences seen after 100 years may be underestimated. The ensemble spread of SLR for all ensemble members ($p$ and $\gamma_0$ combinations) after 100 and 500 years of simulation are further illustrated in Figure 8. The models with the smallest spread and lowest SLR (BCC-CSM2-MR, CAMS-CSM1-0, GFDL-CM4 and NESM3) are also those with the weakest forcing, particularly at $\sim$250–700 m depth (approximate depths of grounding lines) in the largest regions (Weddell, EAIS and Ross) (Figure 7). The EC-Earth3 models generate the strongest forcing at the grounding line depths, and therefore produce the highest SLR in the Weddell, Ross and EAIS sectors.

In general, the EC-Earth3-Veg model produces the largest SLR after 500 years, while the NESM3 model produces the least SLR (Figure A3(a)).The slope of the curves in the log-log plot (Figure A3(b)) indicates the scaling of SLR. Across all models there is little to no change initially in sea level because the forcing is still minimal as it begins to ramp up. This is followed by an abrupt change to a nonlinear increase in sea level for about the first 100 years, concurrent with a linear ramp-up of TF. Then, after 100 years, when the forcing becomes constant and is no longer ramping up, the sea-level increase becomes roughly linear. This pattern is also illustrated in Figure A3(c), where the SLR rate for the model means shows swift acceleration in the first 100 years of the simulation. This is followed by a steadying in SLR rates once the forcing becomes constant. In the case of the EC-Earth models, the rate of change in Antarctic contributions to sea level reaches $\sim$4 mm/year after 100 years. This exceeds the current observed rate of global sea level rise of 3.7 mm/year, which includes all global sources over the period 2006-2018 (Fox-Kemper et al., 2021). In other words, these results suggest that under some model forcings, the rate of contribution to sea level from Antarctica (currently modest) could become comparable to the current global rates by the end of the century.

The qualitative structure of the final sea level contribution as a function of $\gamma_0$ and $p$ is similar across all models, though the magnitudes of mass loss are different across all models (Figure 9). For each model forcing, low $\gamma_0$ values produce little sea level rise, while high values produce the most. The final continental sea level contribution in these experiments depends much more on the variation of melt rate with $\gamma_0$ than on the change in hydrological connectivity near the grounding line with $p$. The ensemble mean correlation ($R^2$) value between final SLR and $\gamma_0$ is 0.93, whereas $R^2$ with $p$ is only 0.15. The linear fits, along with model-specific $R^2$ value (Figure 10), show the same story across all models: On the continental scale, $\gamma_0$ is a much stronger predictor of SLR than is $p$. High values of $p$ alone are not sufficient to force a strong sea-level response, indicating

that the power-law regime dominates in the basal sliding law at the continental scale. However, Joughin et al. (2019) showed
that Thwaites and Pine Island Glaciers exhibit Coulomb-sliding behavior, suggesting that a regional analysis is necessary.

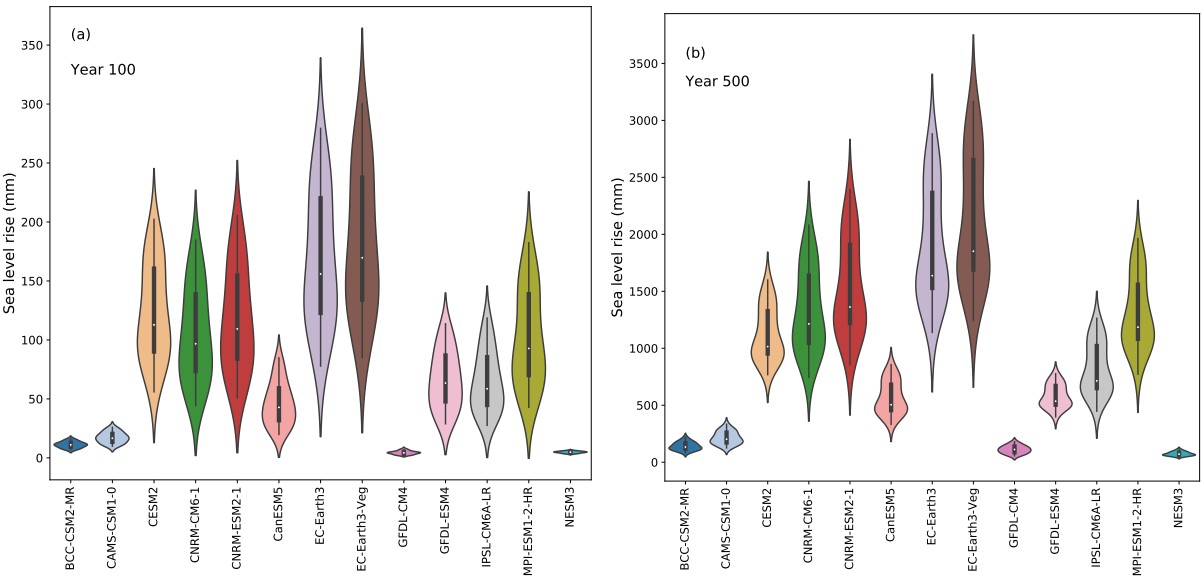

**Figure 8.** Model spread of SLR after (a) 100 years and (b) 500 years. Note different y-scale in the panels.

### 3.1.2 Regional Results

When we analyze the SLR by region, we find that most CMIP6-forced runs give a SLR signal dominated by ice loss from the
Weddell and Ross regions, and to a lesser extent the EAIS (Figure 11). The regions that contribute least to SLR are the AP
and, perhaps surprisingly, the Amundsen. Whereas some models produce strong ocean TF in the Weddell and Ross regions
(up to ∼2°C and ∼3°C respectively at GL depths), the maximum forcing in the Amundsen and AP is fairly weak (∼1°C and
∼1.5°C) (Figure 7). This magnitude of forcing in the Amundsen, coupled with the large regional TF corrections (-1.6°C to
-2°C, Table 3) generated during spin-up, together result in minimal mass loss. The highest model-mean SLR contribution from
the Amundsen region remains below 200 mm.

As with the continent-wide assessment, the regional SLR dependence on $p$ and $\gamma_0$ appears more strongly controlled by
$\gamma_0$ than $p$, particularly when forcing is sufficient to generate large sea level contributions. Specifically, $R^2$ values describing
the correlation strength between sea level contributions and $p$ or $\gamma_0$ are shown in Figure 12. There is a consistently strong
dependence (high $R^2$ values) on $\gamma_0$ and low dependence on $p$ for the Weddell and Ross regions. These regions are also those
that produce the most SLR. The EAIS, though generally generating less SLR, tends to follow the same pattern, with one
exception where correlation with $\gamma_0$ is low (<0.2). This occurs with a CMIP6 model (GFDL-CM4) which produces little SLR
(<20 mm after 500 years) in the region due to weak TF anomalies. Again, the Amundsen and AP show little contribution to
SLR, and also tend to have a weaker correlation with $\gamma_0$, and in some cases show strong correlation ($R^2 \sim 0.8$) with $p$. In the

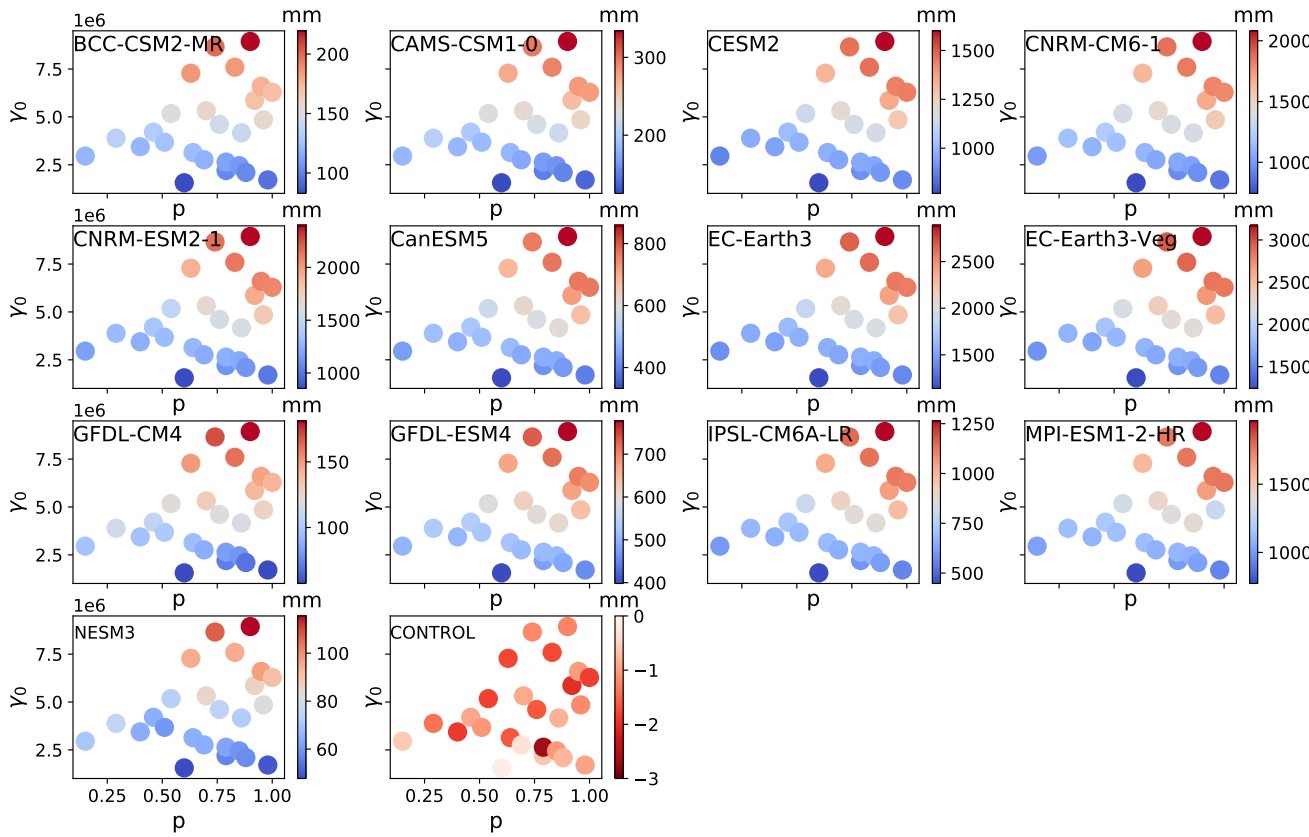

**Figure 9.** Final SLR for each model, $\gamma_0$ and $p$ combination. Note the different colorbar scales for each model.

AP region, the dependence on $p$ is stronger for climate forcing leading to more than 8 mm of SLR (BCC-CSM2-MR, CESM2, CNRM-CM6-1, CNRM-ESM2-1, EC-Earth3, EC-Earth3-Veg). Figures A4 to A8 show all final SLR values for each model as a function of $p$ and $\gamma_0$, along with their best linear fits.

In the Amundsen region, there appears to be a break-point in final SLR as a function of $\gamma_0$ (Figure A5). For $\gamma_0 < 5 \times 10^6$ m/s, sea level remains nearly constant, in some cases rising minimally. For $\gamma_0 > 5 \times 10^6$ m/s, melt rates become large enough that mass loss begins to ramp up as $\gamma_0$ increases. In the case of the warmest (EC-Earth) models, close to half a meter of sea level increase is achieved under high $\gamma_0$ and high $p$ settings in the Amundsen. With cooler AOGCMs (e.g. GFDL-CM4, NESM3) the same high $p$ and $\gamma_0$ settings are not capable of promoting mass loss. This change in behavior with higher $\gamma_0$ in the Amundsen

is likely a result of multiple factors. First, the melt rates generated with lower $\gamma_0$ values are insufficient to push the ice into deeper retrograde bed-slope regions, and second, the melt rates computed with lower $\gamma_0$ values are insufficient to overcome the large negative regional TF corrections resulting from the spinup methodology. Both of these issues will be elaborated in the next section and in Section 4. Further experiments designed to target Amundsen behavior under higher (than CMIP6) forcing are explored in more detail in the following section.

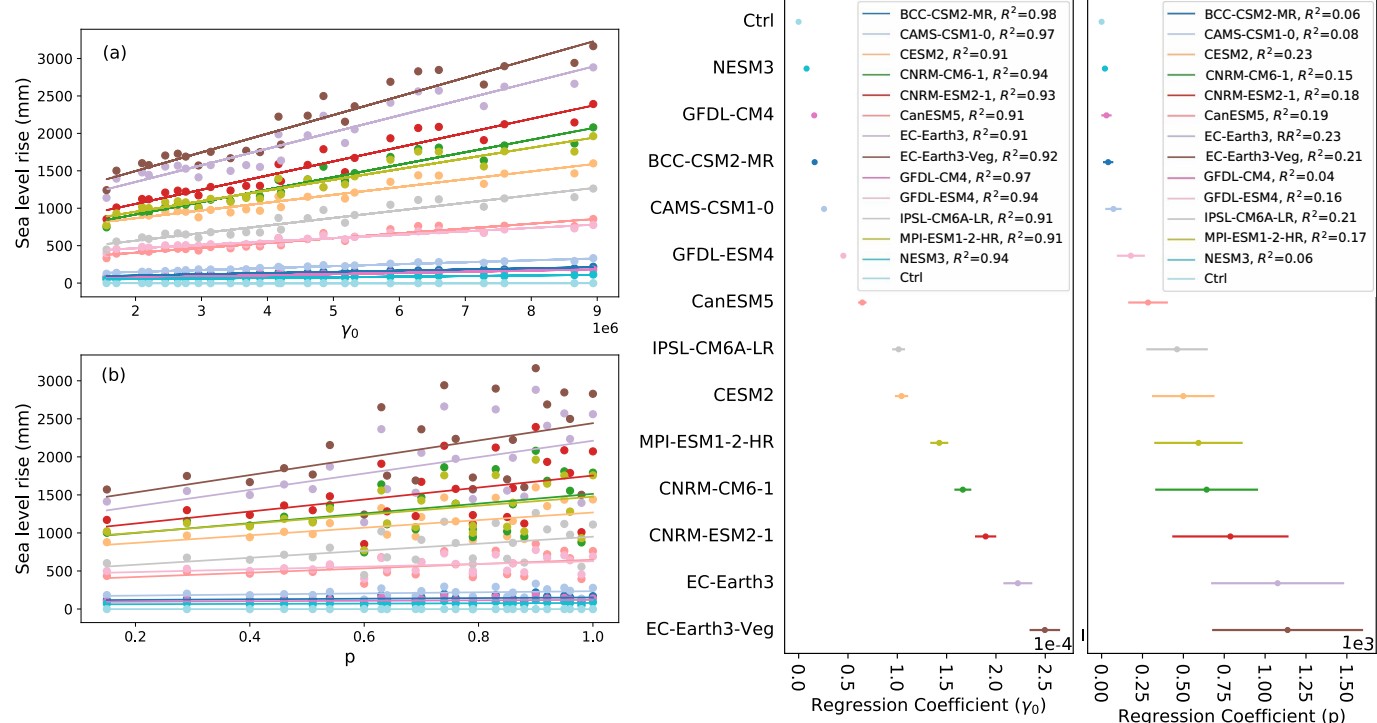

**Figure 10.** Continental SLR as a function of (a) $\gamma_0$ and (b) $p$ with best linear fits. Panels on right show the regression coefficients for the model fits along with their error bars. The $R^2$ value associated with the best fit line is also shown in the legends.

## 3.2 Synthetic TF perturbations in the Amundsen Sea Sector

We explore the sensitivity of the Amundsen sector using regionally-targeted synthetic TFs. Rapid ice retreat in this region has been observed in the past several decades (Rignot et al., 2019), and it has been suggested that Thwaites Glacier collapse may already be underway (Joughin et al., 2014). The modest response in the Amundsen sector in the CMIP6-forced ensemble can be attributed to the weak forcing in almost all the AOGCMs in this region (Figure 7), along with the large (-1.6°C to -2°C) TF correction. As discussed in Section 2.3 and in Lipscomb et al. (2021), in order for the spin-up to match the ice sheet's current configuration, a large negative thermal correction was necessary to cool the ocean to prevent retreat. However, there is strong evidence that the Amundsen Sea Embayment has recently been warming (Rignot et al., 2019; Jenkins et al., 2018; Mouginot et al., 2014). Thus, the assumption of an ice sheet at equilibrium may be a poor assumption for the Amundsen sector. Therefore, we ran a set of synthetic experiments targeting the Amundsen region, described in detail in Section 2.4.

We find that the differences between the 1°C and 1.5°C experiments are fairly minimal over the course of the whole experiment, with the final SLR reaching only ∼100 mm and ∼200 mm respectively (Figures 13 and A9). The 2°C experiment,

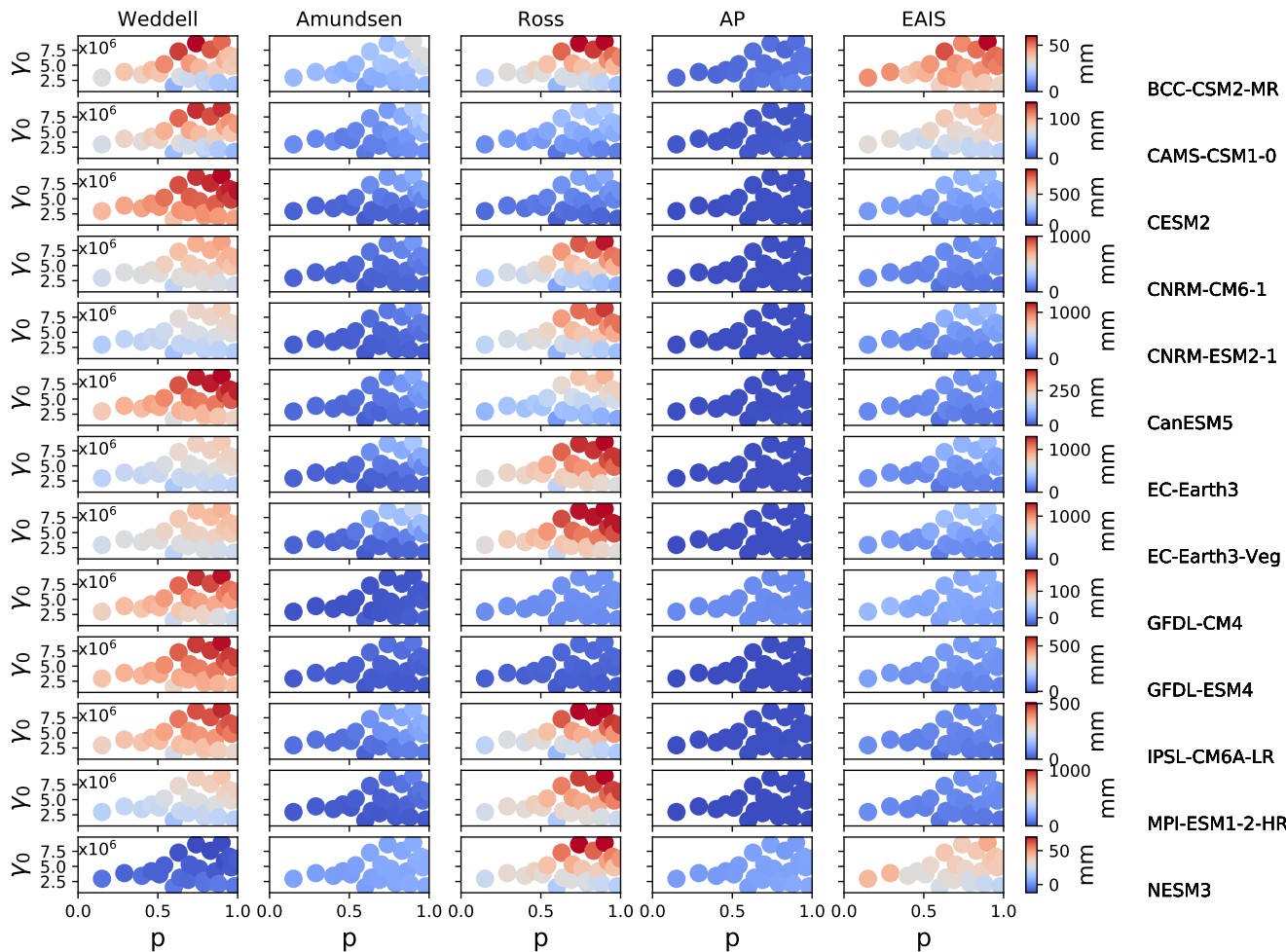

**Figure 11.** Final (500 year) regional SLR contribution as a function of $\gamma_0$ and $p$. Rows show the CMIP6 model used in the forcing. Columns show the region. Note the different colorbar scales for each model.

however, generates over 1.2 m of SLR by the end of the simulation. This indicates almost a 12-fold increase in sea level contributions between the 1°C and 2°C experiments after about 500 years. Such a large disparity in mass loss between experiments only appears after several hundred years of run time. For example, in year 100, the difference between the SLR contributions for the 1°C and 2°C experiments is only two-fold (∼15 mm and ∼30 mm respectively). From 250-350 years, the 2°C experiment shows the greatest acceleration in sea-level contributions (Figure 13). This lag between forcing and sea-level rise is expected, as it has been shown that ice shelf thinning takes place before cumulative mass loss is observed (Hoffman et al., 2019; Jenkins et al., 2018; Mouginot et al., 2014). We suspect that the rapid acceleration of mass loss after year 300 in the 2°C experiment is mostly related to MISI activation, and is exacerbated as the ice ungrounds from high topographic seafloor points (Fig. 14).

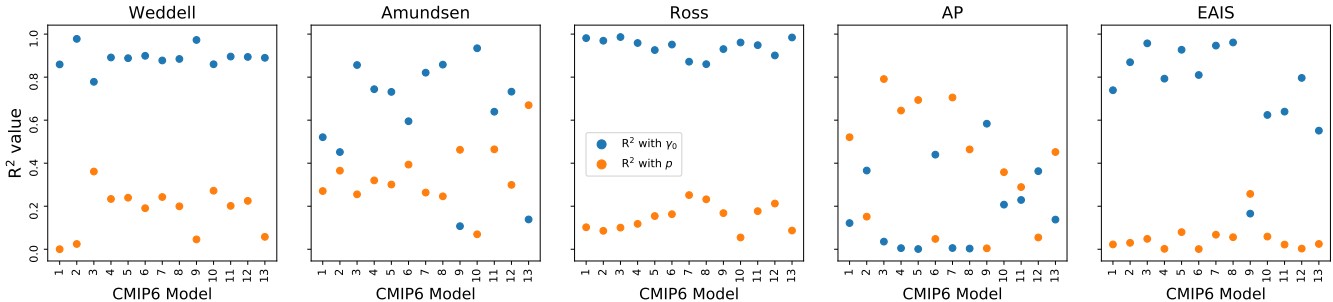

**Figure 12.** $R^2$ correlation values for fits between final (year 500) SLR and $p$ (orange) and $\gamma_0$ (blue). Panels distinguish region. Each blue/orange regional pair represents one CMIP6 model. The correlations are generally higher with $\gamma_0$ than with $p$, particularly in the Weddell, ross and EAIS regions. In the Amundsen and AP, the TF anomalies are generally weaker and the signal becomes less clear. The corresponding order of CMIP6 model from 1 to 13 is: BCC-CSM2-MR, CAMS-CSM1-0, CESM2, CNRM-CM6-1, CNRM-ESM2-1, CanESM5, EC-Earth3, EC-Earth3-Veg, GFDL-CM4, GFDL-ESM4, IPSL-CM6A-LR, MPI-ESM1-2-HR, NESM3.

Despite stronger regional forcing than in the CMIP6 runs, the correlation between $\gamma_0$ and SLR in the synthetic Amundsen runs is not as strong as that seen in the Ross and Weddell regions in the CMIP6-forced runs. Instead, a shift in mass loss rates is observed when $\gamma_0$ and $p$ surpass certain threshold values, similar to that in the CMIP6-forced runs. In the Amundsen, when $p > 0.6$, SLR tends to increase with higher $p$. There also appears to be a threshold in $\gamma_0$ at around $5 \times 10^6$. Below this

value, SLR is modest and does not change much as $\gamma_0$ varies, while above this threshold the ice sheet loses mass quickly as $\gamma_0$ increases. This is particularly evident when the TF anomalies are large enough to overcome the TF correction during the spin-up ($2°C$).

To get a sense of the physical behavior of the ice in the Amundsen during these experiments, we can look at the grounding line retreat over time for a low ($\sim$115 mm) and high ($\sim$1.1 m) mass loss case under $2°C$ TF anomaly (Figure 14). In the low

mass loss case, even with a large TF anomaly, mass loss remains minimal if $p$ and $\gamma_0$ are low. Under high $p$ and $\gamma_0$ values, SLR contributions increase dramatically. In order to achieve a large sea level contribution, the grounding line must be pushed past some key pinning points of high local seafloor topography. Similar behavior near pinning points is noted in the CISM runs in Lipscomb et al. (2021) and in other ice sheet models such as the Ice Sheet System Model (ISSM) (Robel et al., 2019), MPAS-Albany Land Ice (MALI) model (Hoffman et al., 2019), and the adaptive-mesh BISICLES model (Waibel et al., 2018).

Grounding line retreat in their Amundsen experiments (under different melt rate parameterizations) exhibits threshold behavior. In our runs, under sufficient forcing and specific parameter settings (high p and $\gamma_0 > 5 \times 10^6$), the ice is responsive enough that the grounding line can retreat past points of high bed topography, leading to widespread ice sheet collapse that adds another $\sim$1 meter to sea level. In our $2°C$ experiment, seven of the 25 experiments result in Amundsen collapse to varying extents, all contributing more than half a meter to SLR. Of these, all have high values of $p$ and $\gamma_0 > 4.8 \times 10^6$. (An additional $2°C$

experiment with a low p/high $\gamma_0$ combination (as in Fig. A1) did not lead to Amundsen collapse within 500 years.)

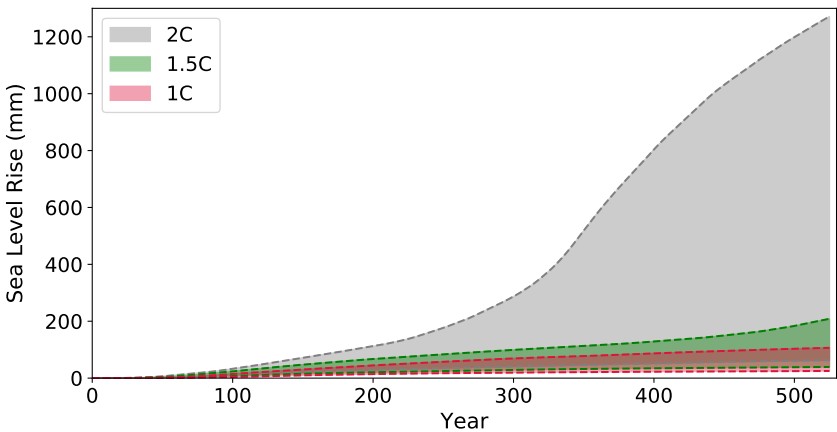

**Figure 13.** Range of sea level rise for all ensemble members, shading color indicates the synthetic thermal forcing experiment.

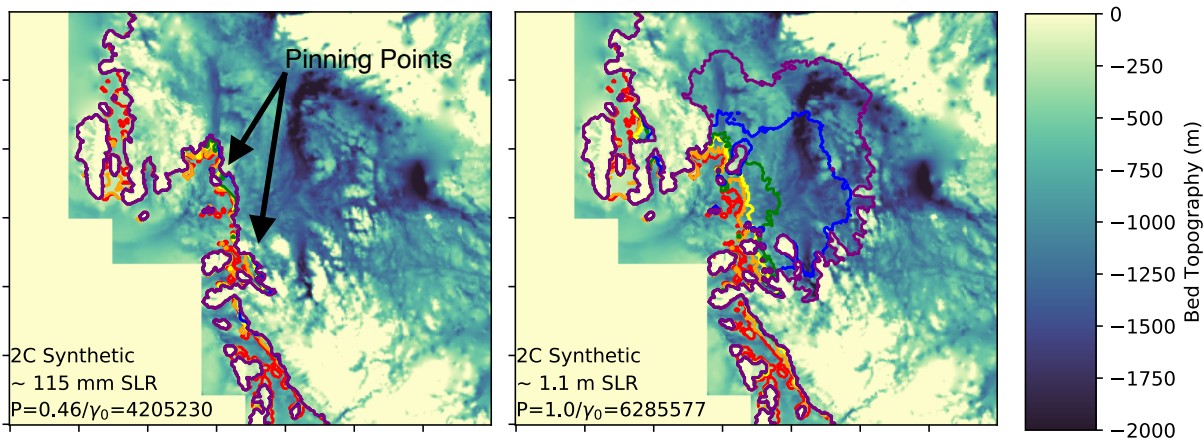

**Figure 14.** Grounding line location evolution over the 2°C synthetic run for (a) $p = 0.46$ / $\gamma_0 = 4205230$ and (b) $p = 1.0$ / $\gamma_0 = 6285577$. Red, orange, yellow, green, blue, purple contours indicate years 0 to 525 at roughly 100 year intervals. Shaded background shows seafloor topography (m). Negative values indicate below sea level. Note that the ice in this area is largely grounded below sea level.

### 3.2.1 Thwaites instability and collapse on longer timescales in the 2°C synthetic framework

Given the large TF correction in the Amundsen region ($\delta T_{Amundsen} \sim$ -2°C), a 2°C warming is only enough to return conditions to present-day observed thermal forcing. Therefore, the 2°C synthetic warming experiments in the ASE can be viewed as committed-SLR experiments under current TF. As such, it is of interest to extend these simulations beyond 500 years to distinguish the stable runs from those with delayed collapse. In this way, we aim to identify the parameter space for Thwaites instability under 'current' TF conditions. To reduce computing time, we chose to extend the runs of a subset of p and $\gamma_0$ values (Table 5).

| Extended Cases | | | | | | | |
|---|---|---|---|---|---|---|---|
| p | 0.15 | 0.54 | 0.6 | 0.7 | 0.74 | 0.96 | 0.98 | 1.0 |
| $\gamma_0$ (m/s) | 2954923 | 5175963 | 1560081 | 5321878 | 8654548 | 4849305 | 1710386 | 6285577 |

**Table 5.** Combinations of p and $\gamma_0$ used in the subset of extended simulations.

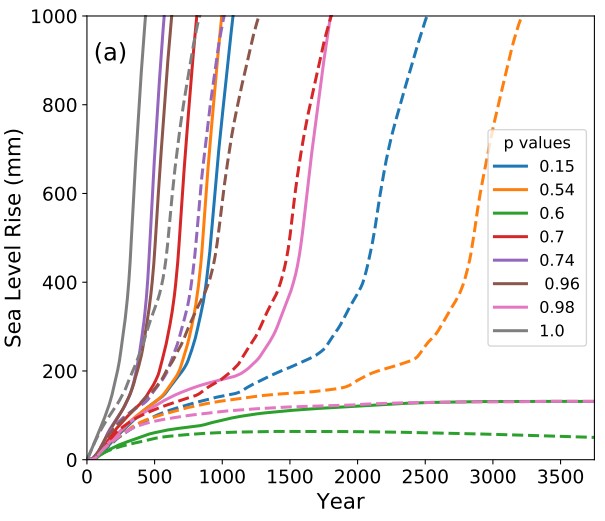 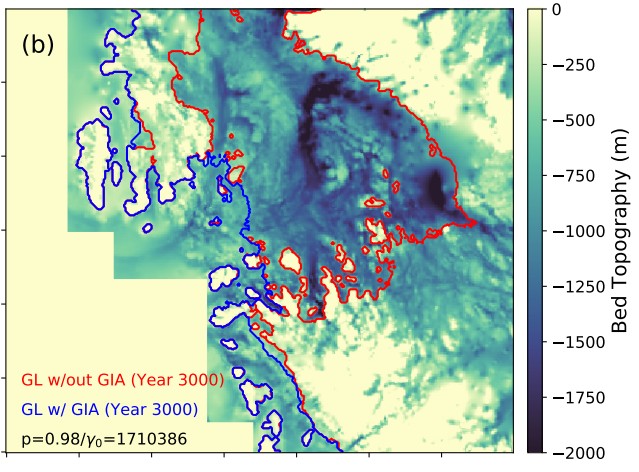

**Figure 15.** (a) Sea level rise for eight extended simulations with (dashed lines) and without (solid lines) a GIA model. The $y$-axis is truncated at 1 m sea level rise to emphasize GIA-related delays of several centuries. (b) Amundsen region grounding line location evolution over the $2°$C synthetic run for $p = 0.98$ / $\gamma_0 = 1710386$ without GIA (red) and with GIA (blue) after 3000 years of simulation time. The shaded background shows seafloor topography (m) without isostatic adjustment. Negative values indicate a bed below sea level.

Thwaites collapse begins within 1500 years in all but one ensemble member. Once the grounding line retreats past some key pinning points (shown in Fig. 14), MISI-type collapse sets in. This builds on the findings in Lipscomb et al. (2021) which, in a more limited set of $p$ and $\gamma_0$ parameter combinations, found Thwaites collapse within 500 years only for $p = 1$. Our extended simulations show that there are, in fact, a wide range of $p$ and $\gamma_0$ values that generate eventual Thwaites collapse. This MISI-type collapse has a characteristic timescale of about two to three centuries, but the period before collapse can be more than 500 years (Fig.15(a)). The pre-collapse period lasts until the grounding line reaches the pinning points associated with SLR of ~130 mm. The case without collapse has a moderate p ($p = 0.6$) and the lowest $\gamma_0$ in our ensemble ($\gamma_0 = 1560081$). We extended this case as far as 9000 years and found that the GL stabilizes on topographic pinning points about 3000 years into the simulation, and remains in this position for the rest of the simulation.

We also ran extended simulations with glacial isostatic adjustment (GIA) enabled. CISM has an elastic lithosphere–relaxing asthenosphere (ELRA) model (Le Meur and Huybrechts, 1996), which represents vertical bed adjustments under a changing ice load. The runs with GIA test whether isostasy can prevent instabilities in some cases, or if it simply delays the process.

With relaxation time scales (100 yr) and lithosphere rigidity ($4\times10^{22}$ N m) characteristic of WAIS (Coulon et al., 2021; Book et al., 2022), we find that GIA nearly always delays, but does not prevent, Thwaites collapse (Fig. 15(a)). The length of delay ranges from 300–800 years and depends on the $p/\gamma_0$ values and the specified SLR threshold. In only one extended scenario ($p = 0.98, \gamma_0 = 1710386$) does GIA prevent Thwaites collapse. Here again, the grounding line stabilizes on high pinning points (Fig. 15(b)), and the TF is too low to drive further grounding line retreat. This case has the second lowest $\gamma_0$ in our ensemble.

### 3.2.2 On the coulomb basal sliding coefficient

This study uses a basal sliding law that allows both power-law and Coulomb behaviors. In our spin-ups, we inverted for the power-law coefficient $C_p$, keeping the Coulomb parameter $C_c$ fixed at 0.5. This value of $C_c$ results in spun-up states that match observations well. However, $C_c$ is not necessarily spatially uniform, and its value can influence the length of the transition zone (the zone where basal sliding transitions from power-law to Coulomb behavior). Leguy (2015) (chapter 7.1.4) showed that $C_c$ has limited impact for small $p$, but influences the length of the transition zone when $p \geq 0.5$. We therefore ran additional spin-ups with low $\gamma_0$ to assess the influence of $C_c$ on ice retreat. With $p = 0.98$ and $\gamma_0 = 1710386$, we tested $C_c = 0.25$ and $C_c = 0.1$. (With $C_c < 0.1$, there is widespread WAIS thinning inconsistent with observations.)

We find that lowering $C_c$ has a limited impact in simulations with CMIP6 climate forcing (Figure A10 ); $\gamma_0$ remains the dominant parameter. In the 2°C synthetic experiment, however, we find that a high p/low $\gamma_0$ combination with $C_c = 0.1$ leads to a similar sea-level response compared to high p/high $\gamma_0$ and $C_c = 0.5$. Thus, lowering $C_c$ makes the Amundsen region more prone to retreat. We recommend further study of the sensitivity to $C_c$ in this region. We also suggest inverting for $C_c$, either instead of or in addition to $C_p$, when using Eq. 3 or similar sliding laws. To this end, work is underway to study new spin-up strategies using the sliding law proposed in Zoet and Iverson (2020), which includes a parameter analagous to $C_c$ but not $C_p$.

## 4 Discussion

Our primary (SSP5-8.5) CMIP6-forced CISM ensemble, consisting of 325 members (13 GCMs × 25 $p$ and $\gamma_0$ combinations), highlights the continuing challenge to constrain uncertainties in Antarctic contributions to sea level, particularly on multi-century timescales. Depending on the magnitude of the TF anomaly and the ice sheet–ocean parameter settings, the final SLR ranges from a minimum of ∼50 mm to a maximum of more than 3 m after 500 years. In all these runs, mass loss is dominated by melt from the Weddell and Ross regions. In some cases, the EAIS makes a moderate sea level contribution, while the AP contributes the least, with no more than 10 mm under any AOGCM forcing. Perhaps surprisingly, most of these simulations do not have large contributions from the Amundsen region.

The strong dependence on $\gamma_0$ in the Ross and Weddell indicates more vulnerability to changing ocean conditions than to basal ice conditions in these regions. Mass loss is roughly proportional to the TF anomaly, although within a certain parameter space ($\gamma_0 < 5 \times 10^6$), mass loss remains modest. Only above this threshold does mass loss become significant. The Ross and Filchner-Ronne (in the Weddell) are both currently cold-cavity shelves (Rignot et al., 2013; Dinniman et al., 2016), and sub-shelf melt rates are limited by weak TF at the grounding line. Once warm water enters these cavities, melt rates increase

drastically. Both shelves have the potential to release vast quantities of grounded ice into the ocean. Other modeling studies have indeed illustrated the potential for the Filchner-Ronne cavity to flip between 'cold' and 'warm' states (Hazel and Stewart, 2020; Hellmer et al., 2017; Naughten et al., 2021), causing an order-of-magnitude increase in sub-shelf melt rates and subsequent

sea level contributions (Siahaan et al., 2021). We find that EAIS mass loss also correlates better with $\gamma_0$ than $p$, particularly when forced by warmer AOGCMs, suggesting more sensitivity to ocean warming than ice parameters. We note that changes in $p$ (a model-internal parameter) are partly compensated by the subsequent calibration of the basal friction parameter, $C_p$. Compensation is also possible for $\gamma_0$ (a forcing-related parameter) via the $\delta T_{sector}$ correction factor. However, $\gamma_0$ directly links ocean temperature changes to mass loss. It is therefore consistent that $\gamma_0$ has a more direct control on ice loss when the

ocean warms.

By contrast, the Amundsen sector sea-level contribution is sensitive to a combination of ice sheet and ocean parameters. Under CMIP6-forced forward runs, the Amundsen response is generally modest, and grounding lines do not retreat significantly. Even under these generally weak AOGCM forcings, the Amundsen exhibits a change in mass loss rates taking it from an unresponsive to a modestly responsive region when $\gamma_0 > 5 \times 10^6$. When $p < 0.6$ and $\gamma_0 < 5 \times 10^6$, regional sea-level contributions

barely exceed 100 mm after 500 years, even for the warmest AOGCM. In this parameter space, varying $p$ and $\gamma_0$ has almost no effect on sea-level contributions. Only above these parameter thresholds is sea-level rise affected by increasing $p$ or $\gamma_0$. For the coldest AOGCM, sea level decreases by the end of the simulations (i.e., there is net ice growth).

Given an individual forcing, the choice of $p$ and $\gamma_0$ has the potential to significantly affect sea-level predictions. At most, for a given future forcing, we find a difference of about 0.2 m after a century, depending on the parameters chosen at spin-up.

While this mass loss is not as drastic as, say, the difference between WAIS stability and WAIS collapse, it would still pose substantial challenges for policy-making and coastal planning. The downstream effects of these parameter choices amplify on multi-century timescales. The final (500-year) SLR prediction varies by up to ~2 m depending on the spin-up parameter choices. Most of this difference arises from mass loss in the Ross and Weddell region (Fig. A2), and $\gamma_0$ is the strongest predictor of such differences on multi-century timescales. That said, the final SLR sensitivity to $p$ and $\gamma_0$ scales similarly across all model

forcings. In other words, no matter the magnitude of ocean forcing, $p$ and $\gamma_0$ alone can generate a 2–3 fold change between the highest and lowest SLR contribution after 500 years. We reiterate that because we did not use a physically meaningful prior for our $p$ and $\gamma_0$ ensemble, these predicted SLR ranges should not be over-interpreted.

The inversion procedure during spin-up gives large negative temperature corrections for the Amundsen sector, and therefore the sensitivity of the Amundsen sector is likely underestimated. Because the CMIP6 forcing is too weak to compensate for

the large negative TF correction in the Amundsen, this forcing generates minimal mass loss compared to the Weddell and Ross. However, the 2°C synthetic simulation overcomes this TF correction, and under the same high $p$ and $\gamma_0$ combinations found in the CMIP6-forced runs, triggers a significant Amundsen collapse within 500 years. We find that partial Thwaites collapse within 500 years (at least an additional 0.5 m of SLR) is possible only when $p > 0.6$, suggesting that partial to full water-pressure support at the grounding line promotes such a collapse. This may be model dependent, as Hoffman et al. (2019)

were able to generate Thwaites collapse with a linear basal friction law and full water-pressure support using a different ice sheet model. Furthermore, $\gamma_0$ must be greater than about $5 \times 10^6$ m/s to trigger a MISI-type instability in these simulations.

Any lower $\gamma_0$ value fails to initiate collapse of any WAIS ice shelf within the modeled 500 years. The synthetic experiments in the Amundsen also illustrate a threshold of instability in the range of 1.5-2°C (with respect to the end of spin-up). This is consistent with the modeling results in Lipscomb et al. (2021) and Rosier et al. (2021), who found similar temperature thresholds for Amundsen-region collapse. This temperature threshold is likely associated with topographic pinning points. Pinning points promote ice-sheet stability by acting as an obstacle to ice shelf flow (Still et al., 2019). Our runs show that in the Amundsen, high seafloor ridges slow ice retreat by allowing the ice to remain grounded for longer. However, under sufficient TF, the ice ungrounds, enabling unfettered retreat.

In several extended 2°C runs, we show that the grounding line can reach critical overdeepenings if given enough time. For all p/$\gamma_0$ cases except one with very low $\gamma_0$, Thwaites collapse is initiated within 1500 years. These runs were done without isostatic feedback, whereas GIA can significantly modify sea-level projections in the Amundsen and other WAIS sectors (Kachuck et al., 2020). Larour et al. (2019) showed that with GIA included, the sea-level contribution from the Amundsen sector was reduced by 20-40% over 250 years. We therefore ran a subset of extended 2°C runs with ELRA isostasy. We found that GIA can delay Thwaites collapse by several centuries, but in most cases collapse remains inevitable. More sophisticated isostasy models would be needed to fully probe GIA impacts on grounding-line stabilization. These extended experiments do not alter the conclusions of the main ensemble of shorter experiments; $\gamma_0$ is more important than $p$ for committed SLR.

We note a number of caveats and assumptions. First, the AOGCM ocean models used to generate our TF generally have low resolution and do not include ice shelf cavities. By assuming that the far-field temperatures can be extrapolated under the shelves, we are missing complex processes and potential feedbacks that shape the sub-shelf cavity circulation and affect melt rates (e.g., time scales of cavity circulation (Snow et al., 2017; Naughten et al., 2019)). For example, once warm water flushes the ice shelf cavities, a positive meltwater feedback can enhance the shelf circulation and the onshore transport of open ocean heat (Hellmer et al., 2017). This would limit our ability to identify such a tipping point without resolving the sub-shelf circulation. Furthermore, the extrapolation of far-field thermal properties into current cold shelf cavities like the Filchner-Ronne and Ross regions may bring an unrealistic amount of heat to grounding lines, overestimating mass loss in these regions (Daae et al., 2020; Naughten et al., 2021).

Without explicitly representing sub-shelf circulation, we have assumed a simple quadratic relationship between TF and melt rates. This melt rate parameterization cannot capture critical processes that transport warm water to grounding lines, such as topographic steering along bed troughs (Nakayama et al., 2018). Due to limited computing resources, we have explored only one such form (non-local slope), and we only consider the SSP5-8.5 forcing scenario (Meinshausen et al., 2020). We neglect other physical processes, such as MICI and atmospheric changes, and our resolution of 4 km is too coarse to capture all grounding-line processes. Also, the simple no-advance calving criterion may underestimate the effects of basal melting on calving-front retreat and buttressing of grounded ice.

Another consideration is the AOGCMs themselves, and the limitations in their representation of high-latitude ocean dynamics. All CMIP6 models used to force these simulations have temperature and salinity biases, particularly in the Southern Ocean (Beadling et al., 2020). The ocean resolution is typically too low to resolve major features such as the Antarctic Slope Current, eddies and tides, ice shelves, and polynyas (Purich and England, 2021; Mack et al., 2019). All these features have the

potential to affect sub-shelf melt rates. For example, Naughten et al. (2018) found that Weddell polynyas have an effect on the Filchner-Ronne cavity temperatures and melt rates since they determine the salinity and density of the cavity source waters. As a result, polynya formation, not resolvable in CMIP6 models, impacts the circulation strength and the melt rates. Finally, since these models are not coupled to the ice sheet, we do not account for meltwater feedbacks.

To overcome many of these issues, it is necessary to better represent sub-shelf circulation and to couple the ocean and ice sheet. While some modeling centers have coupled an interactive Greenland Ice Sheet with an AOGCM, only a few have included ice shelf cavities around Antarctica (e.g., UKESM (Siahaan et al., 2021) and E3SM (Comeau et al., 2022)). CESM developmental code supports a coupled Antarctic ice sheet, but has yet to be validated as of this writing. CESM is also switching to the MOM6 ocean model (Adcroft et al., 2019), which can resolve sub-shelf circulation. It is likely that the ice sheet modeling community will eventually shift away from the constraints of these sub-shelf melt parameterizations.

## 5  Conclusions

In this study, we expand the scope of previous ISMIP6-style simulations by probing in greater detail the dependence of future Antarctic mass loss on two important parameters: $p$ (which affects basal friction near the grounding line) and $\gamma_0$ (which controls the sub-shelf melt rate). By virtue of the spin-up methodology, these parameter settings can condition the ice sheet to be more or less susceptible to ocean thermal forcing, which has significant implications for sea-level projections. We run a 325-member CISM ensemble, where 25 unique combinations of $p$ and $\gamma_0$ are used to generate new spin-ups, each achieving similar spun-up states that are in steady state and whose ice sheet configuration (e.g., ice thickness and velocities) resembles today's ice sheet. These spin-ups do not, however, represent the transient state of the AIS, since the current ice sheet is not in equilibrium (particularly in the Amundsen region). Rather, the ice sheet is spun up to a modern configuration, and these simulations are designed to probe the sensitivity of the AIS around this reference state.

Each spun-up state is run forward, forced with regionally-averaged ocean TF anomalies derived from 13 different CMIP6 models. The thermal anomalies are ramped up linearly for 100 years from the 1995-2005 mean to the 2090-2100 mean, after which they are held constant for 400 years. Our study is novel in that we have identified the parametric thresholds necessary for triggering widespread mass loss within 500 years. We find that with the combination of low basal friction near the grounding line (moderate to high p), high sensitivity of melt rates to TF ($\gamma_0 > 5 \times 10^5$), and sufficient TF anomalies, mass loss becomes significant in multiple basins. This threshold in $\gamma_0$ tends to hold for all major WAIS basins (Amundsen, Ross and Weddell). The choice of $p$ and $\gamma_0$ alone can impact final (500-year) sea level estimates by up to 2 m. The differences are less extreme after 100 years, but still significant, with parameter settings impacting SLR estimates by up to 0.2 m. The Ross and Weddell regions dominate the sea-level contributions in CMIP6-forced forward simulations. Mass loss in these areas is largely controlled by $\gamma_0$ rather than $p$, implying dominance of ocean forcing parameters over ice-sheet parameters. The Amundsen region exhibits a mix of ocean, ice, and temperature thresholds that together determine the sensitivity.

The CMIP6-forced runs fail to produce widespread WAIS collapse after 500 years by virtue of relatively weak forcing in the Amundsen. However, with additional synthetic forcing, we find that large Amundsen mass loss can be triggered with TF

anomalies between 1.5 and 2°C. In these cases, the grounding line retreats from topographic pinning points. Without these stabilizing points, the grounded ice in the basin collapses. Collapse sometimes begins well after year 500, but proceeds quickly once under way, with a characteristic time scale of two or three centuries. Adding GIA feedbacks in extended simulations can delay Amundsen collapse by several centuries, but in most cases does not prevent eventual collapse.

Our study highlights the potential downstream effects of ice conditioning during model spin-up. Since it is possible to achieve a similar spun-up state with different sensitivities to ocean warming, it is imperative to understand the effects of the most influential ice and ocean parameters. Current ice sheet models have difficulty making predictions about Antarctic mass loss, especially on multi-century timescales. More work is necessary to make realistic projections. The sensitivity to model parameters demonstrated in our experiments emphasizes the need to impose better constraints on model initial conditions by using observational constraints for ice sheet transient behavior.

*Code and data availability.* Code and Data are available at: https://github.com/mberdahl-uw/SpinUp_Paper.git

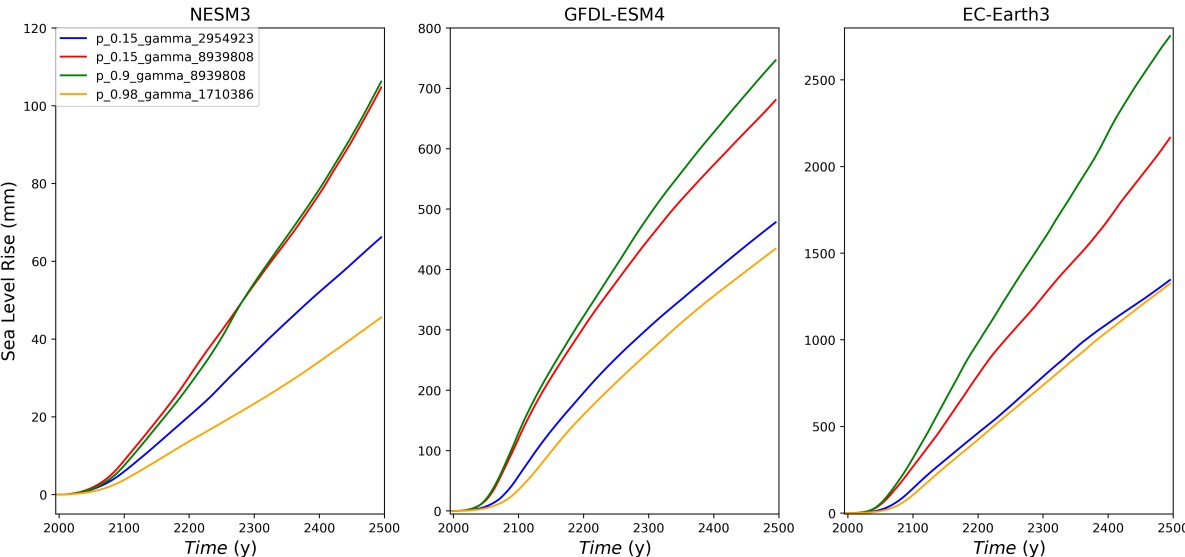

**Figure A1.** Sea level rise experiment using low (NESM3, left panel), moderate (GFDL-ESM4, center panel), and high (EC-Earth3, right panel) CMIP6 climate scenarios, showing results using $p/\gamma_0$ values that are low/low (blue), low/high (red), high/high (green), and high/low (orange). The results show: (1) with low p and high gamma the sea-level response is similar compared to the high p and high $\gamma_0$ combination for low and moderate forcing; (2) p does influence the results under high forcing scenarios; (3) the sea-level response with low p and high $\gamma_0$ is always larger compared to the response with high p and low $\gamma_0$ highlighting the strong influence of $\gamma_0$.

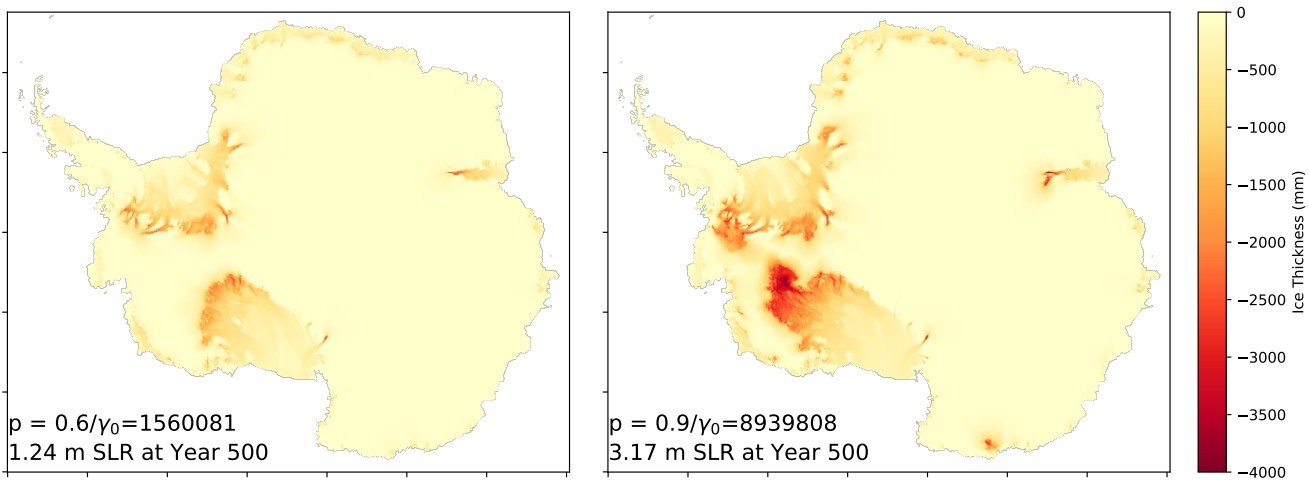

**Figure A2.** Thickness change between beginning and end of simulation for two simulations run with EC-Earth3-Veg. The only difference between these simulations is the p and $\gamma_0$ settings during spin-up. Resulting sea level contributions at year 500 differ by over 2 m. The majority of mass loss occurs in the Weddell and Ross regions.

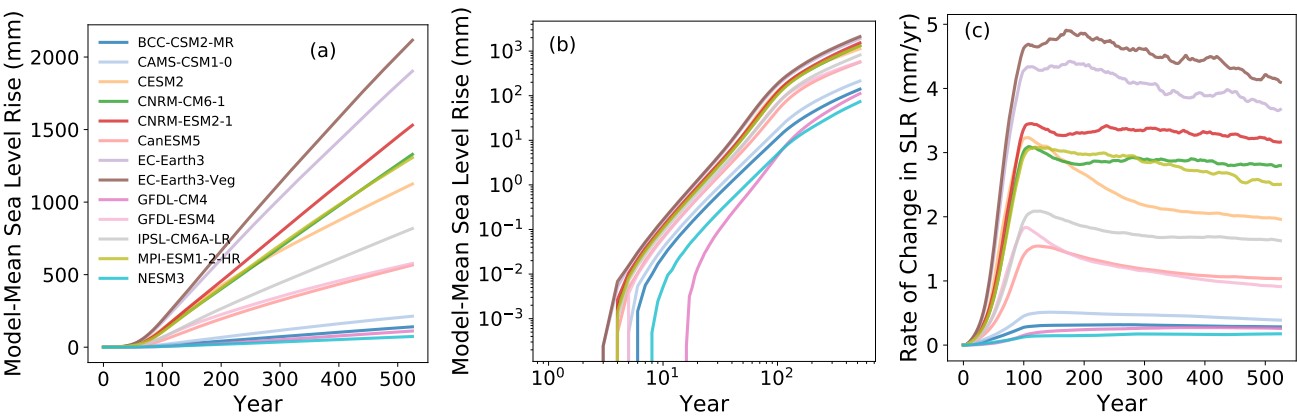

**Figure A3.** (a) Model-mean sea level rise, (b) Model-mean sea level rise on log-log scale and (c) Model-mean SLR rate of change (mm/yr).

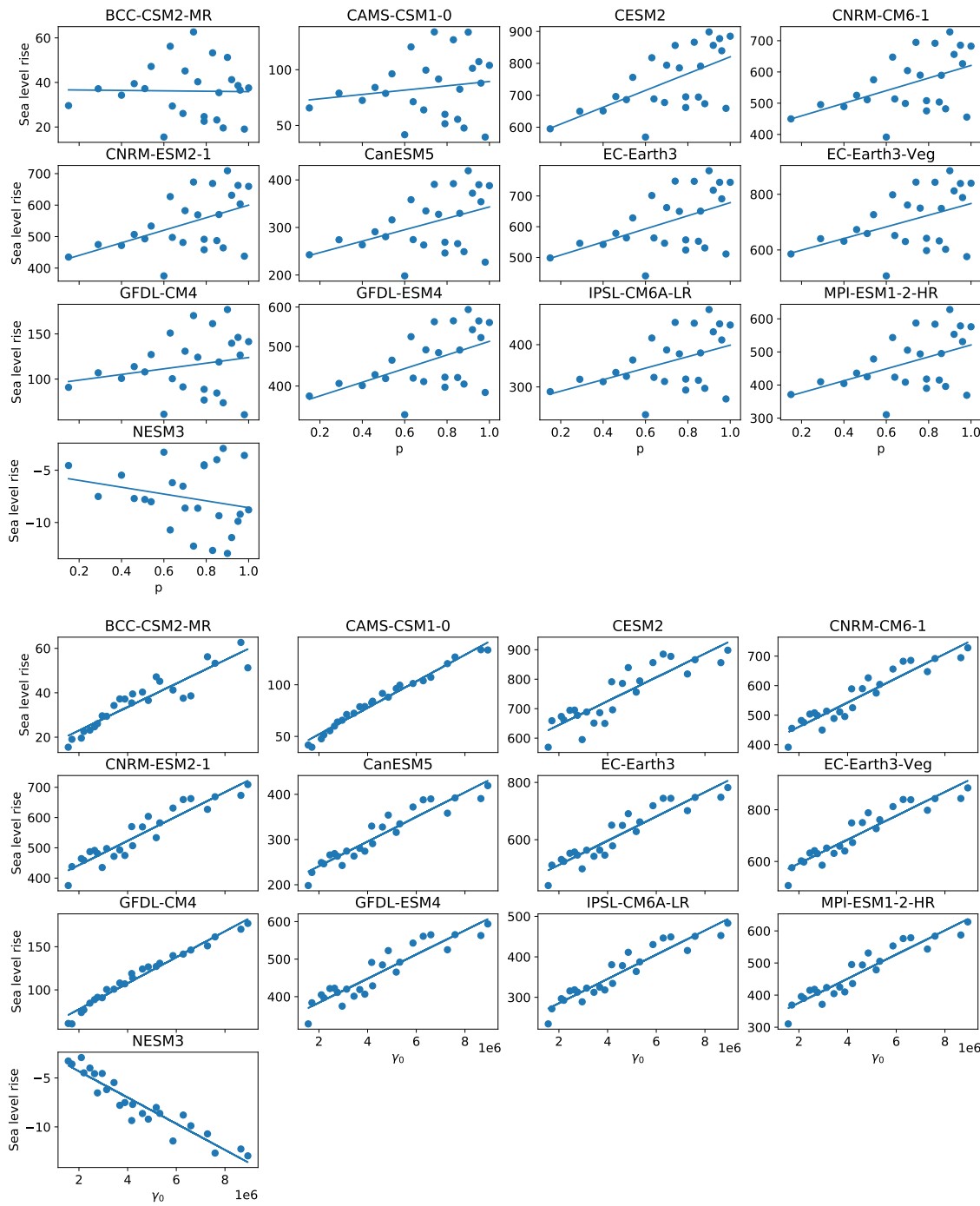

**Figure A4.** SLR as a function of $p$ and $\gamma_0$ in the Weddell region.

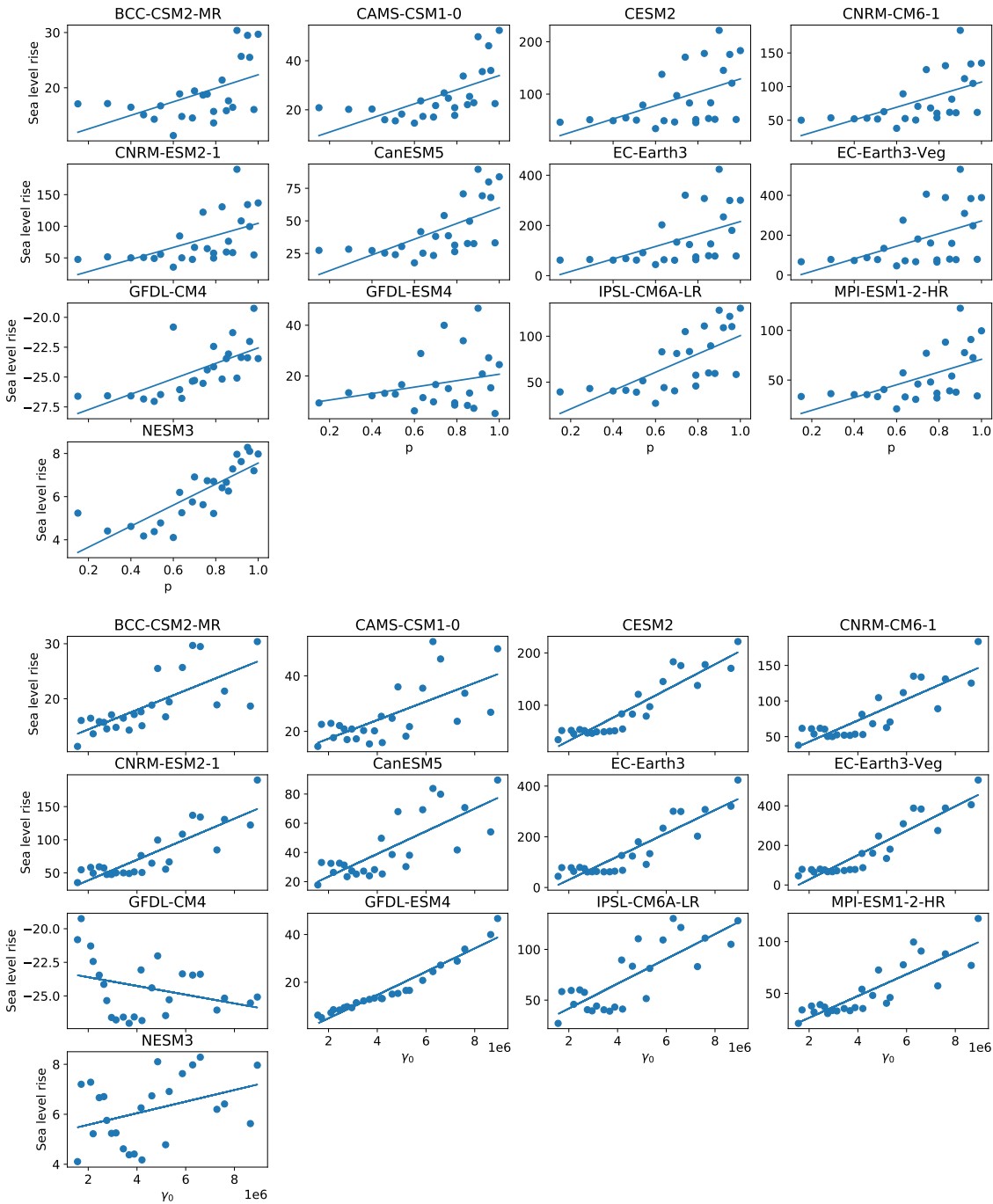

**Figure A5.** SLR as a function of $p$ and $\gamma_0$ in the Amundsen region.

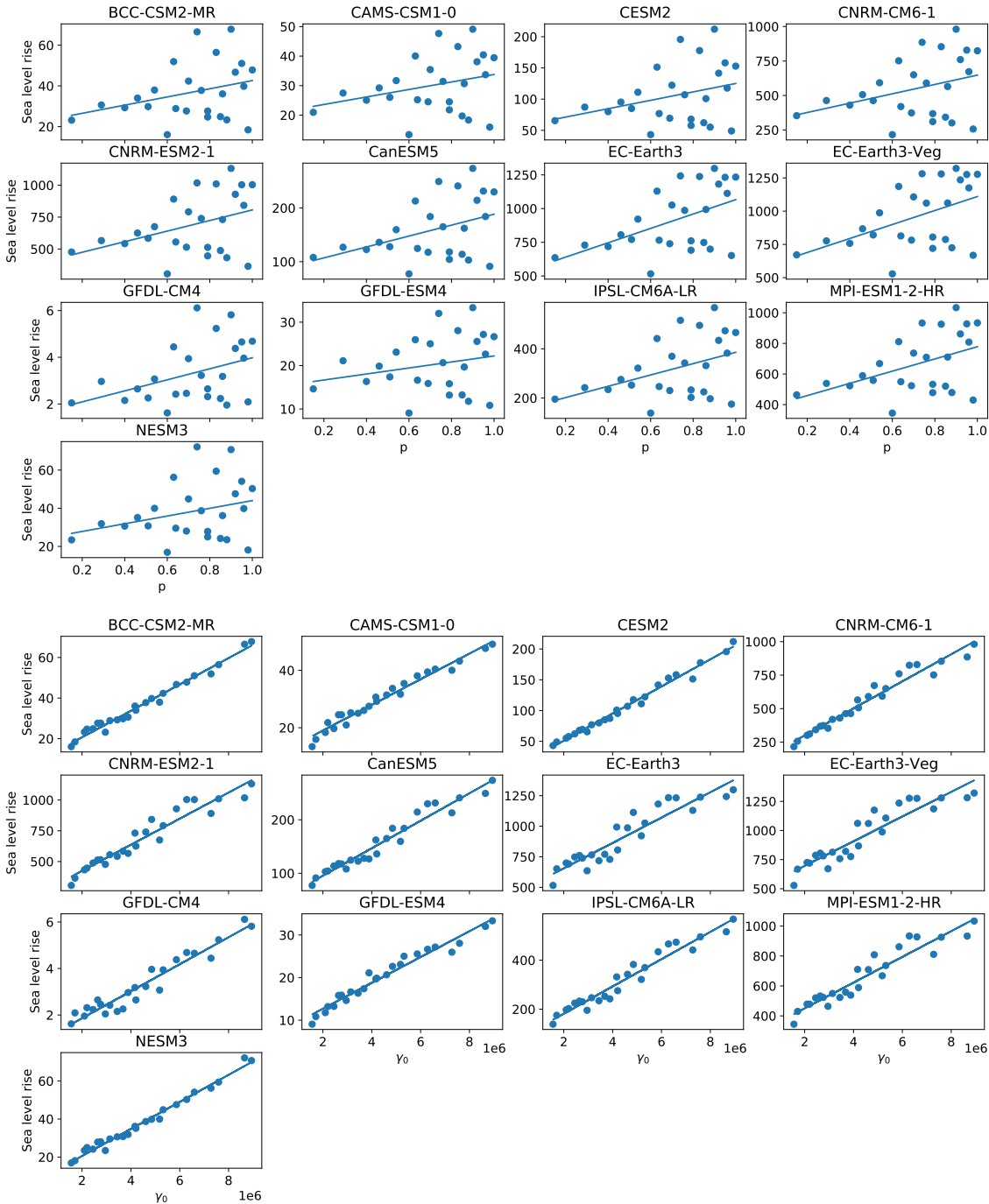

**Figure A6.** SLR as a function of $p$ and $\gamma_0$ in the Ross region.

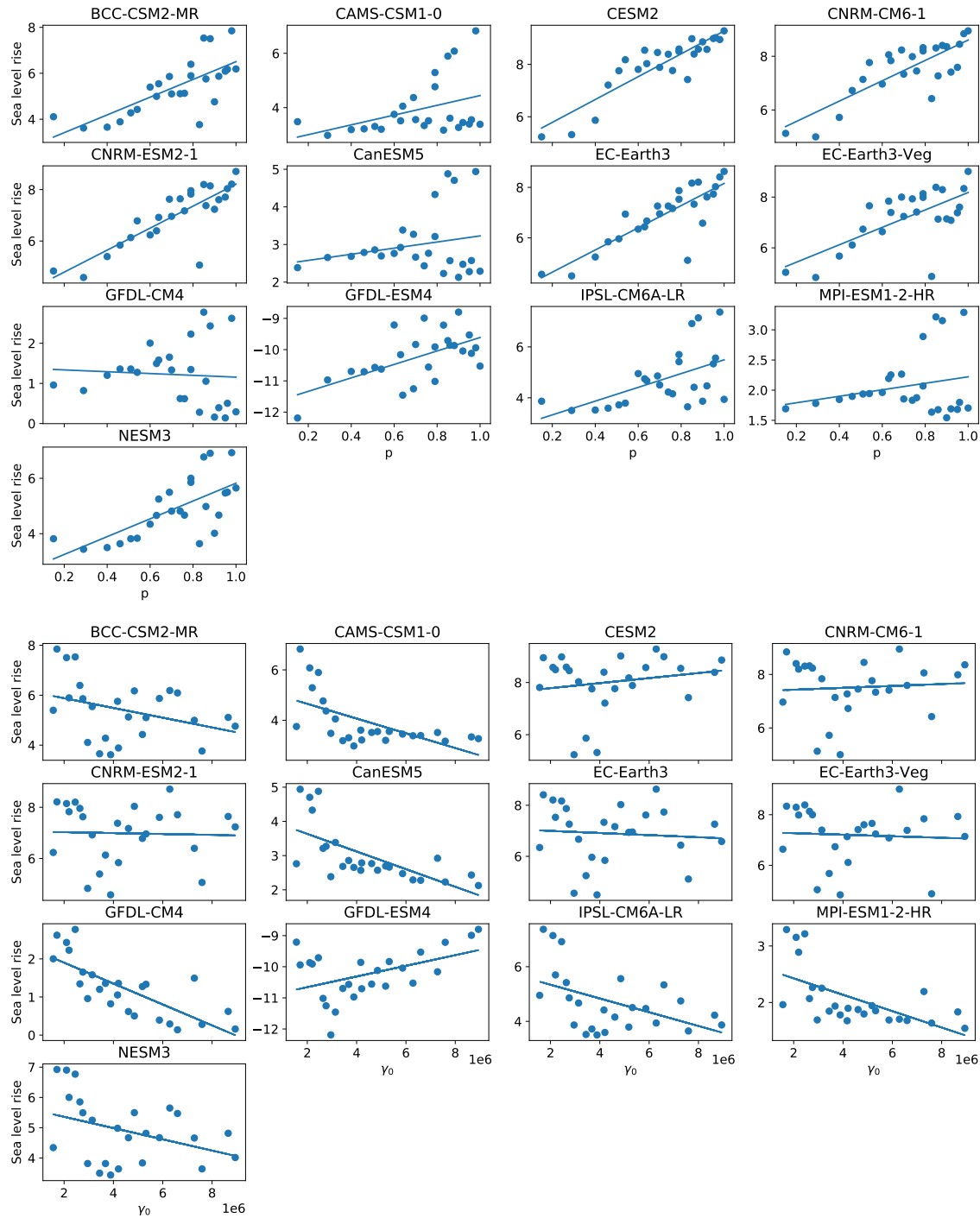

**Figure A7.** SLR as a function o $p$ and $\gamma_0$ in the Peninsula (AP) region.

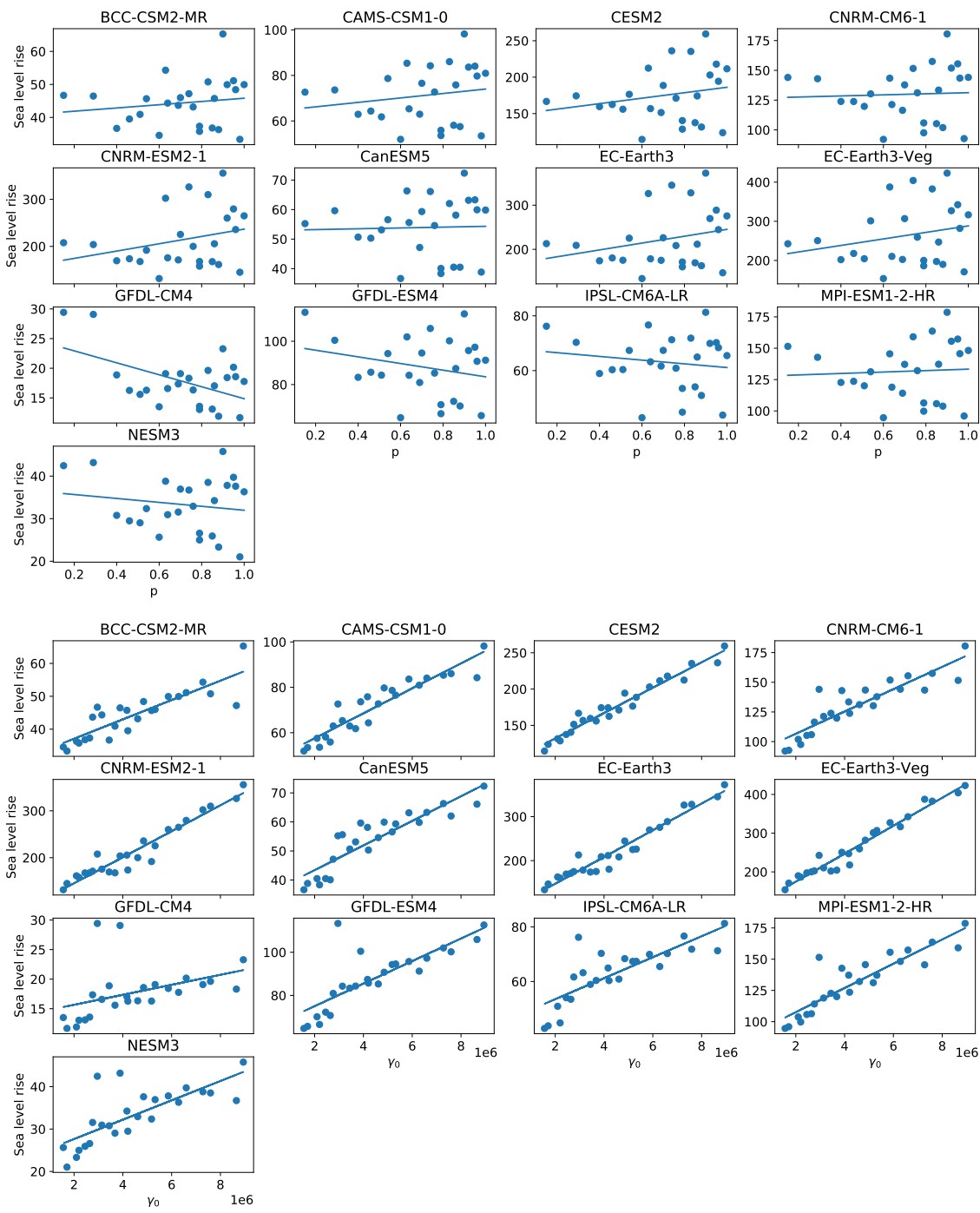

**Figure A8.** SLR as a function of $p$ and $\gamma_0$ in the East Antarctic (EAIS) region.

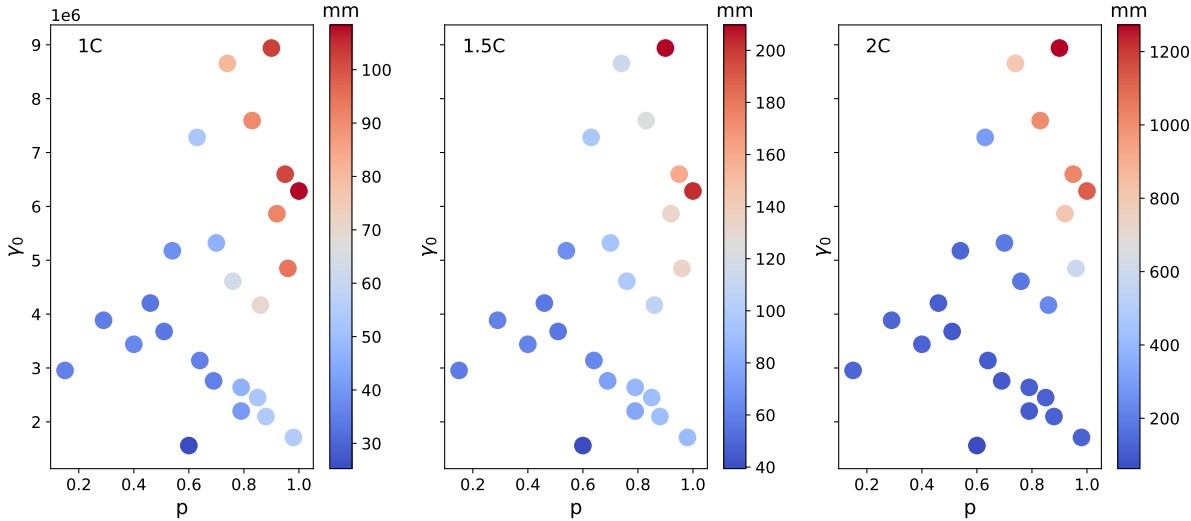

**Figure A9.** Final (500 year) sea level contribution from the Amundsen under three different synthetic forcing scenarios as a function of $\gamma_0$ vs $p$. The panels correspond to the three experiments: $1^{\circ}$C (left), $1.5^{\circ}$C (middle) and $2^{\circ}$C (right). Note the different colorbar scales.

| SLR values after 500 (100) years of simulation | | | | |
|---|---|---|---|---|
| CMIP6 Model Name | Min Final SLR (mm) | Max Final SLR (mm) | Diff Final SLR (mm) | Ratio (Max:Min) Final SLR |
| BCC-CSM2-MR | 82.72 (6.99) | 218.24 (15.94) | 135.52 (8.95) | 2.6 (2.28) |
| CAMS-CSM1-0 | 125.12 (9.91) | 332.82 (25.86) | 207.7 (15.95) | 2.6 (2.6) |
| CESM2 | 768.69 (55.98) | 1599.16 (202.19) | 830.471 (146.21) | 2.08 (3.61) |
| CNRM-CM6-1 | 744.51 (44.33) | 2080.16 (184.63) | 1335.65 (140.3) | 2.79 (4.16) |
| CNRM-ESM2-1 | 853.84 (50.95) | 2392.35 (205.54) | 1538.51 (154.59) | 2.8 (4.03) |
| CanESM5 | 332.52 (19.83) | 855.8 (84.84) | 523.28 (65.01) | 2.57 (4.28) |
| EC-Earth3 | 1139.45 (77.85) | 2881.83 (279.37) | 1742.38 (201.52) | 2.53 (3.59) |
| EC-Earth3-Veg | 1241.82 (85.10) | 3165.48 (300.54) | 1923.66 (215.44) | 2.55 (3.53) |
| GFDL-CM4 | 56.75 (2.27) | 179.89 (7.65) | 123.14 (5.38) | 3.16 (3.37) |
| GFDL-ESM4 | 398.05 (28.90) | 776.32 (113.7) | 378.27 (84.8) | 1.95 (3.93) |
| IPSL-CM6A-LR | 447.69 (27.59) | 1262.62 (118.49) | 814.93 (90.9) | 2.82 (4.29) |
| MPI-ESM1-2-HR | 772.16 (42.85) | 1962.88 (182.37) | 1190.72 (139.52) | 2.54 (4.26) |
| NESM3 | 47.58 (3.26) | 114.27 (6.36) | 66.69 (3.1) | 2.4 (1.95) |

**Table A1.** SLR values after 500 (100) years of simulation for each climate model ensemble showing the minimum, the maximum, the difference, and the ratio between the maximum and minimum. Brackets show values for year 100 years after simulation starts.

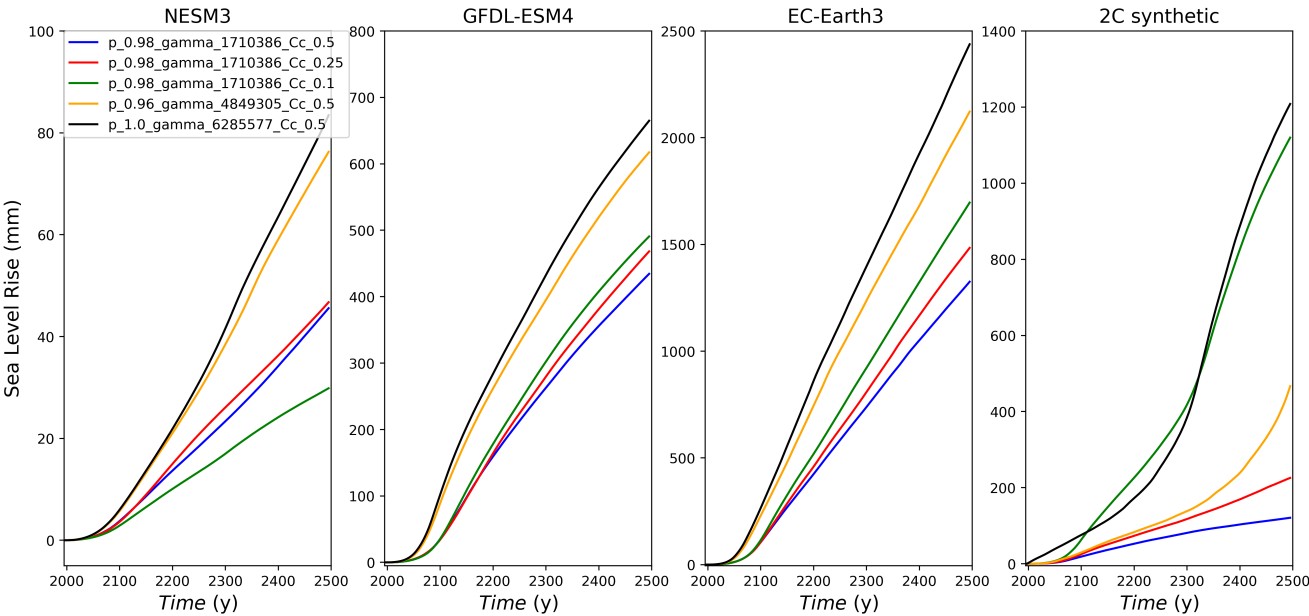

**Figure A10.** Sea level rise experiment using low (NESM3), moderate (GFDL-ESM4), and high (EC-Earth3) CMIP6 climate scenarios and 2°C synthetic experiment, showing results using high p and low $\gamma_0$ values with $C_c = 0.5, 0.25, 0.1$ (blue,red,green respectively) and high p, $C_c = 0.5$, and mid/high $\gamma_0$ values (orange and black respectively). The results show: (1) for fixed p and $\gamma_0$, lower $C_c$ values lead to a stronger sea-level response (the exceptional behavior seen with NESM3 might be due to the low and negative TF forcing in the Amundsen region); (2) For the 3 CMIP6 forced experiments, lowering $C_c$ by a factor of 5 is not enough to match sea level contribution produced by the set ups using higher $\gamma_0$ values; (3) in the 2°C synthetic experiment, the results with $C_c = 0.1$ leads to similar sea-level response compared with the experiment with high $\gamma_0$.

*Author contributions.* MB, GL, and WHL designed the experiments, with input and advice from NU. GL and MB staged and ran the experiments. WHL developed and provided an ice sheet spin-up. MB and GL prepared the manuscript with contributions from all co-authors.

*Competing interests.* The authors declare that they have no conflict of interest.

*Acknowledgements.* We thank Qiang Sun for providing the CMIP6 ocean model output. This work was supported by the U.S. Department of Energy (DOE) Office of Science (Biological and Environmental Research), Early Career Research program, as well as National Science Foundation (NSF) Grant No. 2045075. This material is based upon work supported by the National Center for Atmospheric Research, which is a major facility sponsored by the National Science Foundation under Cooperative Agreement No. 1852977. This research used resources
provided by the Los Alamos National Laboratory Institutional Computing Program, which is supported by the U.S. Department of Energy National Nuclear Security Administration under Contract No. 89233218CNA000001. Further computing and data storage resources, including the Cheyenne supercomputer (doi:10.5065/D6RX99HX), were provided by the Computational and Information Systems Laboratory (CISL) at NCAR.

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
