# Peer review of "Exploring ice sheet model sensitivity to ocean thermal forcing and basal sliding using the Community Ice Sheet Model (CISM)"

_The Cryosphere, 2022_

## Referee Comment (RC3)

**Comments to "Eploring ice sheet model sensitivity to ocean thermal forcing using the Community Ice Sheet Model (CISM)" by Mira Berdahl et al.**

**1    General comments**

In this study, the authors implement the ice-sheet model CISM to explore the mass change of Antarctica ice sheet in 500 years with various combinations of basal sliding and sub-ice-shelf melting parameters.

The study is a supplementary of Lipscomb et al., 2021. In the previous study, the authors explored the uncertainties of Antarctic sea-level contribution caused by grid resolution, forcing scenarios, basal friction law, melt schemes and ocean forcing. In this study, the authors expand the sample range of the basal friction parameter and added the ocean circulation parameter that modulates sub-ice-shelf melting.

My major concern is that this study doesn't add significantly more information to the existed study of Lipscomb et al., 2021. Some significant effort is needed to match the standard of The Cryosphere. I have some suggestions on this:

- Model validation. Model results from simulations present major mass loss from Ross and Weddell sea region, and East Antarctica to a smaller level, while Amundsen sea area contribute the least to sea level. This is very different from present-day observations in the recent decades.

– Historical runs could be conducted before forcing with the future scenarios. The spin-ups are done to have steady-states, while some basins are not at steady-state. And transient runs at historical time could constraint the model (Reese et al., 2020 https://doi.org/10.5194/tc-14-3097-2020).

– The difference between the high limit and low limit of gamma differ by 5 times, which also means 5 times melt rates difference due to gamma. Constraining this parameter by observational melt rate may exclude some samples and reduce the uncertainty of the model results. If that has been done, please describe in the manuscript.

- Model physics: Revisit the parameters related to the Coulomb type friction law. The initial state after spin up give a very different grounding line line positions at some fast streams. I wonder if that's due to the initialization method. At some of the fast streams such as Pine island glacier, Coulomb like friction law applies, in which case $C_p$ plays little role. However, $C_c$ is a fixed value from Asay-Davis et al., 2016, in which experiments and parameters are chosen for ideal geometry. Specifically, $C_c = 0.5$ is chosen to have a continuous basal traction, not necessarily apply to real glaciers. This may also the reason of low sensitivity to basal friction, simply because the nudging parameter doesn't impact the basal traction in the fast flow areas. I suggest more effort on the validation of the parameters related to the basal traction, in order to get the grounding line locations closer to observation.

**2  Specific comments**

- The second part of the introduction should be in method.

- L21: The Antarctic ice sheet...sea level (GMSL). Need to add the time scale.

- L27: 'deep uncertainty' → 'poorly known processes'

- L29: MICI is not physics. Hydrofracture and cliff failure are the processes that induce MICI.

- L47: Put acronym (AIS) the first time 'Antarctic Ice Sheet' is used.

- L69: 'friction' → 'basal friction'

- L62-L76: The motivation of this study is not clearly described. Lipscomb et al., (2021) explored the two parameters in their previous study using the same model and same spin up. This study is a good supplementary to the previous work, but this work does not add much info to the existed work.

- L77-L85: How did you select the CMIP6 models?

- section 1.1: This section is more suitable for method section? Furthermore, this section could be simiplified because some of the ISMIP6 ocean parameterization methods such as PIGL is not used in this study. equation (2): how did you define and calculate the slope? Does it change with time when the geometry changes?

- L117-L123: Please justify the ocean forcing parameterization method. The authors mentioned that the method in this study results in melt rates that differ appreciably from observational estimates. Can you present 1: the deltaTs needed to have reasonable spin-ups, and what's the difference between spin-ups; 2: melt rates and its variation in between the spin-ups.

- L135: $\rho_{sw}$ instead of $\rho_s w$?

- L145: Can you explain the limitations of Lipscomb et al., 2021 that are improved in this study?

- L53: Reference of ABUMIP should be Sun et al., 2020.

- Can you present the model drift of both velocity field and ice geometry? Furthermore, can you present the variation of velocity, ice thickness and grounding line locations among the spin-ups? The initial state of simulations such as different ice geometries could result in essential differences in dynamical responses of the ice sheets.

- L169: delete the sentence 'There is no hydrology in the basal friction field.' because water pressure used in this study is simplified basal hydrology.

- L174: Label the Anumdsen sector and the Kamb Ice Stream in the map.

- How did you decide the range of the parameter gamma0? It seems (from Figure 3) there is no experiment with both high gamma0 and low p values, what's the advantage of the non-uniform sampling methods?

- L198-L203, Figure 4:

  - Dash line missing in Figure 4d?
  - Why grounded ice area and grounded ice mass are different from observation from the beginning?
  - As mentioned before, can you show the spatial variation of the spin-ups?

- L222: 'monotonically increase' → 'monotonically change'? According to Figure 6, negative anomalies exist.

- Quite a few of climate models present non-linear behaviour in the future thermal forcing change. Why not using the original model results?

- L232: What scenarios are the TF anomalies (1, 1.5 and 2 degrees) represent?

- L266: Figure 7 seems to suggest the differences caused by the parameters increase over time (instead of 'less pronounced after 100 years')?

- L279: 'faster-than-linear' → 'non-linear'

- L296-L297: Why smaller p results in higher effective pressure? Seems to be conflict with equation (4).

- L299-L301: This is contradict with observation. I think some model validation work should be done to explore parameter ranges or processes.

- L351-353: The effect of high topographic seafloor points is not clear to me from Fig. A9. The labelled pinning points are close/higher than sea level. Grounding lines form around them because these mountains are never ice free thanks to surface mass balance.

- L409-L410: I think more effort is worth made to improve on constraining the parameters sample range.

- L420: The hypothesis of MISI (or any instability) could only be proved at steady states, which is not the case of this study. Fast retreat doesn't mean MISI.

- L426-427: Again, grounding line hovering around these areas doesn't mean they are stabilizing upstream flow.

---

## Author Response (AR1)

Dear Mira Berdahl and co-authors,

First of all I want to thank you for addressing most of the raised review comments. I very much appreciate that you ran additional projections accounting for GIA effects. Clearly, a full coupling to a basal hydrology model is out of scope ;o) In my view, the central caveat, primarily raised by reviewer #3 is the steady-state assumption using present-day climate. All reviewers agree that this key assumption has to be clearly communicated and discussed. In response to reviewer #2, you already provide more details on the mass loss partitioning after spin-up.

In addition to the actions that you already undertook, I would like to ask you to address my initial comments on 'parameter transferability', 'sampling strategies' and consequences to moderate some conclusions/formulations as well as on the possibility for 'inherent compensations' during the spin-up. Finally, reviewer #3 raised a very specific comment on the sliding law. Therefore, I suggest that your revised article will enter a second review round.

Please see below for our responses regarding 1) parameter transferability, 2) sampling strategies, and 3) inherent compensations. Our response to Reviewer 3's comment on the sliding law has been updated in the Response to Reviewers document, and also copied at the end of this document (most recent changes in purple text).

On this basis, I invite you to submit a revised manuscript. Please consider the pending items that I identified above.

The editor, Johannes Fürst

################################################################################

Justification (visible to authors and reviewers only):
Dear Mira Berdahl and co-authors,

Thank you for submitting to TC/TCD. You are certainly aware that papers accepted for TCD will appear immediately online and are open for comments and review. Before entering the proper review process, submissions undergo a rapid access review by the editor to ensure initial quality standards. This quality control is not meant as a full scientific review but to ensure accordance to the journal remit as well as compliance with the general review criteria on originality, scientific quality and significance. The criteria for this evaluation can be found at: https://www.the-cryosphere.net/peer_review/review_criteria.html. Grades are from 1-4 (excellent - poor).

-- -- -- -- -- -- -- -- -- -- -- -- -- -- -- -- -- -- -- -- -- -- -- -- -- -- -- -- -- -- -- -- -- -- -- -- -- -- -- -- -- -- -- -- -- -- -- -- -- -- -- --
-- -- -- -- -- -- -- -- -- --

**1. ORIGINALITY (Novelty): 2**

-- -- -- -- -- -- -- -- -- -- -- -- -- -- -- -- -- -- -- -- -- -- -- -- -- -- -- -- -- -- -- -- -- -- -- -- -- -- -- -- -- -- -- -- -- -- -- -- -- -- -- --
-- -- -- -- -- -- -- -- -- --

The authors present a model sensitivity study for future sea-level contribution from the Antarctic Ice Sheet under oceanic forcing. They focus on two key parameters in the ice-ocean coupling. The first parameter (gamma_0) links the ocean temperature forcing to sub-shelf melting. The second parameter (p) controls the influence of the basal water or effective pressure on basal friction near the groundling line (GL). As these two parameters are largely unconstrained, they use a stratified Latin-Hyper-Cube (LHC) sampling with 25 members and subsequently produce initial states, by calibrating the basal friction coefficient and a temperature offset to correct ice-shelf melt rates. Starting from each of these 25 initial states, future projections are conducted forced by single-scenario ocean temperature anomalies from 13 Atmosphere-Ocean General Circulation Models (AOGCM). For this purpose, the authors rely on a state-of-the-art ice-sheet model that took part in recent intercomparison projects. The authors conclude that the future sea-level contribution (SLC) strongly depends on the choice of the two key parameters. Relative SLC differences by 2500 exceed a factor 2. Low GL friction and high TF sensitivity precondition widespread multi-meter sea-level rise from Antarctica. Ice loss is primarily funnelled through the Weddell and Ross sectors. The study is complemented by synthetic warming experiments for the Amundsen Sea Sector to increase the ocean temperature forcing beyond the AOGCM anomaly range. In this way, important ice loss is triggered in this area.

The study represents a continuation and extension of the CMIP6 intercomparison with a single ice-sheet model. The asset is the bi-variate character of the sensitivity study. I therefore consider it a valuable contribution in terms of novelty.

-- -- -- -- -- -- -- -- -- -- -- -- -- -- -- -- -- -- -- -- -- -- -- -- -- -- -- -- -- -- -- -- -- -- -- -- -- -- -- -- -- -- -- -- -- -- -- -- -- -- -- --
-- -- -- -- -- -- -- -- -- --

**2. SCIENTIFIC QUALITY (Rigour): 2**

-- -- -- -- -- -- -- -- -- -- -- -- -- -- -- -- -- -- -- -- -- -- -- -- -- -- -- -- -- -- -- -- -- -- -- -- -- -- -- -- -- -- -- -- -- -- -- -- -- -- -- --
-- -- -- -- -- -- -- -- -- --

The objectives of this work are very well articulated and the methodology is explained in adequate depth. Nonetheless, I want to raise a few more general comments at this stage.

-- -- -- -- PARAMETER SELECTION -- -- -- --
In the introduction, you briefly introduce the two key parameters on which you want to focus, namely p and sigma_0. While the latter is linked to the ocean forcing strategy, the latter is model specific. I therefore wondered how transferable this parameter is to other ice-sheet

models apart from CISM. From my understanding, many other models intrinsically assume a fixed value. This is important, so please highlight this transferability. This clarifies why you specifically selected this model-internal parameter. For the second forcing-related parameter, the transferability is obvious. You might already want to highlight the type difference of the two parameters: model internal and forcing related.

We agree with these ideas of transferability, and model-internal vs forcing-related parameters.  Both of these notions are useful to highlight for the reader.  Therefore, we have added new text to the Introduction section, when discussing p and gamma: "
We note that while $\gamma_0$ is forcing-related and therefore transferable across ice sheet models, p is a model-internal parameter and might not be directly transferable. Our formulation with p applies to sliding laws in which basal friction depends on the effective pressure N. Other models with similar sliding laws can therefore benefit from this study."

-- -- -- -- PARAMETER SAMPLING -- -- -- --
Considering your stratified LHC sampling, no combination is chosen in the parameter space where p<0.6 and gamma_0>5e5 (cf. Fig. 3). In my view, the deliberate under-sampling or non-coverage implies that some of your conclusions are not well supported. I want to give some examples.
* * *
RESULTS (L297): There you state p<0.6 precludes high SLR impacts from Antarctica. However, you did not sample p<0.6 and gamma_0> 0.5e6.
RESULTS (L314): 'For p < 0.6, mass loss tends to track p linearly'. This is only based on 6 samples of parameter combinations.
DISCUSSION (L383pp): 'Mass loss is roughly proportional to the TF anomaly, although within a certain parameter space (p < 0:6 and gamma_0 < 5e6), mass loss remains modest. Only above these thresholds in p and
gamma_0 does mass loss ever become significant.'
DISCUSSION (L415pp): 'We find that partial Thwaites collapse within 500 years (at least an additional 0.5m of SLR) is possible only when p > 0.6, […].' In this p-range, you did not sample gamma_0 values larger than 5e6. So your statement only holds for smaller values of gamma_0.
CONCLUSION (L472pp): 'These thresholds in p and gamma_0 tend to hold for all major WAIS basins (i.e. Amundsen, Ross and Weddell).'
* * *
Finally, I want to express my appreciation that you already use concise and considered formulations to avoid misunderstandings in many places. Yet in these few locations and possible some others, I think that you have to moderate your statements. Alternatively, you could extend your parameter sampling.

It is true that our study (as originally designed) does not allow us to assess combinations of low p and high gamma. We chose not to sample this space because previous work indicated that for $0 < p < 0.5$, ocean forcing was the dominant influence compared to p. To further demonstrate this, we have run additional experiments with low p (p=0.15) and high gamma (gamma=8939808) for high (EC-EARth3), moderate (GFDL-ESM4), and low (NESM3) climate forcing scenarios. The results show that simulations with low p and high gamma have similar or lower sea level response compared to high p and high gamma simulation. Thus, gamma is the primary parameter impacting the model response. The parameter p has a significant impact only for high-forcing scenarios (e.g., EC-Earth3).

We added a figure in the Appendix (now Fig A1) to highlight these results with the following caption: "Sea level rise experiment using low (NESM3, left panel), moderate (GFDL-ESM4, center panel), and high (EC-Earth3, right panel) CMIP6 climate scenarios, showing results using $p/\gamma_0$ values that are low/low (blue), low/high (red), high/high (green), and high/low (orange). The results show: (1) with low p and high $\gamma_0$ the sea-level response is similar compared to the high p and high $\gamma_0$ combination for low and moderate forcing; (2) p influences the results under high-forcing scenarios; (3) the sea-level response with low p and high $\gamma_0$ is always larger than the response with high p and low $\gamma_0$ highlighting the strong influence of $\gamma_0$."

In addition, we have reorganized the manuscript and removed references to the p=0.6 threshold.

-- -- -- -- DISCUSSION: INHERENT COMPENSATION -- -- -- --
As the sensitivity to the key parameter p is small, it could be worth to discuss the fact that changes in this parameter are partially compensated by the subsequent calibration of the basal friction parameter. Such an inherent compensation is also possible for gamma_0 via the offset values deltaT. Yet gamma_0 linearly links the oceanic temperature changes to mass loss. It is therefore consistent that gamma_0 has a more direct control on future ice loss when temperature increase. I therefore think that you should discuss this fact by distinguishing the two parameters as model-internal and forcing-related.

This is a good point, and we agree that there are inherent compensations between p and the basal friction parameter calibration, or between gamma and deltaT. We have added comments on this in the Discussion section: "We note that changes in p (a model-internal parameter) are partly compensated by the subsequent calibration of the basal friction parameter, Cp. Compensation is also possible for $\gamma_0$ (a forcing-related parameter) via the $\delta T\_sector$ correction factor. However, $\gamma_0$ directly links ocean temperature changes to mass loss. It is therefore consistent that $\gamma_0$ has a more direct control on ice loss when the ocean warms."

-- -- -- -- -- -- -- -- -- -- -- -- -- -- -- -- -- -- -- -- -- -- -- -- -- -- -- -- -- -- -- -- -- -- -- -- -- -- -- -- -- -- -- -- -- -- -- -- -- -- -- -- --
-- -- -- -- -- -- -- -- -- --
3. SIGNIFICANCE (Impact): 2

-- -- -- -- -- -- -- -- -- -- -- -- -- -- -- -- -- -- -- -- -- -- -- -- -- -- -- -- -- -- -- -- -- -- -- -- -- -- -- -- -- -- -- -- -- -- -- -- -- -- -- -- -- -- -- --

You certainly highlight an important issue/uncertainty on projecting future sea-level contribution from the Antarctic Ice Sheet. Yet I miss clear advices to the cryospheric modelling community about best practices on parameter choices. As the manuscript stands now, you leave it rather vague what to do about these poorly constrained and interdependent key parameters.

-- -- -- -- ADVICES -- -- -- --

Can you give a constructive strategy for parameter selection. Is there any means how the model ensemble can further be assessed. Can you provide some sort of skill scores to rank the 25 parameter combinations. The skills could include root-mean-square errors/deviations between observed and modelled geometric and dynamic variables, the temperature offset deltaT or the ice-shelf thinning rates. You certainly have some procedure at hand for such a quantification.  This idea of scoring/ranking the spin-ups was addressed in the first response to reviewers (Reviewer 3).

-- -- -- -- -- -- -- -- -- -- -- -- -- -- -- -- -- -- -- -- -- -- -- -- -- -- -- -- -- -- -- -- -- -- -- -- -- -- -- -- -- -- -- -- -- -- -- -- -- -- -- -- -- -- -- --

4. PRESENTATION QUALITY: 1

-- -- -- -- -- -- -- -- -- -- -- -- -- -- -- -- -- -- -- -- -- -- -- -- -- -- -- -- -- -- -- -- -- -- -- -- -- -- -- -- -- -- -- -- -- -- -- -- -- -- -- -- -- -- -- --

The paper is well written and the structure is easy to follow. Findings are well supported by useful figures of mostly good quality. An example is the stratified LHC sampling which is nicely summarised in Figure 3 showing the underlying distribution functions. Well done. I only want to suggest to transfer some figures and tables to the appendix or an additional supplement (see specific comments below).

-- -- -- -- -- -- -- -- -- -- -- -- -- -- -- -- -- -- -- -- -- -- -- -- -- -- -- -- -- -- -- -- -- -- -- -- -- -- -- -- -- -- -- -- -- -- -- -- -- -- -- -- -- -- -- --

Finally, please consider that the identified points are certainly not exhaustive. Yet they might well be indicative for issues that will potentially be picked up by reviewers. Please consider addressing them at this stage. In general, I consider the manuscript suitable for entering the review process as a TCD article. You will soon be contacted for initial typesetting.

The editor, Johannes Fürst

###############################################################################################################

SPECIFIC COMMENTS:

L299: typo: an —> a Corrected

L402: On which basis due you speak about the maximum difference This sentence was a bit vague, it has been updated for clarity.

L445: typo: latitude  Corrected

Figures & Tables

Fig. 1 - there are only two panels. The velocity differences are not shown. Please also indicate the ensemble member (p, sigma_0) for this comparison. The caption has been updated to address these comments.

Fig. 2 - This figure can tentatively be joined with Figure 1 as both show comparisons of the calibration stage.  Since Figure 1's left panel is just observations, while Figure 2's left panel is Spin-up minus observations, we think it would be clearer for the reader to keep these figures separate (if they were stacked, the left column subpanels would not be consistent).

Fig. 3 - Same temperature range for all panels facilitates comparability.

We found that the following three figures were referred to frequently enough in the text that they were better to keep in the main section of the paper.

Fig. 4 - Consider transferring it to a Supplement.

Fig. 5 - Consider transferring it to a Supplement.

Fig. 6 - Candidate for Supplement.

Fig.13 - In this figure, I find it confusing that each ensemble member is plotted twice. I prefer a presentation as in Fig. 8 which shows the same information.  This figure has been removed from the paper since it was only cited once and seemed to cause more confusion than necessary.

Fig. A6 - Here the Antarctic Peninsula SLR contribution is shown dependent on the two parameters. In this plot there is an almost perfect linear dependence on the TF scaling (gamma_0). Yet correlation coefficients are small as shown in Fig. 11. I think there is a confusion of results per region (I suspect Fig. A7).  This has been resolved, thanks. It was a matter of the wrong figure filename in the Latex manuscript file.

Table 3 - Consider transferring it into a Supplementary Material because of redundancy with Fig. 3.  Table 3 has been edited during the revisions, and now contains new information, no longer redundant.

Table 5 - Consider transferring it into a Supplementary Material because of redundancy with Fig. 7. Also consider adding the numbers for the end of the century in parenthesis. Optionally transfer some of the relevant numbers (range min/max in relative or absolute units) into Fig.7  We have moved this table to the Supplementary Material, thanks.  The values for Year 100 have also been included in the table now.

################################################################################
Sliding Law comment from Reviewer 3 and our responses.  Most recent updates are noted in purple.
################################################################################

- Model physics: Revisit the parameters related to the Coulomb type friction law. The initial state after spin up give a very different grounding line line positions at some fast streams. I wonder if that's due to the initialization method. At some of the fast streams such as Pine island glacier, Coulomb like friction law applies, in which case Cp plays little role. However, Cc is a fixed value from Asay-Davis et al., 2016, in which experiments and parameters are chosen for ideal geometry. Specifically, Cc = 0.5 is chosen to have a continuous basal traction, not necessarily apply to real glaciers. This may also the reason of low sensitivity to basal friction, simply because the nudging parameter doesn't impact the basal traction in the fast flow areas. I suggest more effort on the validation of the parameters related to the basal traction, in order to get the grounding line locations closer to observation.

We are not sure what specific figure or text the reviewer is referencing here, since we do not present or discuss the grounding line position at the end of spin-up in our manuscript.

In any case, we are spinning up to match the observed thickness.  In most cases, if we match the thickness, the GL is close to the observed location. It is most important to have a good inversion method – though there are always trade-offs between having an accurate GL location and overfitting the system. In the Pine Island region, the model does have trouble finding a stable position at its current location, where it is retreating.  In the spin-up, it often ends up too advanced and too thick.  To highlight the differences in the spin-up ensemble, we added a figure (now Figure 5 to the manuscript) showing the ensemble mean spin-up state.

The reviewer raises a good point regarding the sensitivity of the spun-up state to the Coulomb parameter Cc. It is true that lower Cc would lead to faster sliding in locations with lower effective pressure. Cc matters more in the Coulomb regime in which Cp loses its influence (due to the basal sliding law asymptotic behavior). To test the importance of Cc, we ran some additional tests with lower, spatially uniform Cc values of 0.25 and 0.1. (For Cc < 0.1, it is not possible to obtain a spun-up state consistent with observations; most of WAIS is too thin.) We have added a figure to the appendix showing results with these new Cc runs, using low (NESM3), mid (GFDL-ESM4), high (EC-Earth3), and 2C synthetic experiment.

We now address this explicitly in a new Section (3.2.2) (including a new figure in the Appendix):

"This study uses a basal sliding law that allows both power-law and Coulomb behaviors. In our spin-ups, we inverted for the power-law coefficient Cp, keeping the Coulomb parameter Cc fixed at 0.5. This value of Cc results in spun-up states that match observations well. However, Cc is not necessarily spatially uniform, and its value can influence the length of the transition zone (the zone where basal sliding transitions from power-law to Coulomb behavior). Leguy (2015)

(chapter 7.1.4) showed that Cc has limited impact for small p, but influences the length of the transition zone when p>=0.5. We therefore ran additional spin-ups with low $\gamma_0$ to assess the influence of Cc on ice retreat. With p=0.98 and $\gamma_0$=1710386, we tested Cc=0.25 and Cc=0.1. (With Cc<0.1, there is widespread WAIS thinning inconsistent with observations.)

We find that lowering Cc has a limited impact in simulations with CMIP6 climate forcing (Figure A10); $\gamma_0$ remains the dominant parameter. In the 2C synthetic experiment, however, we find that a high p/low $\gamma_0$ combination with Cc=0.1 leads to a similar sea-level response compared to high p/high $\gamma_0$ and Cc=0.5. Thus, lowering Cc makes the Amundsen region more prone to retreat. We recommend further study of the sensitivity to Cc in this region. We also suggest inverting for Cc, either instead of or in addition to Cp, when using Eq. 3 or similar sliding laws. To this end, work is underway to study new spin-up strategies using the sliding law proposed in Zoet and Iverson (2020)}, which includes a parameter analagous to Cc but not Cp."

More recently, we have been testing a Zoet-Iverson sliding law where we invert for Cc instead of Cp, which does not appear in that formulation.  We have also implemented an inversion scheme in which we invert for deltaT_ocn in every grid cell, instead of just per basin.  With these changes, we can match the PIG GL location and thickness more closely. Preliminary tests with these updates show that PIG sensitivity to ocean warming is not much changed in the new runs where the GL location is more accurate. Thwaites sensitivity, on the other hand, is increased. This suggests that the relative insensitivity of PIG is largely a function of the bed geometry rather than the advanced GL in the spin-up.  Please also see more discussion about end-of-spinup GL positions relative to observed in the specific comments below.

These updates have not yet been submitted for publication and were not available at the time of the experiments described in this paper. (There are plans to report on them in the future.)  Once these updates have been implemented and tested more comprehensively, it would be interesting to re-run this suite of simulations.  Until then, we believe the ensemble presented in this paper is still relevant and of interest to the community.

Leguy, G.: The effect of a basal-friction parameterization on grounding-line dynamics in ice-sheet models, New Mexico Institute of Mining and Technology, 2015.

Zoet, L. K. and Iverson, N. R.: A slip law for glaciers on deformable beds, Science, 368, 76–78, 2020.

**Reviewer #1**

**General comments**

This paper presents an ensemble of simulations of ice flow in Antarctica over a scale of several centuries. The ensemble members are obtained by varying two parameters, one controlling the sensitivity of sub-ice shelf melt to ocean thermal forcing, the other relating to basal friction near the glacier grounding line. Overall I find it to be a good exercise in determining the sensitivity of the types of models currently in common use to unknown parameters and I recommend publication with minor revisions. I have a few suggestions but again I think the paper achieves the goal the authors had in mind and makes a valuable contribution.

We thank the reviewer for taking the time to review this study. Their comments have helped us clarify the main points of the paper, and the caveats it comes with.

My biggest concern is with the low mass loss from the Amundsen sector that arose as a consequence of the thermal forcing correction. The separate experiment just for the Amundsen sector without the thermal forcing correction felt a little like an ad hoc way to make up for the deficiencies of the spin-up process. It doesn't detract from the conclusions of the paper as an exercise in probing the sensitivity of the system to its parameters *around a particular reference state*, with the understanding that this reference state is not identical to the modern. Stating this shortcoming more explicitly in the conclusion would help readers who aren't modeling experts from interpreting more than what these interesting results actually say.

We agree. The Amundsen-only synthetic simulations were a way to surmount the large TF correction factor that is borne out of the goal to achieve a steady-state at the end of spin-up. It is worth reiterating this detail in the conclusions. We added new text in the Conclusion (line 467):

"These spin-ups do not, however, represent the transient state of the AIS, since the current ice sheet is not in equilibrium (particularly in the Amundsen region). Rather, the ice sheet is spun up to a modern configuration, and these simulations are designed to probe the sensitivity of the AIS around this reference state."

Finally, the code repository on github consists of the inputs and outputs as a bunch of NetCDF files, but it could be helpful to have some more code and scripts to aid in reproducing the workflow itself.

The GitHub repository has been updated, and now consists of not only the output data (.nc) files from the CISM ensemble, but also includes a script to generate Figures 7,8,9 and Appendix 2. It also includes a JupyterLab script that shows how the gamma and p Latin

Hypercube sampling (on a deformed grid) works, and how to make Figure 3.   The repository is located here: https://github.com/mberdahl-uw/SpinUp_Paper

**Specific comments**

132: The formulation for the effective pressure in equation 4 with p close to 1 is used often in the literature. It assumes that the ocean is the primary determiner of subglacial hydrology, and one of its implications is that there's no basal water when the glacier bed is above sea level. Now it would be crazy talk for me to suggest that you to run all this with a subglacial hydrology model too, but we know that those assumptions are incorrect -- there's appreciable basal water far upstream of the Siple Coast ice streams that has nothing to do with seawater infiltration.

Yes, we agree. An explicit subglacial hydrology scheme would obviously account for the well-established presence of basal water that is not sourced from the ocean.  It is also true that including such a scheme at this point is beyond the scope of this work, despite remaining a critical next step in ice sheet model development. We added text that states that p *only influences basal sliding for ice located below sea level*, and does not replace a hydrology model. This parameterization is another way to make the bed softer and  promote faster sliding when the ice is below sea level.

We now make an explicit statement acknowledging this (line 138):

"This parameterization only accounts for basal sliding for ice located below sea level. It does not account for subglacial hydrology in regions where the glacier bed is above sea level. A hydrology model for CISM is currently in development."

155: The fact that CISM achieved similar results using a 4km resolution as other ice flow models in these intercomparison experiments doesn't necessarily guarantee that this is an adequate resolution for this particular experiment. Can you provide some other assurances that this resolution is adequate? Standard practice in numerical PDE would be to run the simulation with, say, degree-1 and degree-2 finite elements and checking where the two disagree. It's well-known that grounding line evolution is sensitive to resolution near the grounding line, see e.g. Goldberg et al (2009), Grounding line movement and ice shelf buttressing in marine ice sheets.

This is a fair point that, we agree, should be explicitly addressed in the text. It should be noted that, for continental-scale simulations, ice sheet models are typically run at resolutions of 4 km or coarser (although some models can do grid refinement near grounding lines).

The issue of grounding line sensitivity to grid resolution has been explored in depth with CISM (Leguy et al., 2021, and Lipscomb et al., 2021). Lipscomb et al. (2021) found only moderate sensitivity to grid resolution in multi-century, ocean-forced, AIS experiments when comparing 2km and 4km simulations. This sensitivity was less than the sensitivity to sub-shelf melting and basal friction parameterizations.

Furthermore, Leguy et al. (2021) concluded that on century timescales, grid resolutions of 2-4 km may be sufficient (when using CISM), giving an error in grounding line location of only a few kilometers compared to a simulation at 1 km or higher resolution. They also conclude that the quality of the initial state - in particular the agreement of the initial grounding line location with observations - may be more critical than grid resolution for simulating the transient response.

In the text, we added the following in Section 2.1, line 155, to address this point:

"For continental-scale simulations, ice sheet models are typically run at resolutions of 4 km or coarser (Seroussi et al., 2019). On century timescales, Lipscomb et al. (2021) found CISM was only moderately sensitive to grid resolution in ocean-forced AIS experiments, concluding that 4 km resolution was comparable to 2 km resolution. Leguy et al. (2021) found that CISM grid resolutions of 2-4 km may be sufficient to represent grounding line migration.Therefore, all continental-scale, Antarctic simulations were run on a uniform 4 km grid…"

**Technical corrections**

64: This is annoyingly nitpicky but it's a pet peeve of mine when people say "modulate" unless they're referring to frequency or amplitude modulation of a periodic signal. The word I'd use here is "control" or "determine" because the γ parameter directly controls the melt rate.

Fixed here, and in one other location, thank you.

115: "non-local and non-local" I'm guessing this should be "local and non-local"?

Fixed, thanks.

Figure 4: It looks like the grounded ice mass is sampled coarser in time near the start of the simulation than the total ice mass; is this a mistake?

This is not a mistake. 1-dimensional (1D) and 2-dimensional (2D) variables are written to different output files with frequencies of 1 year and 25 years respectively. Unfortunately, grounded mass is not part of CISM 1D output and was derived from the 2D output. We added the following note to the caption to clarify this:

"Note that the frequency of model output is sparser in panel (b) because it was derived using variables with less frequent output."

**Reviewer #2**

Berdahl *et al.* presents an ensemble of ice sheet simulations, exploring centennial-scale impacts of model initialisation, focusing on two parameters that impact ice sheet sensitivity to climate forcing: a scaling factor for the basal ice shelf melt rate ($Y_0$) and the effective pressure near the grounding line ($p$). They then run forward experiments with climate forcing from 13 CMIP6 models under a high emissions SSP585 scenario. They run additional experiments focused on the Amundsen Sea Embayment (ASE), which show ocean thermal forcing anomalies above 1.5°C increase the likelihood of ASE mass loss. This is well-written paper with a comprehensive analysis that should be of interest for *The Cryosphere*, but I have a few comments that I hope the authors can address.

We thank the reviewer for their thoughtful and thorough review.

General comments:

Spin-up procedure:  In the 10,000 year spin-up runs, what type of climate forcing is used (e.g. paleoclimate, modern)? Figures 1 and 2 show good agreement with modern ice thickness and velocity observations, but it is not clear if the models produce reasonable mass loss under a "historical" climate, and details of the control run are lacking. How do SMB, BMB and calving rates compare to observations? The authors acknowledge that a steady-state assumption might not be valid and perform additional sensitivity experiments targeting mass loss in the ASE, but I also wish this was explained more clearly in Section 2.2.

The spin-up is forced toward modern conditions.  The text in the Methods section states: "The ice sheet is initialized to the present-day thickness using the BedMachineAntarctica data set (Morlighem et al., 2020). The surface mass balance (SMB) is from a late 20th century simulation with the RACMO2.3 regional climate model (van Wessem et al., 2018). SMB is held constant using the RACMO2 1976-2016 climatology in the spin-up and forward runs."

In the original  manuscript, we did not specify the thermal forcing dataset used during the spin up. We are correcting this with the following text in section 2.1, line 170:  "climatological data set spanning 1995-2018 from Jourdain et al. (2020)."

By design, we are trying to achieve a steady-state of the ice sheet that matches observed thicknesses and extent. We are not trying to achieve a snapshot of the current transient evolution of the ice sheet. This is why we do not show comparison with observed SMB, BMB and calving rates. Instead, we focus on generating a 'steady state' AIS. In this sense we can better evaluate the impacts of p and gamma on future mass loss, since the mass loss we observe will almost exclusively be the result of changing p and gamma values and not of the transient model behavior.

However, to address the reviewer's comments, we have added text in the Methods section that relates typical spin-up SMB, BMB and calving fluxes to observations. The new text reads:

"As noted earlier, the RACMO2 historical SMB climatology is used, with spin-up SMB ~2500 Gt/yr compared to observed~2300 Gt/yr (Mottram et al., 2021; Rignot et al., 2019). Observational estimates of basal mass balance (BMB) are ~1300 Gt/yr (Rignot et al., 2013; Depoorter et al., 2013), while typical spin-up values are about 630 Gt/yr. This discrepancy in observed and modeled BMB is in large part due to the large $\delta T\_Amundsen$ values, discussed in greater detail below. Spun-up calving fluxes are around 2000 Gt/yr, while observed values are roughly 1300 Gt/yr (Depoorter et al., 2013). Since the spun-up BMB is reduced from present-day values as a result of ocean cooling, the calving fluxes must increase to make up the difference, which results in spun-up calving fluxes larger than observed."

The control runs are used to account for potential drift after the spin-up procedure. For each set of p and gamma parameters, we run a control of the same length as the forced experiments, and both SMB and TF remain unchanged.

We have added a new paragraph at the end of Section 2.2, line 205, which clarifies that while a steady-state assumption is made, this leads to some issues particularly in the Amundsen (which is certainly not in steady state currently). The text reads:

"The assumption of an ice sheet at equilibrium is unrealistic, especially for the Amundsen sector. The large negative $\delta T$ values in Table 3 reflect this assumption. They indicate that in order to match the ice sheet's current configuration during spin-up, a large negative thermal correction was necessary to cool the ocean to prevent retreat. To overcome the artificially cooled ocean temperatures in the Amundsen, we also run a set of synthetic experiments targeting only the Amundsen region (further details in Sec. 2.4). More discussion on the TF correction factors in Table 3, specifically what they imply with respect to our assumption about a `current' state, can be found in Section 3."

Fixed calving front: It is my understanding that the calving front is held fixed in its current location for the forward simulations. This is unrealistic and I'm wondering to what extent it impacts the study findings. For example, a retreat of the calving front would reduce ice shelf

area / buttressing, and could thereby reduce sensitivity to $Y_0$, and increase sensitivity to *p*. One option to address this would be to use the approach of ISMIP6 and include some ice shelf collapse experiments.

The term 'no-advance calving' can be misleading. The forward runs use a no-advance calving scheme that holds the calving front at its present-day location *unless* the shelf melts from above or below. This means that the calving front is indeed allowed to retreat and an ice shelf front could even potentially re-advance. The term no-advance means that it simply cannot advance *beyond* its current location. It is true though that increased basal melt would likely cause the calving front to retreat by thinning the ice and increasing basal crevassing. In that sense, we may be underestimating the effects of basal melting on buttressing and grounded ice flow.

To clarify, we added the following in Sec. 2.1, line 163:

"A no-advance calving criterion that holds the calving front near its observed location. During forward runs, the calving front is allowed to change location as the ice melts, and it can re-advance, but cannot advance past its original observed location."

In the discussion we have added this sentence to clarify this point: "Also, the no-advance calving criterion may underestimate the effects of basal melting on calving-front retreat and buttressing of grounded ice."

Glacio-isostatic adjustment: The authors do not explicitly state if the model includes glacio-isostatic adjustment. Larour et al. (2019) show that the elastic response of the underlying lithosphere is important in ice sheet projections of the ASE, reducing the sea level contribution by 20-40% over 250 years. Given the study's focus on this region and that the simulations run for 500 years, the authors should consider this effect. Incorporating the solid earth response in the models would likely decrease the overall spread by limiting grounding line retreat and dynamic mass loss of the high melt/low friction simulations, and also increase the TF anomaly needed to trigger mass loss. At the very least, the authors should discuss this as a limitation of the study.

It is true that glacial isostatic adjustment (GIA) plays a role in sea level projections and has the potential to slow down ice retreat, as stated in Larour et al. (2019). Currently, CISM only includes an elastic lithosphere–relaxing asthenosphere (ELRA) model. This model assumes that the lithosphere rigidity and asthenosphere relaxation time are uniform everywhere, which is not the case in Antarctica (Whitehouse et al. (2019)). The relaxation time in the East Antarctic Ice Sheet (EAIS) is longer compared to the West Antarctic Ice Sheet (WAIS), and in particular in the ASE (Kachuck et al., 2020), and the contribution to sea level from EAIS is small compared to that of WAIS.

For this reason, and to address the reviewer's comment, we ran additional experiments using the synthetic 2°C experimental framework (which focuses on the ASE). Since the

relaxation time is uncertain within 1 to 2 orders of magnitude, we used a relaxation time scale of 100 years (Book et al., 2022) and a lithosphere rigidity of $4 \times 10^{22}$ Pa/m$^2$ (Coulon et al., 2021).  In these runs, Thwaites collapse is generally delayed by 300–800 years depending on the SLR threshold and p/gamma values being considered. We found one case in which collapse is altogether avoided if GIA is included, but this is relevant only on multi-millennial time scales.  We have added a new section (Section 3.2.1) that presents results from these GIA runs and other extended simulations without GIA included.

Specific comments:

Line 70: "size of the region" is not clear to me.

Changed from "More precisely it informs the size of the region where friction is influenced by hydrological connections with the ocean."

To The second parameter, p,, affects the effective pressure near the grounding line, and is specific to how CISM handles basal friction. P represents the proportion of marine based ice supported by sea water pressure. It essentially dictates the degree of basal slipperiness, particularly in marine-based ice."

Line 74: Instead of "baked in sensitivities", maybe "committed responses" is more appropriate.

We believe that 'committed responses' might invoke other implications about sea level projections that we want to be careful to avoid. Therefore, we have attempted to clarify what we mean by baked-in sensitivities here. The text now reads:

"A large range of p and ɣ0 combinations can yield acceptable spun-up states that have different sensitivities to future ocean warming. In other words, the choices of p and gamma during the spin-up process impact the resultant basal friction field and sub-shelf conditions, which in-turn affect the ice sheet's sensitivity to ocean thermal forcing."

Fig 1: Could include the relative error

We added the RMSE and corrected the Figure caption. It now reads:

"Observed (left panel) (Rignot et al., 2011) and modeled (right panel) Antarctic surface speed (m/yr, log scale) at end of spin-up. The root mean square error between observed and modeled velocity is 128.7 m/yr. White patches represent missing data."

Similarly, we added the root mean square error to Figure 2. The caption now reads:

"Difference between modeled and observed ice thickness (left) and modeled ice thickness (right) with root mean square error 51.8 m."

Fig 8: There is a clear outlier with the control forcing (i.e. low $Y_0$, mid-range $p$). Can the authors explain this?

We believe this panel was misleading with its original colorbar scheme, and led the reviewer to believe there was an outlier when in fact there was not. The colorbar has now been updated in the figure to clarify this for any future readers. Furthermore, we double checked the data point to be sure it was not unusual. We can confirm that the value was for the p = 0.6, gamma = 1560081 combination, and produces a final sea level change of about -0.15 mm. This is the lowest p-value used in the ensemble, and thus led to a sea level fall, but it is still within the range of the other control run mass loss values. As such, we do not consider it an outlier.

Line 337: See above general comment on spin-up procedure. I think this should be included in the methodology section 2.2.

Yes, thank you. As noted in an earlier comment, we have now added a paragraph at the end of Section 2.2 (Spin-up ensemble Design), which acknowledges the assumption of steady-state, and why this is likely not an accurate assumption particularly for the Amundsen sector. The new text reads:

"The assumption of an ice sheet at equilibrium is unrealistic, especially for the Amundsen sector. The large negative values in Table 3 are a reflection of this assumption. They indicate that in order to match the ice sheet's current configuration during spin-up, a large negative thermal correction was necessary to cool the ocean to prevent retreat. To overcome the artificially cooled ocean temperatures in the Amundsen, we also run a set of synthetic experiments targeting only the Amundsen region (further details in 2.4). More discussion on the TF correction factors in Table 3, specifically what they imply with respect to our assumption about a `current' state, can be found in Section 3."

Line 417: With the caveat that solid earth feedbacks would slow Thwaites retreat / collapse.

As discussed earlier, the manuscript lacks a discussion on solid Earth impact. We have added a new section (Section 3.2.1) with extended simulations (and plot) that test Thwaites collapse thresholds with and without isostatic adjustment included.

Fig A9: This is a useful figure. I suggest adding this to the main text.

Good suggestion, this is done.

**Reviewer #3**

Comments to "Exploring ice sheet model sensitivity to ocean thermal forcing using the Community Ice Sheet Model (CISM)" by Mira Berdahl et al.

**1 General comments**

We thank the reviewer for their detailed review of our manuscript. We believe that our edits based on these comments have improved the manuscript. We hope our responses explain some of the key issues raised. Most importantly, we have run some extensions to our 2°C synthetic simulations, including some simulations where we include glacio-isostatic rebound. These runs are described in a new section of the paper. We have also tried to better explain the reasoning behind our spin-up and parameter sampling methodologies.

In this study, the authors implement the ice-sheet model CISM to explore the mass change of Antarctica ice sheet in 500 years with various combinations of basal sliding and sub-ice-shelf melting parameters.

The study is a supplementary of Lipscomb et al., 2021. In the previous study, the authors explored the uncertainties of Antarctic sea-level contribution caused by grid resolution, forcing scenarios, basal friction law, melt schemes and ocean forcing. In this study, the authors expand the sample range of the basal friction parameter and added the ocean circulation parameter that modulates sub-ice-shelf melting.

My major concern is that this study doesn't add significantly more information to the existed study of Lipscomb et al., 2021. Some significant effort is needed to match the standard of The Cryosphere. I have some suggestions on this:

- Model validation. Model results from simulations present major mass loss from Ross and Weddell sea region, and East Antarctica to a smaller level, while Amundsen sea area contribute the least to sea level. This is very different from present-day observations in the recent decades.

    - Historical runs could be conducted before forcing with the future scenarios. The spin-ups are done to have steady-states, while some basins are not at steady-state. And transient runs at historical time could constraint the model (Reese et al., 2020 https://doi.org/10.5194/tc14-3097-2020).

Yes, we recognize that spin-up methodologies come with caveats regarding the assumption of steady-state. As we noted in the response to Reviewer 2, it is by design that we are trying to achieve a steady-state of the ice sheet that matches current observed thicknesses and configuration. Importantly, we are not trying to achieve a snapshot of the current transient evolution of the ice sheet. This is a sensitivity study, rather than a predictive SLR assessment. As such, we focus on generating a 'steady state' AIS, so that we can better evaluate the impacts of p and gamma on future mass loss, since the mass loss we observe

will almost exclusively be the result of changing p and gamma values and not of the transient model behavior.

As noted in the response to Reviewer 2, we have now added a paragraph at the end of Section 2.2 (Methods: Spin-up ensemble Design), which reiterates the assumption of steady-state, and why this prompts care in the interpretation of our results, particularly for the Amundsen sector.  The new text reads:

"The assumption of an ice sheet at equilibrium is unrealistic, especially for the Amundsen sector. The large negative values in Table 3 are a reflection of this assumption. They indicate that in order to match the ice sheet's current configuration during spin-up, a large negative thermal correction was necessary to cool the ocean to prevent retreat. To overcome the artificially cooled ocean temperatures in the Amundsen, we also run a set of synthetic experiments targeting only the Amundsen region (further details in 2.4). More discussion on the TF correction factors in Table 3, specifically what they imply with respect to our assumption about a `current' state, can be found in Section 3."

- ○ The difference between the high limit and low limit of gamma differ by 5 times, which also means 5 times melt rates difference due to gamma. Constraining this parameter by observational melt rate may exclude some samples and reduce the uncertainty of the model results. If that has been done, please describe in the manuscript.

Gamma values are already empirically-derived. They were taken from the ISMIP6 protocol, which are obtained by calibrating observed melt rates, both averaged for the whole Antarctic (MeanAnt) and for the highest melt rates near Pine Island (PIGL).  This methodology is explained in detail by Jourdain et al. (2020).  In the text, we write:  "Jourdain et al. (2020) generated a distribution of possible $\gamma_0$ values in order to reproduce either the observed present-day Antarctic melt rates (averaged over a sector), MeanAnt calibration, or the (much higher) PIGL calibration melt rates."

The range of the min and max gamma spanned here is actually almost an order of magnitude [$1.47 \times 10^6$ to $1.0 \times 10^7$].  (The ISMIP6 protocol allowed for an even wider range of gamma values than we used in this study.  However, we truncated the highest values, since Nicolas Jourdain suggested to us that they were unrealistically high.) This is explained in more detail in Section 2.2 of the paper.

As for our sampling, we intentionally chose this large range in order to sample the full range of sensitivities being explored by the community.  As described in the paper, we skewed the sampling toward lower gammas to preferentially sample more within the MeanAnt range as this was seen to be more 'reasonable' based on the Lipscomb et al. (2021) results at the time.

The text states: "We develop a distribution of gamma that spans both the MeanAnt and PIGL ranges. … We chose [a distribution] such that values would fall preferentially within the MeanAnt range rather than the high end of the PIGL range (Figure \ref{fig:joint_sampling}, x-axis). Note that the upper value is truncated to be 10^7 instead of ~ 3x 10^7 as experimentation suggests that the latter value is far too high (N. Jourdain, personal communication, Nov 12, 2020)."

- Model physics: Revisit the parameters related to the Coulomb type friction law. The initial state after spin up give a very different grounding line line positions at some fast streams. I wonder if that's due to the initialization method. At some of the fast streams such as Pine island glacier, Coulomb like friction law applies, in which case Cp plays little role. However, Cc is a fixed value from Asay-Davis et al., 2016, in which experiments and parameters are chosen for ideal geometry. Specifically, Cc = 0.5 is chosen to have a continuous basal traction, not necessarily apply to real glaciers. This may also the reason of low sensitivity to basal friction, simply because the nudging parameter doesn't impact the basal traction in the fast flow areas. I suggest more effort on the validation of the parameters related to the basal traction, in order to get the grounding line locations closer to observation.

We are not sure what specific figure or text the reviewer is referencing here, since we do not present or discuss the grounding line position at the end of spin-up in our manuscript.

In any case, we are spinning up to match the observed thickness. In most cases, if we match the thickness, the GL is close to the observed location. It is most important to have a good inversion method – though there are always trade-offs between having an accurate GL location and overfitting the system. In the Pine Island region, the model does have trouble finding a stable position at its current location, where it is retreating. In the spin-up, it often ends up too advanced and too thick. To highlight the differences in the spin-up ensemble, we added a figure (now Figure 5 to the manuscript) showing the ensemble mean spin-up state.

The reviewer raises a good point regarding the sensitivity of the spun-up state to the Coulomb parameter Cc. It is true that lower Cc would lead to faster sliding in locations with lower effective pressure. Cc matters more in the Coulomb regime in which Cp loses its influence (due to the basal sliding law asymptotic behavior). To test the importance of Cc, we ran some additional tests with lower, spatially uniform Cc values of 0.25 and 0.1. (For Cc < 0.1, it is not possible to obtain a spun-up state consistent with observations; most of WAIS is too thin.) We have added a figure to the appendix showing results with these new Cc runs, using low (NESM3), mid (GFDL-ESM4), high (EC-Earth3), and 2C synthetic experiment.

We now address this explicitly in a new Section (3.2.2) (including a new figure in the Appendix):

"This study uses a basal sliding law that allows both power-law and Coulomb behaviors. In our spin-ups, we inverted for the power-law coefficient Cp, keeping the Coulomb parameter Cc fixed

at 0.5. This value of Cc results in spun-up states that match observations well. However, Cc is not necessarily spatially uniform, and its value can influence the length of the transition zone (the zone where basal sliding transitions from power-law to Coulomb behavior). Leguy (2015) (chapter 7.1.4) showed that Cc has limited impact for small p, but influences the length of the transition zone when p>=0.5. We therefore ran additional spin-ups with low $Y_0$ to assess the influence of Cc on ice retreat. With p=0.98 and $Y_0$=1710386, we tested Cc=0.25 and Cc=0.1. (With Cc<0.1, there is widespread WAIS thinning inconsistent with observations.)

We find that lowering Cc has a limited impact in simulations with CMIP6 climate forcing (Figure A10); $Y_0$ remains the dominant parameter. In the 2C synthetic experiment, however, we find that a high p/low $Y_0$ combination with Cc=0.1 leads to a similar sea-level response compared to high p/high $Y_0$ and Cc=0.5. Thus, lowering Cc makes the Amundsen region more prone to retreat. We recommend further study of the sensitivity to Cc in this region. We also suggest inverting for Cc, either instead of or in addition to Cp, when using Eq. 3 or similar sliding laws. To this end, work is underway to study new spin-up strategies using the sliding law proposed in Zoet and Iverson (2020)}, which includes a parameter analagous to Cc but not Cp."

More recently, we have switched to a Zoet-Iverson sliding law where we invert for Cc instead of Cp, which does not appear in the basal sliding formulation.  We have also implemented an inversion scheme in which we invert for deltaT_ocn in every grid cell, instead of just per basin. With these changes, we can match the PIG GL location and thickness more closely.  However, preliminary tests with these updates show that PIG sensitivity to ocean warming is not much changed in the new runs where the GL location is more accurate. Thwaites sensitivity, on the other hand, is increased.  This suggests that the relative insensitivity of PIG is largely a function of the bed geometry rather than the advanced GL in the spin-up.  Please also see more discussion about end-of-spinup GL positions relative to observed in the specific comments below.

These updates have not yet been submitted for publication and were not available at the time of the experiments described in this paper. (There are plans to report on them in the future.)  Once these updates have been implemented and tested more comprehensively, it would be interesting to re-run this suite of simulations.  Until then, we believe the ensemble presented in this paper is still relevant and of interest to the community.

**Specific comments**

 • The second part of the introduction should be in method.
We tried this and found it didn't flow well, so we have decided to keep it as is.  More details on this are below.

• L21: The Antarctic ice sheet...sea level (GMSL). Need to add the time scale.

We have added a timeframe to this sentence. It now reads: "The Antarctic Ice Sheet (AIS) has the potential to contribute multiple meters to global mean sea level (GMSL) on timescales of several centuries."

• L27: 'deep uncertainty' → 'poorly known processes' Changed.

• L29: MICI is not physics. Hydrofracture and cliff failure are the processes that induce MICI. This sentence has been changed to: "One study that included novel physics (eg. hydrofracture and cliff failure leading to Marine Ice Cliff Instability (MICI)) projected much higher 21st century SLR contributions of more than 1 m (DeConto and Pollard, 2016)."

• L47: Put acronym (AIS) the first time 'Antarctic Ice Sheet' is used. Done.

• L69: 'friction' → 'basal friction' Changed.

• L62-L76: The motivation of this study is not clearly described. Lipscomb et al., (2021) explored the two parameters in their previous study using the same model and same spin up. This study is a good supplementary to the previous work, but this work does not add much info to the existed work.
We would argue that this work extends the Lipscomb et al (2021) results in ways that are valuable to the community and TC readers. In this paper, we carry out a more extensive and detailed investigation of the combined impacts of p and gamma than was done in Lipscomb et al. 2021. We find which of these parameters is ultimately more important to sea level predictions on multi-century timescales in different regions, and we discuss the mechanisms at play. We identify the combined parameter space that leads to widespread mass loss after 500 years, which has not been done before.

Furthermore, our new extended (multi-millennial) simulations show that Thwaites collapse is inevitable with many p/gamma settings under the 2°C synthetic forcing, given sufficient time. Lipscomb et al. (2021) ran the model forward for only 500 years and observed Thwaites collapse only with p=1. Our expanded parameter scan shows that there are multiple joint settings where Thwaites could collapse, with p < 1 and a wide range of gamma. We point out that Thwaites could collapse without additional thermal forcing, if the 2°C forcing is interpreted as warming that has already occurred relative to the conditions of the spin-up. The new GIA experiments also add to Lipscomb et al. (2021), who did not simulate any isostatic feedbacks.

• L77-L85: How did you select the CMIP6 models?
The 13 models used in this study were the models available to us at the time of running these simulations. There was no selection process that would rule out the participation based on, for example, model performance. In the methods we wrote:

"TF was computed from 13 CMIP6 climate models and applied as anomalies to each spun-up ice sheet state. Specifically, 3D fields of temperature, salinity and density were extracted from

13 CMIP6 climate models for the high emissions SSP8.5 scenario (Table 4) for two decadally-averaged time slices: 1995-2005 and 2090-2100."

• section 1.1: This section is more suitable for method section? Furthermore, this section could be simplified because some of the ISMIP6 ocean parameterization methods such as PIGL is not used in this study. equation (2): how did you define and calculate the slope? Does it change with time when the geometry changes?

After trying to move this section to Methods, we found that it did not flow as well.  This section is meant to serve as background for the reader, describing the importance of the two parameters being explored in this paper. As such, we leave this section in the Introduction.

We include some discussion here about the PIGL calibration because we do sample gamma values within the PIGL range. Therefore it is important to present the values for this calibration type.  We are including only two melt rate parameterization equations here: (1) is the basic non-local equation, and (2) is the non-local slope-dependent we use throughout the study.

As stated in the text, the slope (theta, equation 2) is defined as the local angle between the ice-shelf base and the horizontal reference.  We added a sentence stating that the slope can change. Line 107 now reads: "The slope can change as the geometry evolves in the simulation."

• L117-L123: Please justify the ocean forcing parameterization method. The authors mentioned that the method in this study results in melt rates that differ appreciably from observational estimates. Can you present 1: the deltaTs needed to have reasonable spin-ups, and what's the difference between spin-ups; 2: melt rates and its variation in between the spin-ups.
The section of text referred to here reads: "To focus computing resources and analysis on one scheme, we choose to limit this study to the slope-dependent non-local form (Eq. 2), since, at the time the ensemble was run, it was believed to be the most realistic scheme (Jenkins et al., 2018). Since we are testing sensitivity to $\gamma 0$, we are not using a specific calibrated parameter range. The simulations in our paper differ from the ISMIP6 protocols in the treatment of $\delta Tsector$. Instead of using the values suggested by Jourdain et al. (2020) to match observational estimates of basal melting in each sector, we tune $\delta Tsector$ to obtain melt rates that drive the ice toward the observed ice thickness near the grounding line, as described by Lipscomb et al. (2021). In some basins, this results in basin-average melt rates that differ appreciably from observational estimates. For more details, see Section 3.1 of Lipscomb et al. (2021)."

First, the deltaT values for the Amundsen sector for each model spin-up (p/gamma combination) are currently shown in Table 3.  This is the region chosen for the table because it requires the largest dT to match observed thicknesses.   Including the values for all sectors would be cumbersome, with 16x25 values in total.  Instead, we have now included a figure (Figure 5) of the mean and standard deviation of dT across the ensemble members. The figure is shown and discussed more in a comment below.

We added clarification in Sec. 2.1 that the thermal forcing used in the spin-up procedure is a climatology based on observation spanning 1995–2018 from Jourdain et al. (2020). Therefore, the difference in melt rates between spin-ups and current observation is partly represented by the change in deltaT.

• L135: $\rho_{sw}$ instead of $\rho_s w$? Corrected, thanks.

• L145: Can you explain the limitations of Lipscomb et al., 2021 that are improved in this study?

Lipsomb et al. (2021) investigated CISM responses to several melt rate parameterisations and only three values of p using the gamma0 calibration from ISMIP6. Here, we focus on only one melt parameterization which is assumed to be the most physically realistic, and not part of the standard ISMIP6 parameterizations. As mentioned in an earlier comment, we expand the scope of the Lipscomb et al. paper by performing a detailed investigation into the relationship in melt rate responses to p and gamma0. In so doing, we get a better sense of the relative contributions of ocean melt rates and effective pressure that lead to Thwaites collapse. The Lipscomb et al. (2021) paper showed that collapse could occur in an extreme case (high warming, p=1 and PIGL calibration), but did not try to map out the parameter space. Our extended simulations show that it is actually possible to trigger Thwaites collapse with p<1. We also infer the importance of each parameter to ocean induced forcing. As part of our revisions, we extend some simulations for multiple millennia to For more on this, please see comments above.

• L153: Reference of ABUMIP should be Sun et al., 2020. Updated, thanks.

• Can you present the model drift of both velocity field and ice geometry? Furthermore, can you present the variation of velocity, ice thickness and grounding line locations among the spin-ups? The initial state of simulations such as different ice geometries could result in essential differences in dynamical responses of the ice sheets.

We recognize that differences in initial ice geometries may result in different dynamical responses.  However, we emphasize that it is most important to have a good inversion method. There are always tradeoffs between having an accurate GL location and overfitting the system. We are spinning up to match the observed thickness, but in most cases, if we get the thickness correct, the GL is close to the observed location.

To show the variation across end-of-spinup states, we have now added a multi-panel figure (Figure 5) to the manuscript in Section 2.2. It shows maps of the end-of-spin-up means and standard deviation (std) across the 25 ensemble members for thickness, velocity and deltaT.  It also shows the difference between the end-of-spin-up mean and observed thickness and velocity. Generally, the deltaT values do not show large variability across spin-ups except for moderately in the Amundsen and Wilkes regions.  These are also the regions that tend to require a larger negative deltaT.  The std in thickness indicates that thickness is more or less the same across ensemble members in the interior, while differences do appear along some

grounding areas. The variability is generally small compared to the absolute thickness. The same is true for velocity.

We have also plotted the grounding lines at the end-of-spin-ups, shown below in Figure 1. It shows all (25) end-of-spin-up grounding lines overlaid on each other in black, with BedMachine observations overlaid as red contours. We have also zoomed in on two regions: the Amundsen and the Ross. On a continental scale, the wide range of gamma and p parameters we use gives similar GL locations. Locally, there can be more variation in GL locations between each spin-up and observations.

In an effort to rank the spin-ups based on GL location or grounded ice area, we found that the results depended too much on the choice of metric and region (i.e., some regions show the spin-ups overestimate grounded ice area, while others regions show it is underestimated). To assess this properly is a much broader issue and is beyond the scope of the current study.

As for the drift in velocity and ice geometry, we are not sure what the reviewer is asking for specifically. In the Control panel in Figure 9, we show that there is minimal sea level variation across spin-ups which is also indicative of minimal variation in the ice sheet state. This implies minimal drift in velocity and ice geometry in the Control run. Furthermore, Figures 4a, b and c all indicate very little drift at the end of spin-up (ice mass, grounded ice mass and grounded ice area). We also impose the condition that there is less than 1% annual change in floating ice area for the continent and just the Amundsen sector at the end of spin-up.

[Figure]

[Figure]

[Figure]

**Fig 1:** Final grounding line positions for all 25 spin-ups (black curves), overlaid on one another. BedMachine observations are overlaid in red. The Amundsen and Ross regions are identified in the top panel. The bottom two panels show these regions in greater detail.

 • L169: delete the sentence 'There is no hydrology in the basal friction field.' because water pressure used in this study is simplified basal hydrology. Done

• L174: Label the Amundsen sector and the Kamb Ice Stream in the map.
Figure 1 has been updated to show the Amundsen sector and Kamb Ice Stream.  Thanks for the suggestion.

• How did you decide the range of the parameter gamma0? It seems (from Figure 3) there is no experiment with both high gamma0 and low p values, what's the advantage of the non-uniform sampling methods?
As stated in Section 2.2, the parameter choices for gamma0 and p were informed by previous work in Lipscomb et al. (2021). The p-value must be, by definition, between 0 and 1.  We preferentially sample p > 0.5 because previous work had shown that the impact of changes when p <0 .5 were less significant.  Gamma was chosen to be within the ISMIP6 range of the MeanAnt and PIGL range. We preferentially chose values closer to the MeanAnt values since the high PIGL values were deemed unrealistic.

On choosing the p-value range and sampling scheme, we write: "From basic physical arguments, p is constrained to be in the range [0,1]. Previous experimental results (Lipscomb  et al., 2021) revealed that the differences in SLR on multi-century timescales between p= 0 and p= 0.5 are smaller than the differences in SLR between p= 0.5 and 1.0.

This suggests that the space could be explored more efficiently by having a greater sampling density for values near 1."

On choosing the gamma-value range and sampling scheme, we write: "Suggested ISMIP6 calibrated median ɣ0 values for the non-local parameterizations are shown in Table \ref{tab:gammas}. The ɣ0 value is closely tied to the physical assumptions. With slope dependence,ɣ0 needs to be about 100 times larger. We develop a distribution of ɣ0 that spans both the MeanAnt and PIGL ranges. We used the distribution $\pi(ɣ0) \propto \frac{1}{(a\gamma_0-1)^2+1}$, bounded on $[1.47 \times 10^6, 1.0 x 10^7]$. We chose $a=3.5 x 10^{-7}$ such that values would fall preferentially within the MeanAnt range rather than the high end of the PIGL range (Figure \ref{fig:joint_sampling}, x-axis). Note that the upper value is truncated to be 10^7 instead of ~ 3 x 10^7 as experimentation suggests that the latter value is far too high (N. Jourdain, personal communication, Nov 12, 2020)."

• L198-L203, Figure 4: – Dash line missing in Figure 4d? – Why grounded ice area and grounded ice mass are different from observation from the beginning? – As mentioned before, can you show the spatial variation of the spin-ups?

The dashed line is intentionally missing from Figure 4d. While it is fair to compare mass and thickness-related metrics to BedMachine because it is our inversion target, we felt it was not relevant to compare the grounding line flux for a few reasons: (1) It is not one of our inversion targets and (2) the current AIS is not in steady state, and we are assuming it is.

The grounded area and grounded ice mass are different from observations at the beginning of the spin-up because each spin-up ensemble is branched from a previous steady-state procedure. We found that this was necessary to ensure a stable inversion procedure for all p and gamma values. In the text, line 195 reads: "Each spin-up is branched from the original spin-up in Lipscomb et al. (2021) (sec. 2.1) and run for at least 10,000 years further." For this reason, the beginning of each of our spin-ups originates at the end-point of a singular previous spin-up.

We have now included maps showing spatial variation (mean and standard deviation) of thickness, velocity and deltaT between spin-ups in the manuscript.

• L222: 'monotonically increase' → 'monotonically change'? According to Figure 6, negative anomalies exist. Yes, thanks. Done.

• Quite a few of climate models present non-linear behaviour in the future thermal forcing change. Why not using the original model results?
It is true that future thermal forcings can have non-linear behavior. However, since this is a sensitivity experiment, we were less concerned with replicating the exact transient thermal forcing trajectories from CMIP models. Rather, we prioritized creating idealized forcing scenarios with which to test the sensitivity to p and gamma. Moreover, we added a constant

forcing extension from 2100 to 2500 which would more or less render any of the more 'realistic' pathways between 1995 and 2100 irrelevant anyway.

• L232: What scenarios do the TF anomalies (1, 1.5 and 2 degrees) represent?
These are not meant to represent specific scenarios.  Rather, these are synthetic simulations, run in an effort to reverse the artificially cooled ocean temperatures in the Amundsen (further details in Sec. 2.4). Because the 2°C anomaly experiments are similar in magnitude to the negative TF corrections, another way to view the 2 degree experiment is that it represents the present-day ASE state. In that case, the 500-year experiments are estimates of committed SLR under warming that has already occurred.

As the text reads: "As discussed in Section 2.3 and in Lipscomb et al. (2021), in order for the spin-up to match the ice sheet's current configuration, a large negative thermal correction was necessary to cool the ocean to prevent retreat. However, there is strong evidence that the recent Amundsen Sea Embayment has been warming (Rignot et al., 2019; Jenkins et al., 2018; Mouginot et al., 2014). Thus, the assumption of an ice sheet at equilibrium may be a bad assumption for the Amundsen sector. Therefore, we ran a set of synthetic experiments targeting the Amundsen region, described in detail in Section 2.4."

 • L266: Figure 7 seems to suggest the differences caused by the parameters increase over time (instead of 'less pronounced after 100 years')?
This is just a wording problem. What we mean to say is that the differences are less pronounced after only 100 years, and become more pronounced the longer you go.  It has been reworded to avoid confusion, and now line 266 reads: "Even though the differences are less pronounced at year 100 than year 500, they still constitute critical impacts on end-of-century sea level contribution estimates."

• L279: 'faster-than-linear' → 'non-linear' Done

• L296-L297: Why smaller p results in higher effective pressure? Seems to be in conflict with equation (4).

Actually, it is not in conflict. One way to see this is to test equation (4) using the extreme cases for p. When p = 0, effective pressure equals overburden pressure (the height-above-flotation term goes to 1 because (1-Hf/H)^0 = 1). When p = 1, then effective pressure = overburden pressure – water pressure, which is smaller than overburden pressure.

• L299-L301: This is contradict with observation. I think some model validation work should be done to explore parameter ranges or processes.
First, the sentence reads: "When we analyze the SLR by region, we find that most CMIP6-forced runs give an SLR signal dominated by ice loss from the Weddell and Ross regions, and to a lesser extent the EAIS (Figure 10)"
This sentence refers to results from future simulations, not to observed current trends -  the Ross and Weddell regions are not where we currently observe large mass loss. These are the

results for future hypothetical scenarios, so we cannot necessarily say they contradict anything, since the future hasn't happened yet.

As discussed earlier in the response to this review, the purpose of this paper is to understand the sensitivity of p and gamma *under the current spin-up framework being used with CISM.* As such, it is not meant to match current observed transient trends, and we openly discuss this caveat in the paper. Rather, we aim to examine the effect of the two important parameters (p and gamma) on future SLR outcomes.

• L351-353: The effect of high topographic seafloor points is not clear to me from Fig. A9. The labelled pinning points are close/higher than sea level. Grounding lines form around them because these mountains are never ice free thanks to surface mass balance.
We think this is a misunderstanding resulting from a colorbar issue. The locations shown by the arrows in figure A9 (now moved to become Figure 13 in the main text, upon the suggestion of another reviewer) are indeed below sea level. Therefore they are pinning points. We have slightly shifted the locations of the arrowheads to help the reader identify the pinning points.

• L409-L410: I think more effort is worth made to improve on constraining the parameters sample range.
The sentence referred to reads: "We reiterate that because we did not use a physically meaningful prior for our p and γ0 ensemble, these predicted SLR ranges should not be over-interpreted." As stated in the paper, our gammas were intentionally sampled across the range used in ISMIP6. P was sampled between 0 to 1, this range constrained by physical arguments. Both of these sampling distributions were weighted to more heavily sample the parameter spaces that were deemed under-explored in previous work. At this point, efforts to constrain the parameters further would be beyond the scope of this study.

• L420: The hypothesis of MISI (or any instability) could only be proved at steady states, which is not the case of this study. Fast retreat doesn't mean MISI.
Our text reads "MISI-type instability". We do not claim that MISI is triggered, and provide no evaluation of such an instability. We are just stating that we observe increased rates of mass loss and retreat once the grounding line is pushed beyond the higher bed topography into the deeper Amundsen Basin.

• L426-427: Again, grounding line hovering around these areas doesn't mean they are stabilizing upstream flow.
The sentences referred to are: "Pinning points affect the ice-sheet stability by acting as an obstacle to ice shelf flow (Still et al., 2019). Our runs show that in the Amundsen, the grounding line tends to stabilize on a few high seafloor ridges." It is possible that the word 'stabilize' is misleading. Instead, we mean to say that the pinning points highlighted in Figure A9 (now figure 13) are enough to impede the fast retreat that is observed once the ice leaves these pinning points. The line has been changed to: "Our runs show that in the Amundsen, high seafloor ridges slow ice retreat by allowing the ice to remain grounded for longer." In our new extended runs (new section 3.2.1) we do find one case (p=0.6), where the GL remains in a stable position

for many millennia (9000 years). In this case, it is likely that the ridges really do stabilize upstream flow.

**References**:

Book, Cameron, Matthew J. Hoffman, Samuel B. Kachuck, Trevor R. Hillebrand, Stephen F. Price, Mauro Perego, and Jeremy N. Bassis. "Stabilizing effect of bedrock uplift on retreat of Thwaites Glacier, Antarctica, at centennial timescales." *Earth and Planetary Science Letters* 597 (2022): 117798.

Coulon, Violaine, Kevin Bulthuis, Pippa L. Whitehouse, Sainan Sun, Konstanze Haubner, Lars Zipf, and Frank Pattyn. "Contrasting response of West and East Antarctic ice sheets to glacial isostatic adjustment." *Journal of Geophysical Research: Earth Surface* 126, no. 7 (2021): e2020JF006003.

DeConto, R., Pollard, D. Contribution of Antarctica to past and future sea-level rise. *Nature* 531, 591−597 (2016). https://doi.org/10.1038/nature17145

Goelzer, H., Coulon, V., Pattyn, F., De Boer, B., & Van De Wal, R. (2020). Brief communication: On calculating the sea-level contribution in marine ice-sheet models. *The Cryosphere*, *14*(3), 833-840.

Kachuck, Samuel Benjamin, Daniel F. Martin, Jeremy N. Bassis, and Stephen F. Price. "Rapid viscoelastic deformation slows marine ice sheet instability at Pine Island Glacier." *Geophysical Research Letters* 47, no. 10 (2020): e2019GL086446.

Larour, E., H. Seroussi, S. Adhikari, E. Ivins, L. Caron, M. Morlighem, and N. Schlegel. "Slowdown in Antarctic mass loss from solid Earth and sea-level feedbacks." *Science* 364, no. 6444 (2019): eaav7908.

Leguy, G.: The effect of a basal-friction parameterization on grounding-line dynamics in ice-sheet models, New Mexico Institute of Mining and Technology, 2015.

Whitehouse, P. L., Gomez, N., King, M. A., & Wiens, D. A. (2019). Solid Earth change and the evolution of the Antarctic Ice Sheet. *Nature communications*, *10*(1), 1-14.

Zoet, L. K. and Iverson, N. R.: A slip law for glaciers on deformable beds, Science, 368, 76–78, 2020.

---

## Author Response (AR2)

Dear Mira Berdahl and co-authors,

First, I want to thank you for answering the pending points that I raised during the revision phase. I am delighted to inform you that I finally received all review reports both suggesting publication in The Cryosphere with technical correction. So you have alleviated most of their initial concerns adequately. There is only one pending remark on the steady-states requirement from reviewer #2. Please consider it.

In conclusion, I decided that the article is forwarded to production in TC with the chance for technical corrections. You will soon be contacted by the responsible production officers.

The editor, Johannes Fürst

We thank the editor for their kind assessment. Below we have noted the pending remark by reviewer 2, and our response to it.

Reviewer #2 comments
Just some thoughts on the authors' reply: 'Importantly, we are not trying to achieve a snapshot of the current transient evolution of the ice sheet. This is a sensitivity study, rather than a predictive SLR assessment. As such, we focus on generating a 'steady state' AIS, so that we can better evaluate the impacts of p and gamma on future mass loss.' I don't think 'steady state' is a necessary requirement for sensitivity study, especially to advise on future mass loss. The sensitivity of future mass loss to parameters are influenced by initial states (Reese et al., 2020; initMIP). Again, I appreciate that additional experiments on the Amundsen basin are conducted, and model validation should probably be a community effort.

In our previous response to editors, we mentioned that the spin-up to a 'steady state' AIS allowed a better evaluation of the impacts of p and gamma on future mass loss. We, however, recognize what the reviewer is mentioning here, in that the initial state of the ice sheet impacts future mass loss. To clarify this to the reader we have now added a parenthetical line in the manuscript, Section 2.2:

"(We acknowledge that a parameter study does depend on the initial state of the ice sheet (Reese et al., 2020) and our current spin-up strategy does not capture the recent observed Antarctic mass change which could impact the results of this study.)"